

4 **Changes in surface ozone in South Korea on diurnal to decadal time scale**

5 **for the period of 2001-2021**

7 Si-Wan Kim[1*], Kyoung-Min Kim[2], Yujoo Jeong[1,2], Seunghwan Seo[2], Yeonsu Park[2], and

8 Jeongyeon Kim[2]

10 [1]Irreversible Climate Change Research Center, Yonsei University, Seoul, South Korea

11 [2]Department of Atmospheric Sciences, Yonsei University, Seoul, South Korea

15 *Corresponding author: Si-Wan Kim (e-mail siwan.kim@yonsei.ac.kr)

20 Short title: Ozone changes in South Korea

**Abstract**

Several studies have reported an increasing trend of surface ozone in South Korea over the

past few decades, using different measurement metrics. In this study, we examined the

surface ozone trends in South Korea by analyzing the hourly or daily maximum 8-hour

average ozone concentrations (MDA8) measured at the surface from 2001 to 2021. We

studied the diurnal, seasonal, and multi-decadal variations of this parameter at city,

province, and background sites.

We found that the 4th highest MDA8 values exhibited positive trends in 7 cities, 9

provinces, and 2 background sites from 2001 to 2021. For the majority of sites, there was

an annual increase of approximately 1-2 ppb. After early 2010, all sites consistently

recorded MDA8 values exceeding 70 ppb, despite reductions in precursor pollutants such

as $NO_2$ and CO. The diurnal and seasonal characteristics of ozone exceedances, defined as

the percentage of data points with hourly ozone concentrations exceeding 70 ppb, differed

between the Seoul Metropolitan Area (SMA) and the background sites.

In the SMA, the exceedances were more prevalent during summer compared to spring,

whereas the background sites experienced higher exceedances in spring than in summer.

This indicates the efficient local production of ozone in the SMA during summer and the

strong influence of long-range transport during spring. The rest of the sites showed similar

exceedance patterns during both spring and summer. The peak exceedances occurred

around 4-5 PM in the SMA and most locations, while the background sites primarily recorded exceedances between 7-8 PM and throughout the night.

During the spring of the COVID-19 pandemic (2020-2021), ozone exceedances decreased at most locations due to significant reductions in $NO_x$ emissions in South Korea and China compared to the period of 2010-2019. The largest decreases in exceedances were observed at the background sites during spring. For instance, in Gosung, Gangwondo (approximately 600 m above sea level), the exceedances dropped from 30% to around 5% during the COVID-19 pandemic.

Regional model simulations confirmed the concept of decreased ozone levels in the boundary layer in Seoul and Gangwon-do in response to emission reductions. However, these reductions in ozone exceedances were not observed in major cities and provinces during the summer of the COVID-19 pandemic, as the decreases in $NO_x$ emissions in South Korea and China were much smaller compared to spring. This study highlights the distinctions between spring and summer in the formation and transport of surface ozone in South Korea, emphasizing the importance of monitoring and modeling specific processes for each season or finer time scales.

## 1. Introduction

Ozone, a greenhouse gas and harmful air pollutant, can accumulate in the lower atmosphere through photochemical reactions involving nitrogen oxides and volatile organic compounds from both human activities and natural sources (National Research Council, 1991; Monks et al., 2015). The increasing concentrations of ozone near the surface and in the troposphere are concerning. Gaudel et al. (2018) reported a significant increase in ozone levels in South Korea from 2000 to 2014, while North America and Europe experienced decreasing trends, using data from surface monitors, ozonesondes, and aircraft observations. Other studies have also observed rising ozone trends in South Korea between 2001 and 2018 in their analysis of the long-term variations of multiple pollutants over Seoul (Kim and Lee, 2018) and South Korea (Kim et al., 2018) or in the reviews of current status and future directions of tropospheric ozone studies in South Korea (Lee et al., 2020) or in the trend estimates of the surface ozone observations (Yeo and Kim, 2021). Ozone in South Korea can be influenced by ozone and its precursor in China (Oh et al., 2010; Lee and Park, 2022; Colombi et al., 2023). However, Gaudel et al. (2018) did not include Chinese data due to a lack of reported information. Recent studies have highlighted a rapid increase in ozone levels in China from 2004 to 2020, especially after 2013 (Li et al., 2019; Wang et al., 2020; Wang et al., 2022). Gaudel et al. (2020) also found that tropospheric ozone in

China and South Korea increased between 1996 and 2016. Considering the proximity of the two countries and their potential for ozone and precursor exchange, it is essential to study the ozone trends in South Korea in relation to those in China. Additionally, as spring and summer have distinct transport patterns and source-receptor relationships relevant to surface and tropospheric ozone (e.g., Cooper et al., 2010), it would be valuable to investigate ozone trends separately for these seasons.

The COVID-19 pandemic brought about significant changes in atmospheric composition (Bauwens et al., 2020; Koo et al., 2020; Seo et al., 2021). Analyzing deviations from long-term trends during the pandemic can provide valuable insights for future environmental policies aimed at mitigating ozone pollution. In this study, we examine ozone trends and exceedances in South Korea from 2001 to 2021, focusing on the warm seasons of spring and summer, including the COVID-19 period. In this study, we analyzed the 4[th] highest daily maximum 8 hours-average ozone concentrations (MDA8 $O_3$) at various locations in South Korea for a global comparison because this is a metric used for the US Environmental Protection Agency National Ambient Air Quality Standard and the recent study by Wang et al. (2022) utilized the same metric for their study of Chinese ozone pollution. We also introduced a new metric of ozone exceedance, defined as the percentage of data points with hourly ozone concentrations exceeding 70 ppb. Previous published

works about surface ozone in South Korea have not focused on the two metrics used in our study. We analyze diurnal, seasonal, and decadal variations at 7 cities, 9 provinces, and 2 background sites. Furthermore, we discuss the factors contributing to the observed temporal changes based on regional model results.

The manuscript is organized as follows. In section 2, the surface and satellite data, global and regional modeling, and other methods to utilize the data are explained. In section 3, the results are summarized as long-term trends of ozone and its precursors, characteristics of diurnal variations, and spatiotemporal variations during the pandemic. The regional model results based on various emission scenarios are also shown to identify the source-receptor relationship. Finally, the results are summarized and future research directions are suggested in the conclusions.

**2. Data and Method**

**2.1. Long-term surface observational data**

The hourly surface air quality monitoring data are obtained from the Airkorea website (https://www.airkorea.or.kr), including ozone ($O_3$), $NO_2$, $SO_2$, CO, $PM_{10}$, and $PM_{2.5}$ ($PM_{2.5}$ data are provided since 2015). As of March 2020, there are about 500 monitoring stations over South Korea. These routine monitor data are available for many decades and can serve

as a main data set to examine long-term trends. We utilized hourly and daily maximum 8

hour-average $O_3$ concentrations. The surface monitoring sites used in this study and the

data availability are summarized in the Supporting Information 1 (SI1, Table S1) and

Supporting Information 2 (SI2). $O_3$, $NO_2$ and CO data are also averaged for spring and

summer months. These surface monitoring data were used to investigate the impact of the

COVID-19 pandemic in the Seoul Metropolitan Area.

## 2.2. Highway toll number and mobile phone usage data

To examine changes in mobility pattern during the COVID-19 pandemic, traffic counts

from the Korea Expressway Corporation daily transit data were used

(http://data.ex.co.kr/portal/). The expressway transit data covering 3 years (2019-2021) of

traffic passing toll gates were quantified from Hi-Pass (electronic toll collection system)

and cash toll collection. Vehicles passing toll gates were not classified in details.

To examine changes in mobility pattern during the COVID-19 pandemic, daily

mobile phone movement provided by Android (Google COVID-19 Community Mobility

Reports, 2020) and Apple (Apple COVID-19 Mobility Trends Report, 2020) are used.

Android mobility data tracked movements of people using cell phones at the same spot,

while Apple's mobility report collects personal vehicle routing requests from Apple Maps.

For Google and Apple mobility report, we used the Transit station Mobility metrics and driving mobility index in Seoul Metropolitan Area, respectively. The reports must be carefully used as it does not directly quantify on-road traffic.

**2.3. Satellite data: tropospheric $NO_2$ columns**

The TROPOspheric Monitoring Instrument (TROPOMI) on board of a low Earth polar orbiting satellite, European Space Agency (ESA) Sentinel-5 Precursor (S-5P) satellite with equator passing time 13:30 local time. The instrument provides measurements at unprecedently high spatial, temporal, and spectral resolutions (Veefkind et al., 2012). In this study we utilized two available tropospheric $NO_2$ datasets from TROPOMI, NASA's standard product (SP) version 4.0 (Lamsal et al., 2021) and KNMI's (Royal Netherlands Meteorological Institute) product obtained from DOMINO v2.0 and QA4ECV v1.1 (Derivation of TROPOMI tropospheric $NO_2$) processing systems (Boersma et al., 2018). The spatial resolution of KNMI's tropospheric $NO_2$ retrieval product is 3.5 km x 7 km (3.5 km x 5.5 km since 6 August 2019) and that of NASA's product is 3.5 km x 5.5 km. Level 2 data with pixels passing quality assurance > 0.75 and the cloud fraction < 0.5 were selected for analysis following recommendations provided by Sentinel-5 precursor TROPOMI Level 2 product User Manual for nitrogen dioxide (Eskes et al., 2019).

TROPOMI data are regridded to a standard grid with a horizontal resolution of 0.1° latitude × 0.1° longitude (11 × 11 km) and monthly averaged values were derived. As the random error in the TROPOMI single-pixel uncertainties influence 40 to 60% of the tropospheric column abundance, temporal and spatial averaging may remove the random errors (Bauwens et al., 2020).

We conducted the sensitivity test by applying different sampling conditions and found consistent results irrespective of quality control parameters: larger tropospheric $NO_2$ column reduction during spring than during summer between 2019 and 2020-2021 (COVID-19 periods). Differences between KNMI and NASA retrievals are large when the the filtering condition of quality assurance > 0.5 and cloud radiance fraction < 0.4 is applied. When the stricter filter is applied, differences between KNMI and NASA retrievals are small. Therefore, the stricter filter (quality assurance > 0.75 and cloud radiance fraction < 0.5) is selected. Since the NASA product released in November, 2022 were generated in a consistent manner for May 2018-December 2021, we mainly present the NASA MINDS product. We summarized the sensitivity tests in the Supporting Information 3 (SI3). The distribution of absolute tropospheric $NO_2$ columns for different years are also shown in the SI3.

**2.4. CAM-Chem model simulations**

The atmospheric component of Community Earth System model (CESMv2.2), Community

Atmosphere Model with Chemistry version 6 (CAM6-chem) is developed by National

Center for Atmospheric Research (NCAR) (https://www2.acom.ucar.edu/gcm/cam-chem).

The CAM-chem adapted MOZART-T1 as the tropospheric chemistry mechanism

(Emmons et al., 2020). The simulation used in this study was configured with 1° horizontal

resolution. The sea surface temperature was prescribed, and meteorological fields were

nudged to Modern-Era Retrospective analysis for Research and Applications version 2

(MERRA-2) instead of using self-produced meteorological field

(https://gmao.gsfc.nasa.gov/reanalysis/MERRA-2/) (refer to SI1 Figure S1 for

performance of the model wind). The simulation was performed from 2000 to 2020 and

applied CMIP6 emission inventory (2000-2014) and SSP5-8-5 emission inventory (2015-

2020). The first 3 years were regarded as a spin-up. In this study, we utilized the CAM-

Chem results to estimate the impact of stratospheric ozone on the surface in each season.

CAM-Chem calculates the contribution of stratospheric ozone to tropospheric ozone, $O_{3S}$

as a three-dimensional variable in space. Originally, $O_{3S}$ is ozone value above tropopause.

Then $O_{3S}$ is transported below tropopause and undergoes chemical losses in the model.

Evaluations of the CAM-Chem results against the data from the ozonesondes that were

launched in Pohang, South Korea are shown in the Supporting Information (SI1, Figure S2;

Jeong et al., 2023). The model results and observations reasonably agree in terms of

seasonal variability and absolute values. Especially, the CAM-Chem results agree with the

observations at the 200 hPa level, close to tropopause.

**2.5. WRF-Chem model simulations**

The Weather Research and Forecasting (WRF) model coupled with Chemistry (WRF-

Chem) is developed by National Oceanic and Atmospheric Administration (NOAA) and

National Center for Atmospheric Research (NCAR) and collaborating institutes (Grell et

al., 2005). We utilized WRF-Chem v4.4 to simulate regional meteorological fields and

chemical compositions.

Our WRF-Chem set up utilizes the horizontal resolution of 28 x 28 $km^2$ and 60

vertical levels. The simulation period is from 24th April 12 UTC to 11th June 12 UTC in

2016. We restart the simulation at 12 UTC every day to reduce computing errors. The first

7 days of model simulation is regarded as spin-up period. The analysis period is selected

as 1st May to 10th June based on local time. The Global Forecast System (GFS) Final (FNL)

analysis data are used for meteorological input and boundary conditions

(https://rda.ucar.edu/datasets/ds083.2/). We used the Community Atmosphere Model with

Chemistry (CAM-Chem) output to the chemical boundary and first initial conditions (https://www.acom.ucar.edu/cam-chem/cam-chem.shtml) (Buchholz et al., 2019). The Model of Emissions of Gases and Aerosols from Nature (MEGAN) is used for biogenic emissions (Guenther et al., 2006).

There are 7 model sensitivity runs that adopt different emission scenarios. The control run is based on the standard EDGAR-HTAPv3 emission inventory representing 2016 (Crippa et al., 2023). Park et al (2021) informed that biomass burning was not an important factor affecting air quality in South Korea during KORUS-AQ. Therefore, biomass burning emissions are omitted in this study. "No China" case removes all anthropogenic emissions in China. "No Seoul" case eliminates all anthropogenic emissions in Seoul. There is one case that decreased Chinese VOC emissions by 50%. There are two cases that reduced Chinese $NO_x$ emissions by 50%: the one case has the same VOC emissions as in the control case while the other case has the 50% reductions of VOC emissions as well. Lastly, there is one case that reduced Chinese $NO_x$ emissions by 75%. The WRF-Chem sensitivity runs are summarized and are discussed in Section 3. The extensive evaluations of the model results against the surface and airborne data from the KORUS-AQ field campaign in 2016 are shown in the Supporting Information (SI1 Table S2-S4 and Figure S3-S8) and Kim K.-M. et al (2023). More discussions about all bottom-

1 up emission inventories utilized for the WRF-Chem simulations and comparison of those

2 with the emissions used for the CAM-Chem simulations are included in SI1 Table S5.

**3. Results**

**3.1. Surface ozone trends**

In this study, ozone and its precursor concentrations in 7 cities, 9 provinces, and 2

background sites in South Korea (Figure 1) are analyzed at diurnal, seasonal, and decadal

time scales. Figure 2 and 3 shows the 4[th] highest daily maximum 8 hours-average ozone

concentrations (MDA8 $O_3$) for the cities and provinces for ozone season (May-September)

from 2001 to 2021. The results from statistical analysis (slope, standard-deviation, p-value,

signal-to-noise ratio) are summarized in Table 1. P-values were presented as suggested by

Chang et al. (2021) and Wasserstein et al. (2019) for the purpose of estimating uncertainties

in trends. The 4[th] highest MDA8 $O_3$ increases by 1.0-1.5 ppb yr$^{-1}$ with very high certainty

for most of cities and provinces across South Korea in this period. In nearly all cities and

provinces, the 4[th] highest MDA8 $O_3$ has been higher than 70 ppb since 2010 or earlier (see

gray dashed line in Figures 2 and 3). The trend in Jeollanam-do (JLN) is small with very

low certainty (p=0.67) partly because the MDA8 $O_3$ was high before 2010. The monitoring

sites in Jeollanam-do include the Yeosu-Kwangyang region in which many petrochemical

industries and iron steel complexes are located. This region experienced severe ozone

problems in the 1990's to early 2000's (Ghim, Y. S. 2000). Because of discontinuity of data

records, the background sites are omitted in the trend analysis in Figure 2 and 3. Although

there are missing data during the ozone season for the background sites, we estimated the

trends of the 4[th] highest MDA8 $O_3$ for a reference (Table 1). The estimated trend for

Ulleung Island is 0.42 ppb $yr^{-1}$ from 2001 to 2021 with low or very low certainty. The trend

for Ulleung Island increases to 0.92 ppb $yr^{-1}$ for 2001-2019 with medium certainty. The

estimated trend for Gosung is 0.73 ppb $yr^{-1}$ from 2001 to 2021 with medium to high

certainty. The trend for Gosung increased to 1.45 ppb $yr^{-1}$ for 2001-2019 with very high

certainty. Widely increasing long-term ozone trends in South Korea indicate a regional

nature of this pollutant, potentially influenced by East Asian emissions, chemical

transformations, and long-range transport (Colombi et al., 2023; Lee and Park, 2022).

Therefore, it is imperative to understand the local and regional processes that enhance

surface ozone. Ozone originated from Asia is known to be efficiently transported to North

America during springtime (Jacob et al., 1999; Jaffe et al., 1999; Jaffe et al., 2003; Cooper

et al., 2010; Lin et al., 2012; Langford et al., 2017; Jaffe et al., 2018) and summertime

(Fiore et al., 2002; Liang et al., 2007) as well. Investigating seasonal differences in ozone

in South Korea may provide insights on the relative importance of local and regional

processes.

**3.2. Difference between spring and summer: background value, exceedance,**

**stratospheric influence, and precursor concentrations**

3.2.1. Background values at the daytime and nighttime periods

Table 2 summarizes the abundances and differences between spring and summer ozone

concentrations averaged for the daytime (10-20 Local Time (LT)) and the nighttime (01-06

LT) periods. Standard deviations denoting year-to-year variability are also shown in the

table. For the nighttime period, the averaged ozone concentration in spring is always higher

than that in summer: differences between the two seasons range from 3.3 to 15.4 ppb and

are generally larger than standard deviations. The differences for the background sites are

approximately 13 ppb at this time. This clearly indicates the importance of large-scale

influences in spring. The results are the same for the daytime period except for Seoul and

Gyeonggi-do: the mean ozone concentrations in Seoul and Gyeonggi-do in summer are

slightly higher than those in spring. The differences in the daytime period are small for

Incheon, Daegu, and Chungcheongbuk-do, suggesting the importance of local chemistry

in the areas during summer. Overall, standard deviations are larger than the differences in

the daytime period except Busan, Jeollanam-do, Jeju Island, Gyeongsangbuk-do,

Gangwon-do, and the background sites. The differences for the background sites in the

daytime period are approximately 13-15 ppb, which are similar to the values in the

nighttime period.

The surface ozone data for 01-06 LT over polluted regions are often omitted in the

analysis because ozone loss reacting with NO is an important process to control ozone

levels at nighttime. In this study, we utilized the ozone data at this time to find information

about background ozone because ozone is transported throughout a day and this process is

essential in the studied region. WRF-Chem sensitivity runs demonstrated increase of ozone

from upwind sources at this time (refer to SI1 Figure S9).

Additionally, we also calculated $O_x$ (=$O_3$ + $NO_2$) that are not affected by the reaction of

ozone with fresh local $NO_X$ emissions during nighttime and compared this value at daytime

and nighttime for each season (SI1 Table S6 and S7 for $NO_2$ and $O_X$, respectively). Overall,

$O_X$ during spring is higher than that during summer and nighttime differences are

prominent, which is similar to the conclusions from the analysis of $O_3$.

3.2.2. Ozone exceedances

Figure 4 illustrates the ratio of summer ozone exceedances to spring ozone exceedances

for the cities, provinces, and background sites. In Seoul, Incheon, and Gyeonggi-do, there

are more exceedances in summer than in spring, indicating the significance of local ozone

production during the summer season in these areas. Conversely, at the background sites

such as Gosung and Ulleung Island, springtime exceedances dominate, highlighting the

importance of high springtime ozone levels and their transport within and beyond Asia. For

the remaining regions, springtime and summertime exceedances are comparable, or

springtime exceedances are slightly higher than those in summer. Note that meteorological

conditions in Seoul and Gyeonggi-do (differences between the two seasons) are similar to

other cities and provinces (see SI1 Table S8). Therefore, the meteorological factors are not

main drivers of high summertime exceedances in Seoul and Gyeonggi-do region.

The diurnal variations of exceedances, as shown in Figure 5, confirm these findings. The summertime ozone exceedances are notably enhanced during the daytime, from 13 to 20 local time (LT), suggesting efficient photochemical ozone production during this season. The peak exceedances occur at 17 LT in Seoul and Gyeonggi-do, and one hour earlier at 16 LT in Incheon. Incheon, being situated adjacent to the West Sea (as depicted in Figure 1), experiences airflow from Incheon to Seoul under typical westerly or seabreeze conditions. The late-afternoon peaks (4-5 PM) in the region and the one-hour delay in peak exceedances in Seoul compared to the time of exceedances in Incheon imply that local circulation plays a significant role in the buildup and distribution of ozone within the Incheon, Seoul, and Gyeonggi-do region.

Springtime and summertime ozone exceedances predominantly occur during the daytime, with some extent of exceedances at night, in Daejeon, Busan, and Daegu (Figure 5). Notably, the peaks in spring occur approximately two hours later than those in summer for the three cities, indicating a potential influence of transport during the spring season. Negligible exceedances are observed from midnight to 10 LT in the three cities due to high NOx pollution and the depletion of ozone associated with $NO_x$ during this time period.

At the background sites, springtime exceedances are much higher compared to summer, and nighttime exceedances are as frequent as daytime exceedances. In Gosung, springtime exceedances account for approximately 20% of the observations throughout the

day, whereas summertime exceedances are less than or equal to 10% (Figure 5). The

observation site in Gosung is located at an altitude of approximately 600 meters above sea

level, providing a unique opportunity to examine long-range transported plumes and

background information at higher altitudes (refer to SI1 Table S9 and Figure S10). Diurnal

variations of exceedances during spring and summer for all individual sites are illustrated

in SI1 (Figure S11-S13).

### 3.2.3. Influence of stratospheric ozone

Stratospheric ozone can deeply intrude into the lower troposphere, leading to elevated

surface ozone levels, particularly during the spring season (Lin et al., 2012; Lin et al., 2015).

It is important to assess the contribution of stratospheric ozone to surface ozone in South

Korea and understand its potential impact on surface ozone trends in the region using

results from the CAM-Chem model. The derivation of the contribution of stratospheric

ozone in the CAM-Chem is explained in the Supporting Information. Figure 6 presents the

contribution of stratospheric ozone to surface ozone in South Korea for each season.

According to our global chemistry-climate model simulations, stratospheric ozone has the

greatest influence on surface ozone during winter and spring, increasing levels by 17-23

ppb. The model suggests that approximately 37% and 76% of surface ozone in spring and

winter, respectively, can be attributed to stratospheric ozone (refer to SI1 Table S10 for

summary of the CAM-Chem results for all seasons at surface and 1 km above ground level).

However, during the summer season, the impact of stratospheric ozone on surface ozone is minimal, accounting for only around 4% of the surface ozone concentration. Therefore, it would be valuable to analyze ozone trends and exceedances separately for spring and summer. It is worth noting that the contribution of stratospheric ozone to surface ozone does not exhibit clear trends during the period from 2001 to 2021 (not shown). Note that the contribution of stratospheric ozone to tropospheric ozone at each altitude and time shown in this study should be a qualitative measure since the representation of this process has uncertainties and needs further assessment.

3.2.4. Long-term trends of surface $NO_2$ and CO concentrations

In contrast to the trends of ozone, $NO_2$ and CO that are ozone precursors decreased both in spring and summer from 2001 to 2021 (Table 3 and 4). There are no systematic differences in the trends of $NO_2$ and CO between the two seasons. $NO_2$ has declined in Seoul, Busan, Daegu, Gwangju, Incheon, Gyeongsangbuk-do, and Gyeonggi-do with very high certainty. For the rest of sites, the declining $NO_2$ trends were found with medium-to-high certainty (refer to Chang et al., 2021 for assessment of uncertainty in the trend analysis). Seo et al. (2021) investigated the trend of $NO_2$ in the Seoul area utilizing satellite tropospheric $NO_2$ columns and surface $NO_2$ observations from 2005 to 2019 and found decrease of $NO_2$ only between 2015 and 2019. They did not find significant trends between 2005 and 2015. Therefore, the trends in our study are strongly influenced by recent $NO_2$ reductions prior to and during the COVID-19 pandemic. CO reductions are evident for a wider region with very high certainty. Only the CO trend in Jeollanam-do was estimated with high certainty

(instead of very high certainty). The decreasing trends of $NO_2$ and CO were estimated with

very low certainty in Ulsan throughout this period. Overall, signs of slopes agree between

emission inventory and ambient concentrations at least for the cities, but site-to-site

variations do not agree even for the cities. There are disagreements of signs of slopes

between emission inventory and ambient concentrations for the provinces (refer to SI1

Table S11 and S12). This can be attributed to the uncertainties in the bottom-up emission

inventories of $NO_x$ and CO in South Korea.

Ozone increases in South Korea despite reduction of main precursors at local scale

can be attributed to the increase of long-range transport of ozone or potentially "VOC-

limited" (or "$NO_x$-saturated") local photochemical regime of South Korea. "VOC-limited"

regime is the condition in which $NO_x$ (sum of NO and $NO_2$) concentration is high and VOC

is a limiting factor to form ozone. In this case, VOC reduction would decrease ozone, while

$NO_x$ reduction would nonlinearly increase ozone. Since long-range transport from China

is frequent during spring, it is useful to identify characteristics of ozone exceedance in

spring separate from summer.

**3.3. Changes detected during the COVID-19 pandemic (2020-2021) compared to**

**2002-2019**

Nationwide social distancing protocol enforced by Korean government started February 25

of 2020 and lasted until April 18 of 2022, although levels of protocol differ. During spring

in 2020 (until May 6, 2020), facilities for public use (libraries, swimming pools, museums,

and national parks) and religious, indoor sports, entertainment facilities were forced to

close, and people were refrain from going out except for buying necessities, visiting a

doctor, and commuting to/from work. Since May 6 of 2020, as number of new confirmed

COVID-19 cases remain relatively steady, the guidelines have shifted from social

distancing to distancing in daily life, no restrictions on people going out. Because a cluster

of new COVID-19 cases emerged in mid-August, social distancing protocol (since August

16 until early October) was again forced by the government, people were strongly

recommended to stay indoors. After August 16 of 2020, there were well-defined

government protocols as Level 1, 2, and 3: Level 1 is no restricted personal gathering and

daily life, Level 2 allows personal gathering up to 8 people and discourage unnecessary

and unurgent travel, and Level 3 allows personal gathering up to 3 people, requires remote

work and online classes, and discourage travels. Most days in spring and summer in 2021

were the period under the Level 2 protocol. In summary, most distinct changes in social-

distancing protocols and traffic/mobile activities occurred between spring and summer in

2020 in South Korea (refer to SI1 Figure S14-15).

3.3.1. Changes in ozone exceedances and local precursors during springtime

The frequency of springtime ozone exceedances increases from period P1 (2002-2010) to

period P2 (2011-2019) across all observation sites in South Korea (Figure 7). However,

during the COVID-19 period (P3: 2020-2021), the frequency of exceedances significantly

decreases at most sites. Notable reductions are observed in Daejeon, Daegu, Chungcheongbuk-do, Gyeongsangnam-do, Gyeongsangbuk-do, Gangwon-do, as well as the background sites Gosung (Gangwon-do) and Ulleung Island. In Gosung, the percentage of ozone exceedances drops from 30% during P2 to 5% during P3 in spring. Although Gosung is located close to the East Sea and is the region farthest from China within a similar latitude range, it is still susceptible to long-range transported ozone due to its high elevation (see SI1 Figure 10 for the elevation map and diagram of a possible ozone transport path).

Across all sites, the concentration of $NO_2$ shows little change from P1 to P2, with an average decrease of 5%. However, during the COVID-19 period (P2 to P3), there was an average reduction of 25% in $NO_2$ concentrations. CO concentrations also experienced a decrease of 22% from P1 to P2 and a further decrease of 14% from P2 to P3. However, the reductions in CO are relatively minor compared to the changes in $NO_2$ observed during the COVID-19 period. The decrease in ozone exceedances during COVID-19 may be associated with the reductions in $NO_2$ concentrations during this time.

A notable finding is the significant reduction in ozone levels at the background sites, such as Gosung and Ulleung Island, between P2 and P3. This suggests a cleaner background influenced by changes in emissions from sources in Asia and long-range transport. It is important to note that there were no significant changes in $NO_2$ and CO concentrations observed at the background sites from P2 to P3. There are several studies

reporting the increase of near-surface ozone after COVID lockdowns in the urban areas

(e.g., Shi & Brasseur, 2020) because of expected non-linear relationship between ozone

and $NO_x$ in the highly polluted regions. However, there are also studies reporting reductions

of ozone concentrations from 1 to 8 km altitude in the northern extra-tropics during COVID

(Steinbrecht et al., 2021). Parrish et al. (2020) reported zonal similarity of tropospheric

ozone changes at northern mid-latitudes. Therefore, ozone reductions from P2 to P3 across

the sites in South Korea may be associated with decreased background ozone at northern

mid-latitudes to some extents. On top of this, local and regional emission changes during

COVID may also play a role in reducing ozone exceedances in South Korea in this season.

3.3.2. Changes in ozone exceedances and local precursors during summertime

During summer, ozone exceedance frequencies also increase from P1 to P2 for all sites:

Chungcheongnam-do has the largest increase from 3.2% to 11.3% and Gyeonggi-do,

Daejeon, Jeollabuk-do, Gyeongsangnam-do and Gyeongsangbuk-do have similar increases

(Figure 8). The ozone exceedances in the background sites Gosung, and Ulleung Island

also increase in this period. $NO_2$ and CO concentrations decreased marginally from P1 to

P2. During COVID-19, the ozone exceedance frequencies in summer increase in Seoul,

Incheon, Gyeonggi-do, and Chungcheongnam-do, substantially decrease in Gangwon-do

and the background sites, and does not show changes from P2 for the rest of sites. Because

$NO_2$ concentrations decrease from P2 to P3 for Seoul, Incheon, Gyeonggi-do, and

Chungcheongnam-do contrasting with increases of ozone exceedance, chemical regime for

these regions during summer is likely to be VOC-limited ($NO_x$-saturated) as mentioned

above and as in previous studies (e.g., Kim et al., 2020). Ozone exceedance substantially

decreases in the background sites from P2 to P3 during summer, indicating cleaner air at

large-scale as shown in Steinbrecht et al. (2021).

3.3.3. Changes in precursor concentrations at a regional scale during spring and summer:

TROPOMI tropospheric $NO_2$ columns

Figure 9 presents the spatial distributions of NASA TROPOMI tropospheric $NO_2$ columns

(Lamsal et al., 2022) in spring (MAM) and summer (JJA) across East Asia, along with their

changes from 2019 to 2020 and from 2019 to 2021. The plot illustrates significant

reductions in $NO_2$ columns during the spring of COVID-19 in most areas of China, South

Korea, and the surrounding seas. Changes in traffic activities in the Seoul Metropolitan

Area were also detected between 2019 and 2020 (refer to SI1 Figure S14 and 15). The

number of cars counted at the highway tolls in this region decreased by 6% in March, April,

and May in 2020 compared to 2019, but this trend was reversed in June (SI1 Figure S14).

Furthermore, observed concentrations of $NO_2$, $SO_2$, CO, $PM_{10}$, and $PM_{2.5}$ during the spring

of 2020 showed reduction of 15-30% (SI1 Figure S15). Changes in traffic counts in the

Seoul Metropolitan Area between 2019 and 2021 were small (SI1 Figure S14 and S15).

But observed concentrations of $NO_2$, $SO_2$, CO, $PM_{10}$, and $PM_{2.5}$ were also reduced during

spring in 2021 compared to 2019 by 10-30% except for $PM_{10}$ that was enhanced due to

Asian dust events in spring 2021 (SI1 Figure S15).

As depicted in Figure 9, TROPOMI tropospheric $NO_2$ columns also decreased

during the summer in the same region, although in fewer locations and to a lesser extent

compared to the spring, during the COVID-19 period. The observed $NO_2$, $SO_2$, CO, $PM_{10}$,

and $PM_{2.5}$ concentrations in the Seoul Metropolitan Area were also reduced during summer

in 2020 or 2021 compared to 2019 by 2-20%. Surface $NO_2$ concentrations reduced by ~10%

during summer, which is smaller than the reductions during spring (~20%; see SI1 Figure

S14 and S15). Overall, the reductions in 2020/2021 from 2019 during summer are smaller

than those during spring at the surface in the Seoul Metropolitan Area, which is similar to

the seasonal changes detected from space.

The substantial decrease in $NO_2$ in China during spring, observed by satellite, is

likely to contribute to significant reductions in ozone levels in South Korea due to long-

range transport. Additionally, local reductions in $NO_x$ emissions in South Korea can lead

to ozone decreases if the reductions are significant enough, especially in the "VOC-limited"

chemical regime prevalent in this area. However, further investigation is required to

understand the detailed source-receptor mechanism of ozone and its precursors in each

season, which warrants long-term air quality model simulations in future studies. The next

section of this study discusses the sensitivity of ozone concentrations in Seoul and

Gangwon-do to various emission scenarios in China and South Korea, albeit within a

limited time period.

### 3.4. Impacts of changes in East Asian emissions on surface/boundary layer ozone in South Korea: a modeling analysis

3.4.1. Changes in surface/boundary layer ozone due to emission reductions: East Asian region

In this section, we will discuss WRF-Chem model simulations conducted during the KORUS-AQ 2016 field campaign (primarily in May; refer to Crawford et al., 2021 for detailed information) to gain insights into the impacts of emission changes on ozone concentrations in East Asia, including South Korea. We have extensively evaluated our model results with the airborne and surface observations acquired during the KORUS-AQ campaign and the routine surface monitors in China and South Korea. The model decently simulated boundary-layer ozone over South Korea (3% difference) for the cases that were strongly influenced by long-range transport. For local emission dominating cases, the model underestimated boundary-layer ozone over South Korea by 20%. The results are summarized in SI1 (Table S3 and Figure S8) and Kim K.-M. et al. (2023). This study considers two extreme cases: the "No China" case, where all anthropogenic emissions in China are removed, and the "No Seoul" case, where all anthropogenic emissions in Seoul are removed. Additionally, several other scenarios are examined, including a 50% reduction in Chinese $NO_X$ emissions only, a 50% reduction in Chinese VOC emissions only, a 50%

reduction in both Chinese $NO_X$ and VOC emissions, and a 75% reduction in Chinese $NO_X$ emissions only.

Our study reveals both increases and decreases in ozone concentrations due to emission changes resembling those during the COVID-19 period. Specifically, near-surface ozone concentrations in polluted regions increase, while ozone concentrations in the elevated layer show reductions (refer to Figures 10 and 11). A novel finding is the decrease in downwind ozone, from the near surface to the upper layer, resulting from reductions in $NO_X$/VOC emissions in upwind pollution hotspots (refer to Figures 10 and 11 for several sensitivity runs). For instance, a 50%-75% reduction in Chinese $NO_X$ emissions leads to decreased ozone concentrations in Korea, surrounding seas, and the Pacific Ocean, from the surface to the upper layers. However, near-surface ozone in Northeast China increases due to these emission changes.

Reductions in Chinese VOC emissions result in decreased ozone concentrations from the surface to the upper layer and from hotspots to downwind areas. Our study suggests potential changes in photochemical regimes with altitude over pollution hotspots, indicating $NO_X$-saturated conditions near the surface and $NO_X$-limited conditions in the elevated layer. Thus, the combined effects of vertical and horizontal ozone transport, as well as local production dependent on altitude, would determine the ultimate changes in ozone concentrations at specific locations and altitudes. It is important to note that the

1 accuracy of VOC emission estimates also influences the assessment, but this aspect is

2 highly uncertain and requires further study.

3.4.2. Vertical sensitivity of ozone changes in South Korea to East Asian emission

reductions

Figure 11 presents the vertical profiles of simulated ozone concentrations for different

emission scenarios. In Seoul, the 50% reduction in Chinese $NO_X$ emissions only slightly

decreases ozone concentrations near the surface but decreases them above 500 m AGL

(above ground level) to a larger extent. The 50% reduction in Chinese VOC emissions

causes a decrease in ozone concentrations from the surface to 2000 m AGL. In the elevated

layer (> 1500 m AGL) in Seoul, the reduction in Chinese $NO_X$ emissions leads to a greater

decrease in ozone concentrations compared to the reduction in Chinese VOC emissions.

The scenario with a 50% reduction in both Chinese $NO_X$ and VOC emissions efficiently

decreases ozone concentrations from the surface to 2000 m AGL, particularly above 1000

15 m AGL. The scenario with a 75% reduction in $NO_X$ emissions decreases ozone

concentrations near the surface similarly to the scenarios with a 50% reduction in $NO_X$ and

VOC emissions, but it causes the largest ozone reductions above 1000 m AGL, except for

the "No China" emission scenario. The "No China" emission scenario results in ozone

concentrations 10-15 ppb lower than the control case at all altitudes. On the other hand, the

"No Seoul" emission scenario leads to ozone concentrations about 20 ppb higher than the

control case near the surface, partly due to significantly reduced ozone depletion reactions

with NO. The sensitivity test results for Seoul and Gosung, Gangwon-do are similar, except

that all emission scenarios (including "No Seoul" and 50% reduction in Chinese $NO_X$

scenarios) cause a decrease in ozone concentrations in Gangwon-do. Both $NO_X$ and VOC

emission reductions in China contribute to cleaner air in Gangwon-do, with the largest

cleaning effect observed above 500 m AGL. This may explain the sharp decline in ozone

exceedances observed in Gosung, located at an elevation of approximately 600 m AGL,

during the COVID-19 pandemic (Figure 7). Refer to SI1 (Table S6 and Figure S10) about

altitudes of monitoring sites in Gangwon-do including Gosung. The sensitivity runs clearly

demonstrate the long-range transport of Chinese ozone or the influence of Chinese

emissions on the eastern part of the Korean Peninsula, such as Gangwon-do, from May to

the beginning of June 2016. Both reductions in Chinese VOC emissions and $NO_X$

emissions contribute to improving ozone pollution in the boundary layer (1-3 km) in South

Korea.

3.4.3. Comparisons with recent modeling research

Lee and Park (2022) investigated seasonal differences in ozone utilizing a chemical

transport model. They reported the April mean ozone concentration of 39.3 ppb, which is

slightly higher than the July counterpart (38.3 ppb) from their model simulations for the

year 2016 and the selected surface monitor sites for 4 main regions (Seoul,

Chungcheongbuk-do, Gwangju, and Busan). Our study summarizes the differences between spring (March, April, May) and summer (June, July, August) for 21 years including 192 monitoring sites covering the whole of South Korea focusing on the analysis of long-term surface ozone observations. On overage, the observed spring mean ozone is 34.3 ppb and the summer mean ozone is 29.0 ppb over South Korea in our study. Lee and Park (2022) indicated that ozone air quality in South Korea is determined mainly by year-round regional background contributions (peak in spring). With some differences in details, the results from the two studies are qualitatively similar arguing high springtime background ozone value. One unique aspect of our modeling study is demonstrations of the impact of emissions in Seoul on Gangwon-do, causing slight ozone decrease in Gangwon-do with zero-Seoul emissions from surface to 2 km in May 2016. Our study highlights the diverse impacts of surface emission changes (over China or Seoul) on downwind ozone at different altitudes (Figure 11).

Colombi et al. (2023) performed an analysis on the effect of precursor changes on observed surface ozone increases in South Korea. A main difference between Colombi et al. (2023) and our study is the period of the study and whether it focuses on the surface ozone or vertical sensitivity explaining ozone variability at different locations in South Korea. Our study investigated surface ozone and ozone at various altitudes to consider the transport within and above the boundary layer between China and South Korea. Colombi et al. (2023) analyzed the surface ozone and $NO_2$ concentrations mainly over the Seoul

Metropolitan Area from 2015 to 2019. The increase of ozone was mostly attributed to decrease in $NO_2$ for the studied period.

Both Lee and Park (2022) and Colombi et al. (2023) indicated high background ozone concentration external to East Asia (or South Korea), suggesting difficulty of achieving ozone standards. Our study agrees to this point. Probably one different message is that reducing emissions of $NO_x$ and VOC here and there all together have positive impacts on reducing ozone downwind. For example, emission reductions associated with the COVID-19 would lead to decrease of ozone at most sites over South Korea in spring. However, our study also indicates that the ozone exceedances notably increased in SMA and Chungcheongnam-do (where large mobile, industrial, and power plant emission sources are located) during the summer of the COVID-19 pandemic.

**4. Conclusions**

We conducted a study on the spatiotemporal variability of surface ozone in 7 cities, 9 provinces, and 2 background sites in South Korea from 2001 to 2021. The 4[th] highest maximum daily 8-hour average (MDA8) ozone concentrations showed an increasing trend in all cities, most provinces, and background sites during this period (or 2001 to 2019), with a yearly increase of 1-2 ppb. After 2010, these concentrations reached approximately 70 ppb or higher. If the US EPA National Ambient Air Quality Standards were applied, most of the monitoring sites in South Korea would have been considered nonattainment

areas for the past decade.

Ozone exceedances in this study were defined as the ratio of data with

concentrations exceeding 70 ppb to the total data, which aligns with the US EPA standard.

In Seoul, Incheon, and Gyeonggi-do, ozone exceedances were more frequent in summer

than in spring. However, the opposite trend was observed in Daejeon, Gwangju, Jeollanam-

do, Gyeongsangbuk-do, Gangwon-do, Jeju Island and the background site Gosung and

Ulleung Island. In other areas, the frequencies of exceedances were similar between spring

and summer. The majority of ozone exceedances occurred between 16-19 LT (4-7 PM).

Interestingly, exceedances also occurred frequently at night in background sites such as

Gosung and Ulleung Island, indicating a strong influence of long-range transport on surface

ozone levels in these locations.

Ozone exceedances increased from period P1 (2002-2010) to period P2 (2011-2019)

across all observation sites in South Korea during spring and summer. Overall, $NO_2$

concentrations showed declining trends from 2001 to 2021, but significant and relatively

large decreases were only evident after the mid 2010s. $NO_2$ concentrations for P1 and P2

were similar and increase of CO/VOC concentrations between the two periods were not

detected or reported. Therefore, it is not clear what drove increase of ozone exceedances

over South Korea from P1 to P2. The observed increase in ozone exceedances from P1 to

P2 in South Korea may be partially attributed to the rise in anthropogenic emissions

originating from China. More modeling experiments covering the P1 to P2 period would

help identify the main factors behind the ozone increases. It is important to investigate not only changes in anthropogenic emissions but also the impact of climate change on ozone variations during this period. We observed significant reductions in ozone exceedances across almost all monitoring sites in South Korea during the spring of the COVID-19 pandemic (period P3, 2020-2021), which was attributed to decreased anthropogenic activities and subsequent lower emissions in both China and South Korea. It should be noted that ozone exceedances substantially increased in SMA and Chungcheongnam-do during the summer of the COVID-19 pandemic. We conducted sensitivity tests using a regional chemical model to investigate the impact of emission changes on ozone pollution in South Korea for a limited period in spring. The results suggest that reductions in Chinese $NO_X$ emissions as well as VOC emissions can contribute to the improvement of ozone pollution in South Korea. These findings provide valuable insights for future efforts to address ozone pollution in South Korea and emphasize the need for further research to project air quality and prioritize actions for the next decade or so.

In the future, employing multidecadal mathematical modeling on a local to global scale in both hindcast and forecast modes would be beneficial for better understanding ozone trends in South Korea. Additionally, reliable VOC observations and conducting intensive field campaigns, similar to the KORUS-AQ 2016, would provide crucial information to unravel the complexities of ozone chemistry in this region and facilitate the careful monitoring of changes in atmospheric composition relevant to ozone. Monitoring

ozone within the boundary layer and at higher altitudes is crucial for enhancing our understanding of ozone trends in South Korea. A network of ozonesondes at multiple sites with the capability of weekly launches would be a valuable complement to a large field campaign.

**Code/Data availability**

- The surface monitor data for South Korea can be downloaded from https://www.airkorea.or.kr/web/.
- Korea Expressway Corporation transit data: Daily traffic counts using highway, available at: http://data.ex.co.kr/portal/, last access: 31 December 2022.
- KORUS-AQ data: NASA/LARC/SD/ASDC. (2022). KORUS-AQ Aircraft Merge Data Files [Data set]. NASA Langley Atmospheric Science Data Center DAAC. Retrieved from https://doi.org/10.5067/ASDC/SUBORBITAL/KORUSAQ_Merge_Data_1
- NASA TROPOMI $NO_2$ columns are available at https://disc.gsfc.nasa.gov/datasets/TROPOMI_MINDS_NO2_1.1/summary?keywords=tropomi%20no2.
- KNMI TROPOMI $NO_2$ columns are available at https://disc.gsfc.nasa.gov/datasets/S5P_L2__NO2____HiR_2/summary?keywords=tropomi%20no2.

- CAM-Chem (CESM) code is available at

    [https://www.cesm.ucar.edu/models/cesm2/release_download.html](https://www.cesm.ucar.edu/models/cesm2/release_download.html).

- WRF-Chem model can be downloaded from

    [https://www2.mmm.ucar.edu/wrf/users/download/get_sources.html](https://www2.mmm.ucar.edu/wrf/users/download/get_sources.html).

**Author contribution**

SWK initiates, designs, analyzes surface monitor data, and writes the manuscript, KMK, SHS, and SWK design and conduct WRF-Chem model runs, JYJ, JYJ, and SWK design and conduct CAM-Chem model runs, SHS processes the airkorea data, YSP and SHS process, analyze, and visualize TROPOMI data, and YSP and JYJ collect and analyze the highway traffic data. All authors edit the manuscript.

**Competing interests**

Authors declare no competing interests.

**Acknowledgements**

This subject is also supported by the National Research Foundation of Korea (NRF) grant funded by the Korea government (MSIT) (No. 2020R1A2C2014131). The first author also acknowledges support from NRF-2018R1A5A1024958. KMA and KISTI supercomputing center support computing resources used (KSC-2021-RND-0040). We thank Louisa Emmons for helping with setting CAM-Chem. The National Center for Atmospheric

1    Research (NCAR) is sponsored by the National Science Foundation (NSF)

2    (NNX16AD96G). We also thanks to the KORUS-AQ science team for producing the

3    extensive number of observations from ground to aircraft.

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

**List of Tables**

Table 1. Trend estimates based on the 4th highest MDA8 O3 values from 2001 to 2021. The data were acquired from the surface monitoring network (www.airkorea.or.kr). Unit of slope and limit (2 sigma = 2 standard deviation) is ppb yr-1. SNR denotes signal-to-noise ratio defined as the ratio of absolute value of slope to standard deviation. For the use of P-value and SNR, refer to Chang et al. (2021). Data during typical ozone season (May-September) are analyzed. Some data for the background sites are missing (refer to SI2).

Table 2. Spring and summer ozone concentrations in Korean metropolitan cities and provinces. Both daytime (10-20 LT) and nighttime (01-06 LT) averages are shown. Standard deviations (S.D.) denoting year-to-year variability are shown in the parenthesis. Differences in concentrations between spring and summer ($O_{3\ spring}$ - $O_{3\ summer}$) are in the bracket. The cities and provinces listed in the table are in counterclockwise order in regards to the South Korean map. The data from 2002 to 2019 are utilized.

Table 3. The observed trends of $NO_2$ concentrations in spring and summer from linear fits of the data covering 2001-2021. The data were acquired from the surface monitoring network (www.airkorea.or.kr). Unit of slope and limit (2 sigma = 2 standard deviation) is ppb yr$^{-1}$. SNR denotes signal-to-noise ratio defined as the ratio of absolute value of slope to standard deviation. For the use of P-value and SNR, refer to Chang et al. (2021). Due to data limitations, the trend for the background sites could not be calculated.

Table 4. The observed trends of CO concentrations in spring and summer from linear fits of the data covering 2001-2021. The data were acquired from the surface monitoring

1  network ([www.airkorea.or.kr](http://www.airkorea.or.kr)). Unit of slope and limit (2 sigma = 2 standard deviation) is

2  ppb yr$^{-1}$. SNR denotes signal-to-noise ratio defined as the ratio of absolute value of slope

3  to standard deviation. For the use of P-value and SNR, refer to Chang et al. (2021). Due to

4  data limitations, the trend for the background sites could not be calculated.

1 **Figure captions**

Figure 1. The locations of cities, provinces, and background sites in South Korea. The red,

black, and blue color denote city, province, and background site, respectively: Cities – SUL

(Seoul), INC (Incheon), DJN (Daejeon), GWJ (Gwangju), BSN (Busan), ULS (Ulsan),

DGU (Daegu); Provinces – GGI (Gyeonggi-do), CCB (Chungcheongbuk-do), CCN

(Chungcheongnam-do), JLB (Jeollabuk-do), JLN (Jeollanam-do), JEJ (Jeju Island), GSN

(Gyeongsangnam-do), GSB (Gyeongsangbuk-do), GWO (Gangwon-do);   Background

sites – ULL (Ulleung Island), and GSU (Gosung, Gangwon-do).

Figure 2. The trend of the 4[th] highest daily maximum 8 hours average (MDA8) $O_3$

concentrations in the South Korean metropolitan cities from 2001 to 2021. Only the data

for May-September (ozone season) are used. Bars denotes standard deviations among the

sites within the city. The slopes (S) and correlation coefficients (r) from linear fits are

shown in parentheses. Grey dashed line indicates 70 ppb that is the air quality standard

defined by the US Environmental Protection Agency.

Figure 3. The same as in Figure 2 except for South Korean provinces.

Figure 4. Ratio of $O_3$ exceedances in summer to exceedances in spring. The exceedances

are defined as the fraction of the data with hourly ozone concentration greater than 70 ppb

among all available data.The red line indicates an one to one line. X-axis denotes names of

cities, provinces, and background sites. Cities – SUL (Seoul), INC (Incheon), DJN

(Daejeon), GWJ (Gwangju), BSN (Busan), ULS (Ulsan), DGU (Daegu); Provinces - GGI (Gyeonggi-do), CCB (Chungcheongbuk-do), CCN (Chungcheongnam-do), JLB (Jeollabuk-do), JLN (Jeollanam-do), JEJ (Jeju Island), GSN (Gyeongsangnam-do), GSB (Gyeongsangbuk-do), GWO (Gangwon-do);   Background sites - ULL (Ulleung Island), and GSU (Gosung, Gangwon-do). The data for 2002-2019 are utilized.

Figure 5. Diurnal $O_3$ exceedances over selected sites. (Top) cities and province in Seoul Metropolitan Area, (middle) other cities, (bottom) background sites. The data for 2002-2019 are utilized.

Figure 6. The contribution of stratospheric $O_3$ ($O_{3s}$) to the $O_3$ concentrations in each season at surface and 1 km above ground level in South Korea. The plotted values are extracted from the CESMv2.2 results for the entire country.

Figure 7. (Top) $O_3$ exceedances (%), (middle) $NO_2$, and (bottom) CO concentrations in South Korean cities, provinces, and background sites during spring for 2002-2010, 2011-2019, and 2020-2021 (COVID-19). X-axis denotes names of cities, provinces, and background sites. Citis - SUL (Seoul), INC (Incheon), DJN (Daejeon), GWJ (Gwangju), BSN (Busan), ULS (Ulsan), DGU (Daegu); Provinces - GGI (Gyeonggi-do), CCB (Chungcheongbuk-do), CCN (Chungcheongnam-do), JLB (Jeollabuk-do), JLN (Jeollanam-do), JEJ (Jeju Island), GSN (Gyeongsangnam-do), GSB (Gyeongsangbuk-do), GWO (Gangwon-do);   Background sites - ULL (Ulleung Island), and GSU (Gosung, Gangwon-do).

1    Figure 8. The same as Figure 7 except for summer.

Figure 9. Differences in TROPOMI tropospheric $NO_2$ columns between 2019 and 2020 or

between 2019 and 2021 (Difference = $NO_{2\ 2020\ or\ 20-1}$ - $NO_{2\ 2019}$). Unit: molecules $cm^{-2}$. Wind

vectors at 700 hPa from ERA-5 are shown for MAM and JJA, respectively. Here, ERA-5

denotes the European Centre for Medium-range Weather Forecasts Reanalysis $5^{th}$

generation (DOI: 10.24381/cds.6860a573).

Figure 10. Differences in the WRF-Chem simulated ozone concentrations ($\Delta O_3$ =

$O_3$_emission reduction case-$O_3$_control case) at (top) surface and (bottom) 1000 m above

ground level. Green to blue colors (yellow to red colors) denotes reduced (increased) ozone

concentration due to the emission changes. All model simulation results are utilized.

Figure 11. Vertical profiles of ozone from the WRF-Chem model simulations based on

various emission scenarios: (top) Seoul, and (bottom) Gosung, Gangwon-do. The model

results from 10 LT to 20 LT are averaged.

Table 1. Trend estimates based on the 4th highest MDA8 $O_3$ values from 2001 to 2021. The

data were acquired from the surface monitoring network (www.airkorea.or.kr). Unit of

slope and limit (2 sigma = 2 standard deviation) is ppb yr$^{-1}$. SNR denotes signal-to-noise

ratio defined as the ratio of absolute value of slope to standard deviation. For the use of P-

value and SNR, refer to Chang et al. (2021). Data during typical ozone season (May-

September) are analyzed. Some data for the background sites are missing (refer to SI2).

| Location | | Slope (ppb yr$^{-1}$) | 2-Sigma (ppb yr$^{-1}$) | P value | SNR |
|---|---|---|---|---|---|
| City | Seoul (SUL) | 1.19 | 0.38 | < 0.01 | 6.23 |
| | Incheon (INC) | 1.07 | 0.37 | < 0.01 | 5.72 |
| | Daejeon (DJN) | 1.22 | 0.49 | < 0.01 | 4.96 |
| | Gwangju (GWJ) | 0.98 | 0.46 | < 0.01 | 4.30 |
| | Busan (BSN) | 0.98 | 0.36 | < 0.01 | 5.47 |
| | Ulsan (ULS) | 1.40 | 0.34 | < 0.01 | 8.14 |
| | Daegu (DGU) | 1.12 | 0.46 | < 0.01 | 4.89 |
| Province | Gyeonggi-do (GGI) | 1.26 | 0.27 | < 0.01 | 9.33 |
| | Chungcheongbuk-do (CCB) | 0.79 | 0.51 | < 0.01 | 3.09 |
| | Chungcheongnam-do (CCN) | 1.45 | 0.47 | < 0.01 | 6.12 |
| | Jeollabuk-do (JLB) | 1.83 | 0.32 | < 0.01 | 11.30 |
| | Jeollanam-do (JLN) | 0.08 | 0.39 | 0.67 | 0.41 |
| | Jeju Island (JEJ) | 0.74 | 0.50 | < 0.01 | 2.96 |
| | Gyeongsangnam-do (GSN) | 0.83 | 0.52 | < 0.01 | 3.18 |
| | Gyeongsangbuk-do (GSB) | 1.10 | 0.35 | < 0.01 | 6.32 |
| | Gangwon-do (GWO) | 0.74 | 0.46 | < 0.01 | 3.22 |
| Background | Ulleung Island (ULL) (2001-2019) | 0.42 (0.92) | 0.84 (0.92) | 0.34 (0.06) | 1.00 (2.00) |
| | Gosung (GSU) (2001-2019) | 0.73 (1.45) | 0.82 (0.75) | 0.09 (<0.01) | 1.78 (3.88) |

Table 2. Spring and summer ozone concentrations in Korean metropolitan cities and provinces. Both daytime (10-20 LT) and nighttime (01-06 LT) averages are shown. Standard deviations (S.D.) denoting year-to-year variability are shown in the parenthesis. Differences in concentrations between spring and summer ($O_{3\ spring}$ - $O_{3\ summer}$) are in the bracket. The cities and provinces listed in the table are in counterclockwise order in regards to the South Korean map. The data from 2002 to 2019 are utilized.

| Location | | Daytime period<br>Spring / Summer<br>(S.D.) [Difference] | Nighttime period<br>Spring / Summer<br>(S.D.) [Difference] |
|---|---|---|---|
| City | Seoul (SUL) | 34.7(5.4) / 35.5(6.0) [-0.8] | 20.5(3.8) / 17.0(4.3) [3.5] |
| | Incheon (INC) | 34.7(4.0) / 32.7(4.8) [2.0] | 25.1(2.9) / 19.7(3.8) [5.4] |
| | Daejeon (DJN) | 41.7(6.4) / 36.7(7.1) [5.0] | 22.9(3.3) / 18.7(3.8) [4.2] |
| | Gwangju (GWJ) | 40.1(6.0) / 35.2(5.7) [4.9] | 28.5(5.3) / 23.6(4.2) [4.9] |
| | Busan (BSN) | 40.5(4.2) / 33.7(4.2) [6.8] | 30.3(3.1) / 21.9(3.0) [8.4] |
| | Ulsan (ULS) | 38.9(5.8) / 33.1(6.1) [5.8] | 25.7(2.9) / 18.2(3.2) [7.5] |
| | Daegu (DGU) | 40.0(5.8) / 37.3(5.9) [2.7] | 24.0(3.8) / 19.1(3.5) [4.9] |
| Province | Gyeonggi-do (GGI) | 37.8(4.6) / 38.1(6.0) [-0.3] | 20.7(3.0) / 17.4(3.7) [3.3] |
| | Chungcheongbuk-do (CCB) | 42.6(4.1) / 39.6 (4.6) [3.0] | 24.9(3.6) / 20.5(3.2) [4.4] |
| | Chungcheongnam-do (CCN) | 41.4(4.7) / 37.0(7.1) [4.4] | 29.3(3.8) / 22.3(4.6) [7.0] |
| | Jeollabuk-do (JLB) | 38.3(8.1) / 34.3(8.0) [4.0] | 26.6(5.7) / 22.8(5.4) [3.8] |
| | Jeollanam-do (JLN) | 43.1(4.2) / 34.9(4.0) [8.2] | 33.4(3.4) / 23.9(2.7) [9.5] |
| | Jeju Island (JEJ) | 49.3(6.0) / 34.2(4.3) [15.1] | 44.0(5.7) / 28.6(3.9) [15.4] |
| | Gyeongsangnam-do (GSN) | 44.7(4.9) / 39.8 (5.2) [4.9] | 28.9(3.1) / 21.6(2.6) [7.3] |
| | Gyeongsangbuk-do (GSB) | 45.5(5.3) / 37.5(5.4) [8.0] | 28.5(4.2) / 20.1(3.3) [8.4] |
| | Gangwon-do (GWO) | 44.5(4.0) / 39.1(4.7) [5.4] | 27.9(2.9) / 20.5(3.0) [7.4] |
| Background | Ulleung Island (ULL) | 56.6(7.2) / 43.9(6.1) [12.7] | 55.9(6.5) / 43.1(6.0) [12.8] |
| | Gosung (GSU) | 58.3(8.4) / 42.9(4.9) [15.4] | 58.1(8.8) / 44.9(5.3) [13.2] |

Table 3. The observed trends of $NO_2$ concentrations in spring and summer from linear fits of the data covering 2001-2021. The data were acquired from the surface monitoring network (www.airkorea.or.kr). Unit of slope and limit (2 sigma = 2 standard deviation) is ppb $yr^{-1}$. SNR denotes signal-to-noise ratio defined as the ratio of absolute value of slope to standard deviation. For the use of P-value and SNR, refer to Chang et al. (2021). Due to data limitations, the trend for the background sites could not be calculated.

| Stations | | $NO_2$ Spring (Summer) | | | |
|---|---|---|---|---|---|
| | | Slope (ppb $yr^{-1}$) | 2 Sigma (ppb $yr^{-1}$) | P-value | SNR |
| City | Seoul (SUL) | -0.77 (-0.72) | 0.22 (0.15) | < 0.01 (< 0.01) | 6.94 (9.57) |
| | Incheon (INC) | -0.37 (-0.50) | 0.22 (0.17) | < 0.01 (< 0.01) | 3.36 (5.88) |
| | Daejeon (DJN) | -0.10 (-0.12) | 0.14 (0.09) | 0.21 (0.02) | 1.43 (2.53) |
| | Gwangju (GWJ) | -0.51 (-0.35) | 0.15 (0.09) | < 0.01 (< 0.01) | 6.94 (7.74) |
| | Busan (BSN) | -0.64 (-0.49) | 0.16 (0.11) | < 0.01 (< 0.01) | 8.12 (8.93) |
| | Ulsan (ULS) | -0.04 (-0.06) | 0.23 (0.19) | 0.73 (0.51) | 0.34 (0.63) |
| | Daegu (DGU) | -0.65 (-0.51) | 0.18 (0.13) | < 0.01 (< 0.01) | 7.21 (8.15) |
| Province | Gyeonggi (GGI) | -0.41(-0.44) | 0.22 (0.16) | < 0.01 (< 0.01) | 3.80 (5.58) |
| | Chungcheongbuk (CCB) | -0.18(-0.16) | 0.20 (0.15) | 0.09 (0.05) | 1.82 (2.15) |
| | Chungcheongnam (CCN) | -0.10(-0.12) | 0.15 (0.12) | 0.21 (0.08) | 1.38 (1.97) |
| | Jeollabuk (JLB) | -0.17(-0.25) | 0.18 (0.14) | 0.08 (< 0.01) | 1.90 (3.61) |
| | Jeollanam (JLN) | -0.21(-0.21) | 0.16 (0.14) | 0.02 (< 0.01) | 2.56 (2.95) |
| | Jeju Island (JEJ) | -0.18(-0.16) | 0.20 (0.15) | 0.10 (0.04) | 1.76 (2.20) |
| | Gyeongsangnam (GSN) | -0.12(-0.10) | 0.17 (0.11) | 0.18 (0.08) | 1.42 (1.88) |
| | Gyeongsangbuk (GSB) | -0.76(-0.49) | 0.18 (0.13) | < 0.01 (< 0.01) | 8.47 (7.74) |
| | Gangwon (GWO) | -0.16(-0.20) | 0.14 (0.10) | 0.03 (< 0.01) | 2.37 (4.18) |

Table 4. The observed trends of CO concentrations in spring and summer from linear fits of the data covering 2001-2021. The data were acquired from the surface monitoring network ([www.airkorea.or.kr](www.airkorea.or.kr)). Unit of slope and limit (2 sigma = 2 standard deviation) is ppb yr$^{-1}$. SNR denotes signal-to-noise ratio defined as the ratio of absolute value of slope to standard deviation. For the use of P-value and SNR, refer to Chang et al. (2021). Due to data limitations, the trend for the background sites could not be calculated.

| Stations | | CO    Spring (Summer) | | | |
|---|---|---|---|---|---|
| | | Slope (ppb yr$^{-1}$) | 2 Sigma (ppb yr$^{-1}$) | P-value | SNR |
| City | Seoul (SUL) | -7.56 ( -5.34) | 2.94 (1.66) | < 0.01 (< 0.01) | 5.15 (6.44) |
| | Incheon (INC) | -7.65 ( -4.64) | 3.62 (2.46) | < 0.01 (< 0.01) | 4.23 (3.77) |
| | Daejeon (DJN) | -15.53 ( -9.71) | 5.68 (5.56) | < 0.01 (< 0.01) | 5.47 (3.49) |
| | Gwangju (GWJ) | -10.64 ( -8.00) | 3.60 (3.94) | < 0.01 (< 0.01) | 5.91 (4.06) |
| | Busan (BSN) | -12.32 (-11.05) | 3.90 (3.80) | < 0.01 (< 0.01) | 6.32 (5.82) |
| | Ulsan (ULS) | -4.80 ( 0.75) | 5.54 (5.28) | 0.10 (0.78) | 1.73 (0.28) |
| | Daegu (DGU) | -23.49 (-19.87) | 5.50 (5.30) | < 0.01 (< 0.01) | 8.54 (7.50) |
| Province | Gyeonggi (GGI) | -14.50 ( -8.82) | 2.18 (1.54) | < 0.01 (< 0.01) | 13.30 (11.42) |
| | Chungcheongbuk (CCB) | -17.68 ( -6.49) | 6.70 (3.92) | < 0.01 (< 0.01) | 5.28 (3.31) |
| | Chungcheongnam (CCN) | -20.95 ( -9.33) | 8.32 (4.62) | < 0.01 (< 0.01) | 5.04 (4.04) |
| | Jeollabuk (JLB) | -21.33 (-15.07) | 5.88 (4.34) | < 0.01 (< 0.01) | 7.26 (6.95) |
| | Jeollanam (JLN) | -5.86 ( -5.32) | 4.40 (4.60) | 0.02 (0.03) | 2.66 (2.31) |
| | Jeju Island (JEJ) | -10.74 ( -6.95) | 5.00 (5.64) | < 0.01 (0.02) | 4.30 (2.46) |
| | Gyeongsangnam (GSN) | -6.76 ( -3.92) | 4.44 (3.58) | < 0.01 (0.04) | 3.04 (2.19) |
| | Gyeongsangbuk (GSB) | -27.54 (-17.48) | 9.00 (6.64) | < 0.01 (< 0.01) | 6.12 (5.27) |
| | Gangwon (GWO) | -15.31 ( -9.03) | 4.34 (4.16) | < 0.01 (< 0.01) | 7.05 (4.34) |

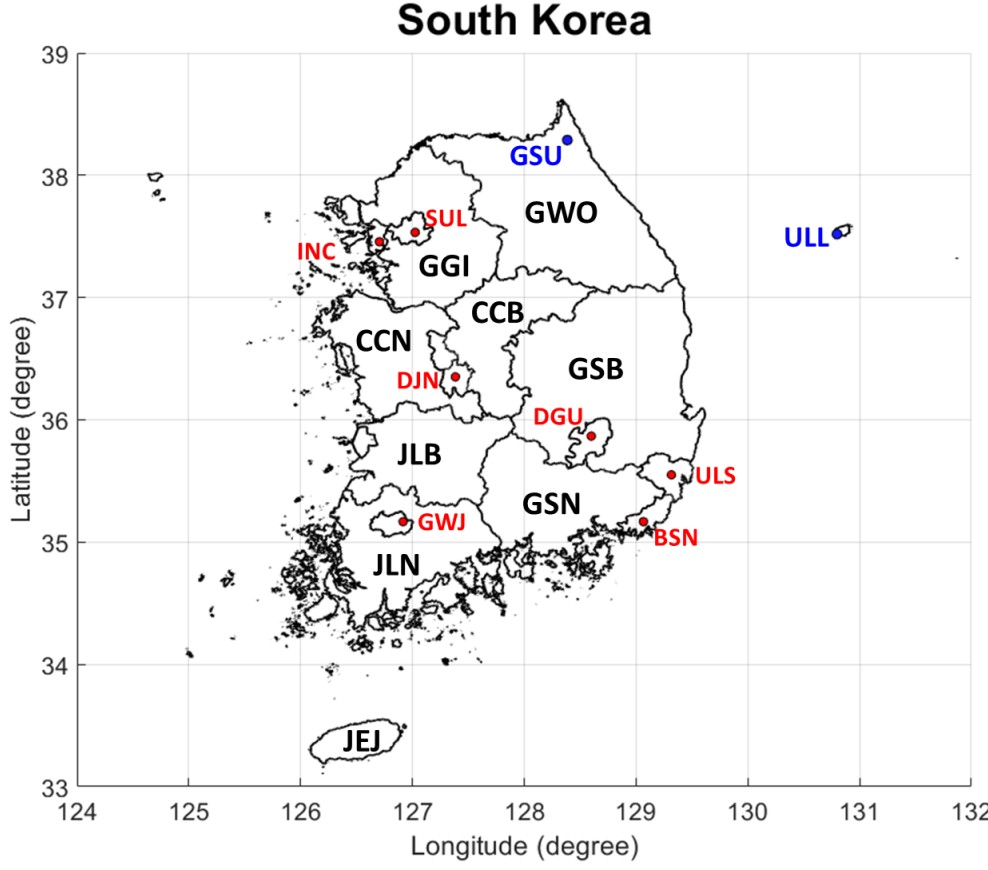

Figure 1. The locations of cities, provinces, and background sites in South Korea. The red,

black, and blue color denote city, province, and background site, respectively: Cities – SUL

(Seoul), INC (Incheon), DJN (Daejeon), GWJ (Gwangju), BSN (Busan), ULS (Ulsan),

DGU (Daegu); Provinces - GGI (Gyeonggi-do), CCB (Chungcheongbuk-do), CCN

(Chungcheongnam-do), JLB (Jeollabuk-do), JLN (Jeollanam-do), JEJ (Jeju Island), GSN

(Gyeongsangnam-do), GSB (Gyeongsangbuk-do), GWO (Gangwon-do);    Background

sites - ULL (Ulleung Island), and GSU (Gosung, Gangwon-do).

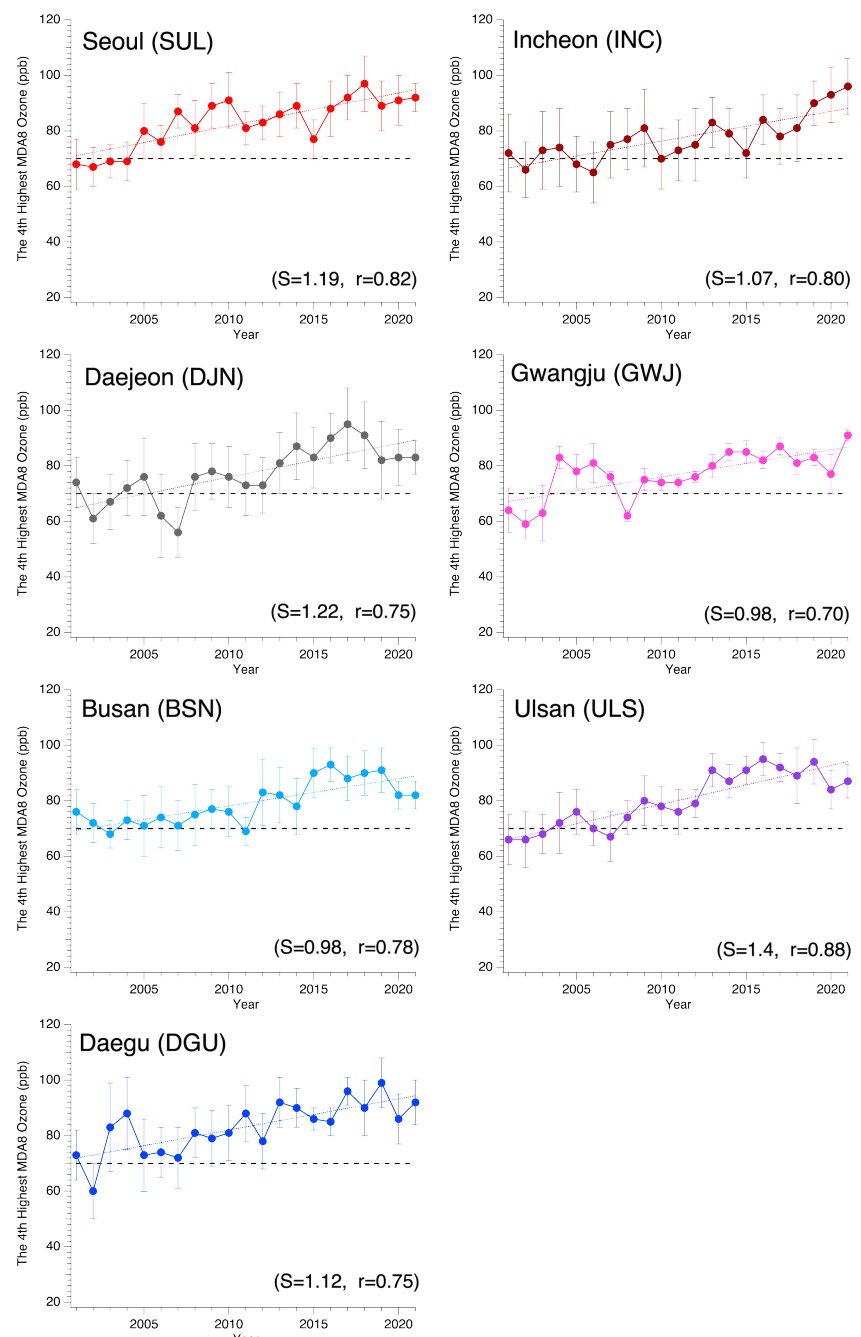

Figure 2. The trend of the 4th highest daily maximum 8 hours average (MDA8) $O_3$

concentrations in the South Korean metropolitan cities from 2001 to 2021. Only the data

for May-September (ozone season) are used. Bars denote standard deviations among the

sites within the city. The slopes (S) and correlation coefficien (r) from linear fits are shown

in parentheses. Grey dashed line indicates 70 ppb that is the air quality standard defined by

the US Environmental Protection Agency.

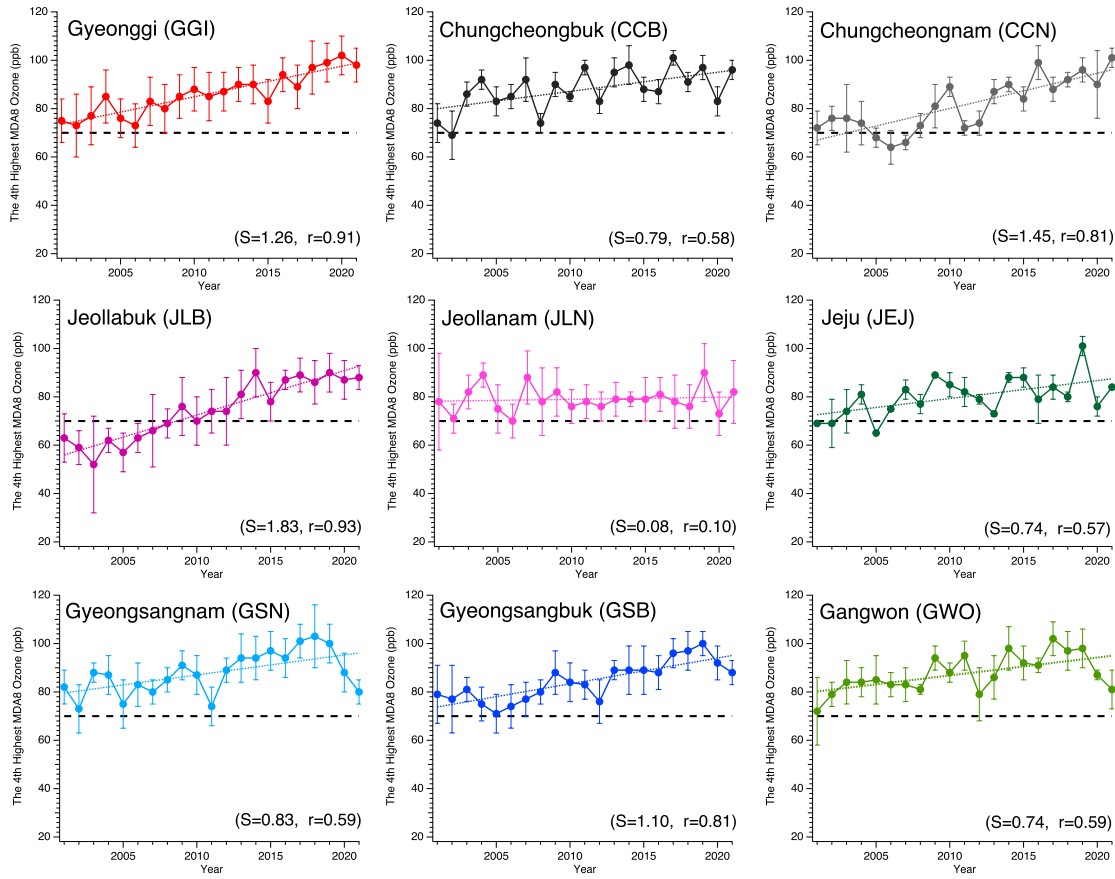

Figure 3. The same as in Figure 2 except for South Korean provinces.

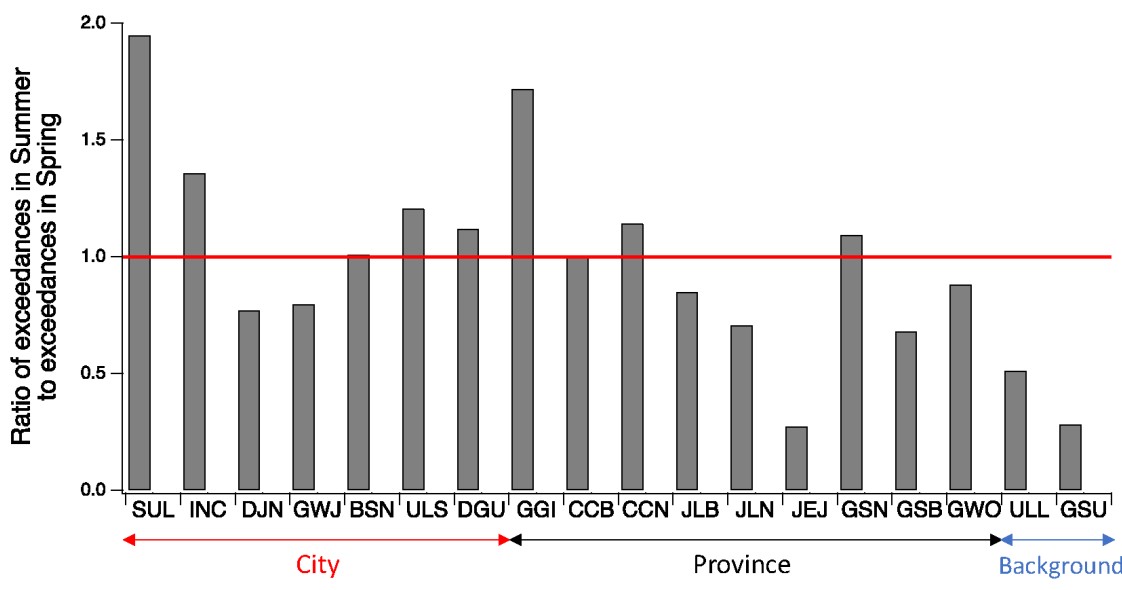

Figure 4. Ratio of O₃ exceedances in summer to exceedances in spring. The exceedances
are defined as the fraction of the data with hourly ozone concentration greater than 70 ppb
among all available data. The red line indicates an one to one line. X-axis denotes names
of cities, provinces, and background sites. Cities – SUL (Seoul), INC (Incheon), DJN
(Daejeon), GWJ (Gwangju), BSN (Busan), ULS (Ulsan), DGU (Daegu); Provinces - GGI
(Gyeonggi-do), CCB (Chungcheongbuk-do), CCN (Chungcheongnam-do), JLB
(Jeollabuk-do), JLN (Jeollanam-do), JEJ (Jeju Island), GSN (Gyeongsangnam-do), GSB
(Gyeongsangbuk-do), GWO (Gangwon-do); Background sites - ULL (Ulleung Island),
and GSU (Gosung, Gangwon-do). The data for 2002-2019 are utilized.

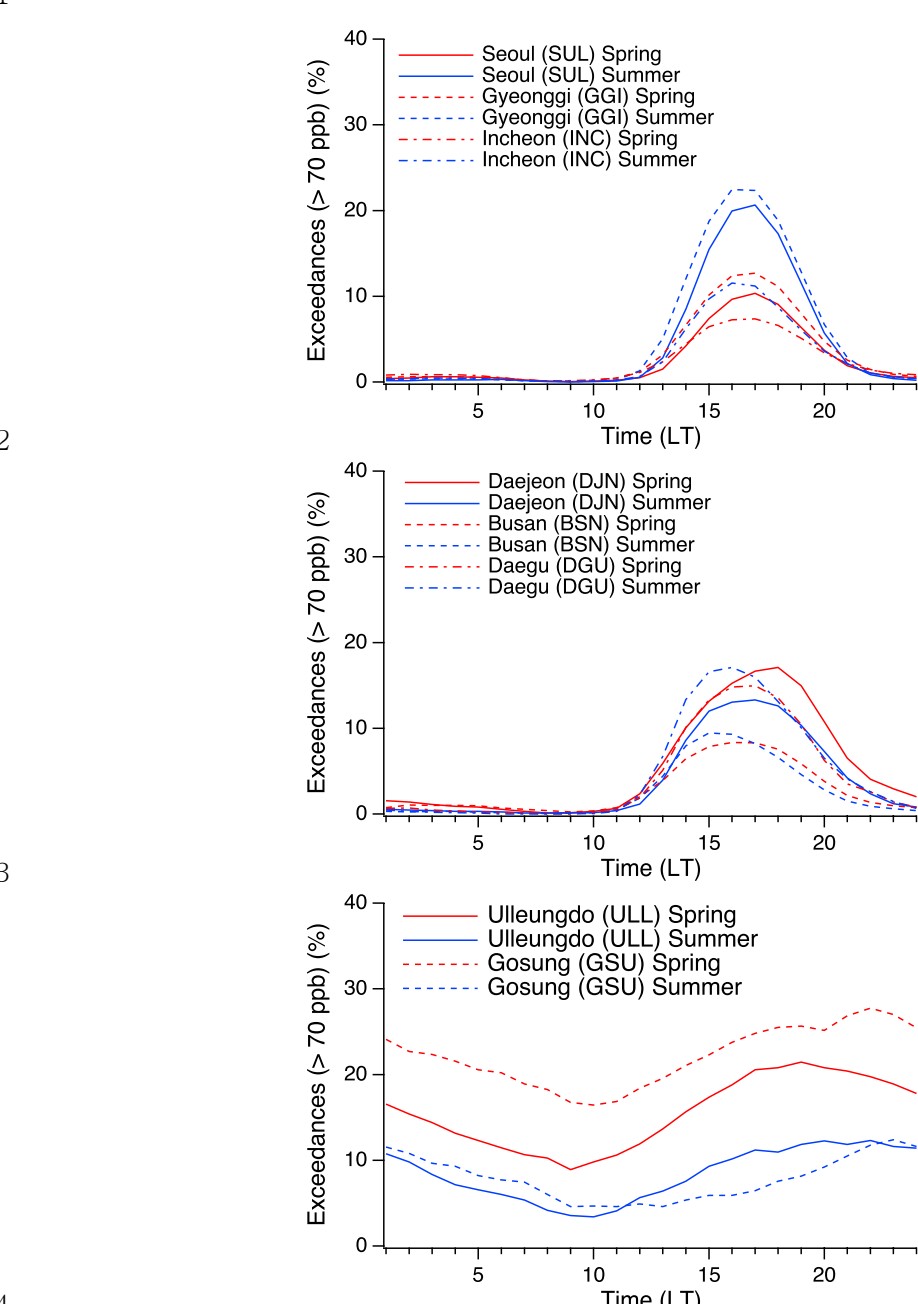

5 Figure 5. Diurnal O$_3$ exceedances over selected sites. (Top) cities and province in Seoul

6 Metropolitan Area, (middle) other cities, (bottom) background sites. The data for 2002-

7 2019 are utilized.

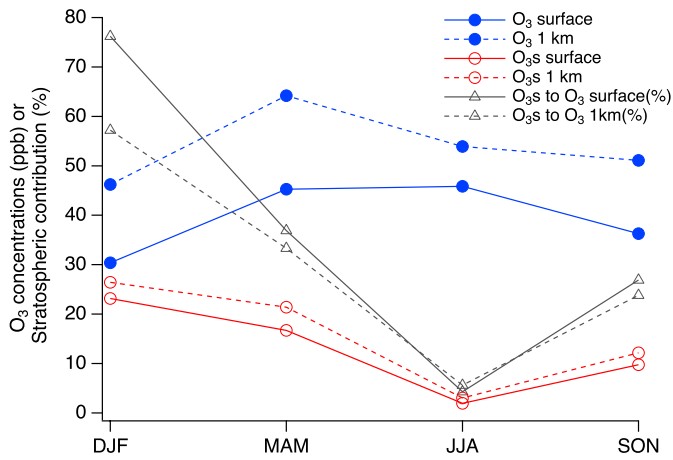

2  Figure 6. The contribution of stratospheric $O_3$ ($O_{3s}$) to the $O_3$ concentrations in each season

3  at surface and 1 km above ground level in South Korea. The plotted values are extracted

4  from the CESMv2.2 results for the entire country.

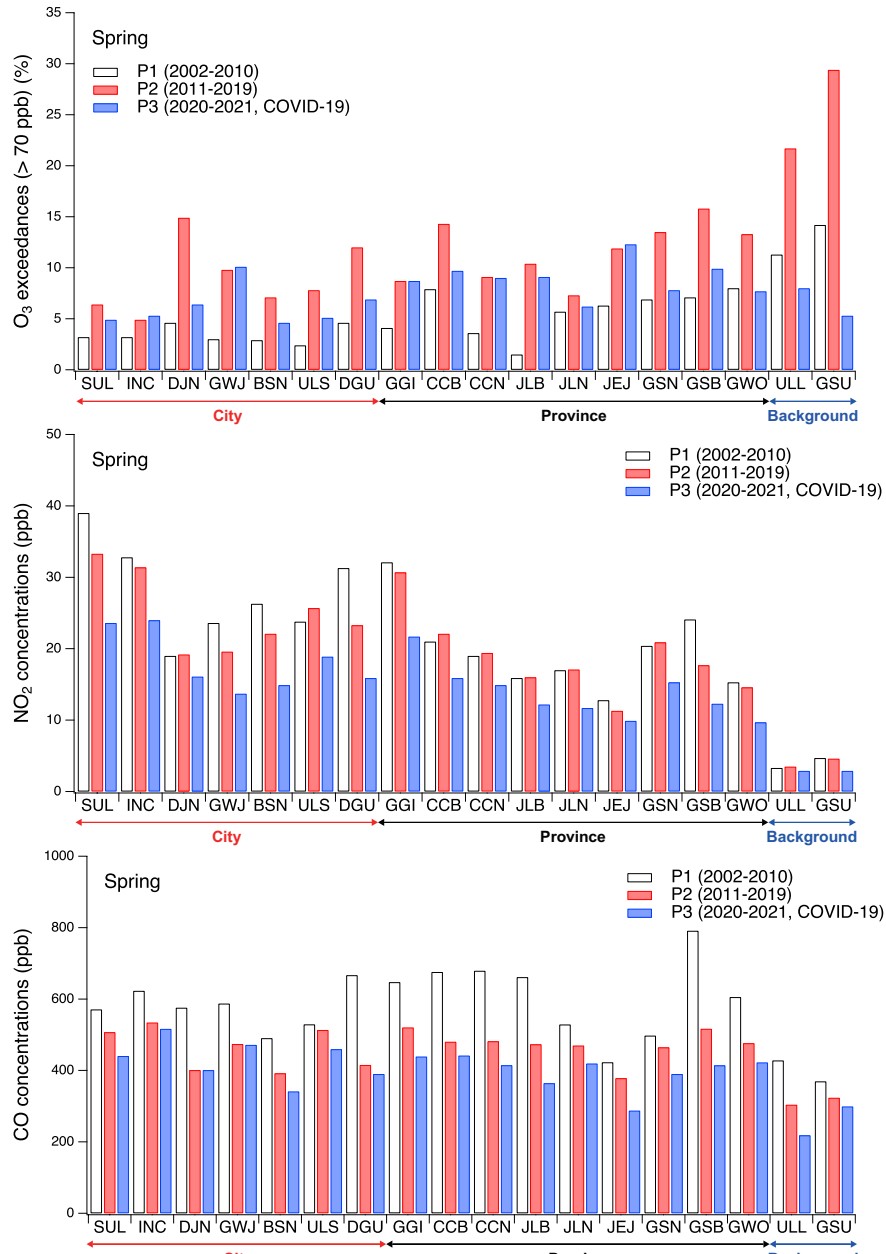

Figure 7. (Top) $O_3$ exceedances (%), (middle) $NO_2$, and (bottom) CO concentrations in South Korean cities, provinces, and background sites during spring for 2002-2010, 2011-2019, and 2020-2021 (COVID-19). X-axis denotes names of cities, provinces, and background sites. Cities - SUL (Seoul), INC (Incheon), DJN (Daejeon), GWJ (Gwangju), BSN (Busan), ULS (Ulsan), DGU (Daegu); Provinces - GGI (Gyeonggi-do), CCB (Chungcheongbuk-do), CCN (Chungcheongnam-do), JLB (Jeollabuk-do), JLN (Jeollanam-do), JEJ (Jeju Island), GSN (Gyeongsangnam-do), GSB (Gyeongsangbuk-do), GWO (Gangwon-do); Background sites - ULL (Ulleung Island), and GSU (Gosung, Gangwon-do).

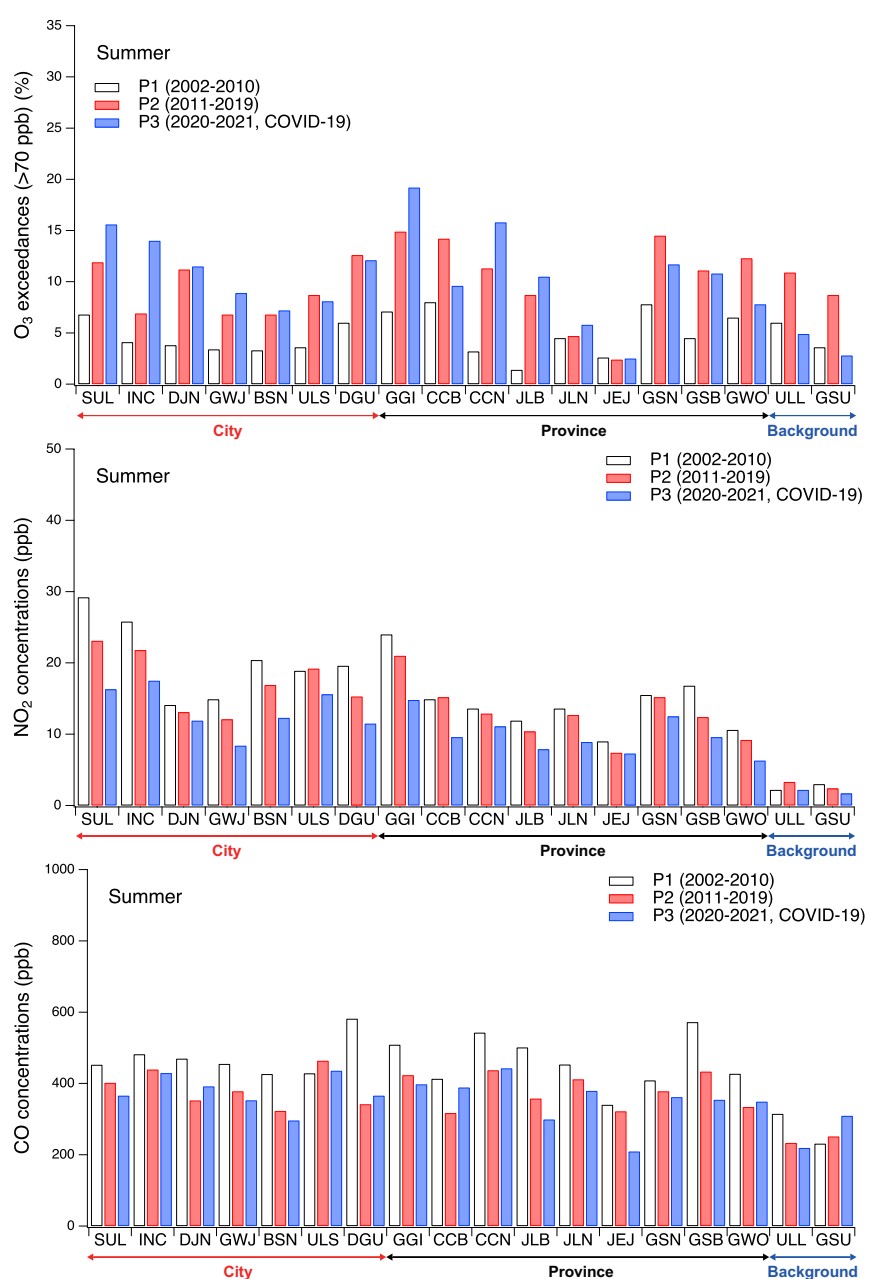

Figure 8. The same as Figure 7 except for summer.

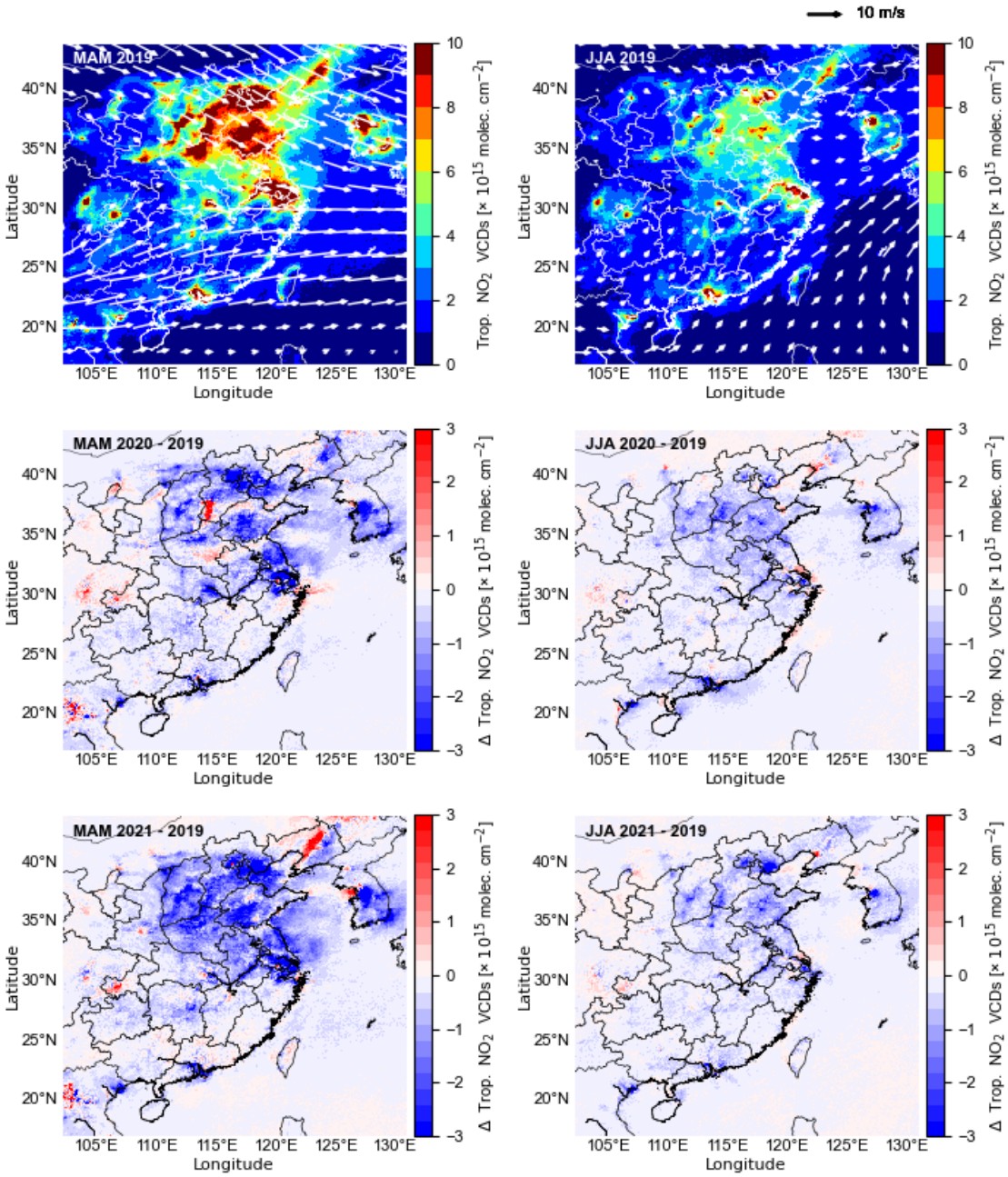

Figure 9. Differences in TROPOMI tropospheric $NO_2$ columns between 2019 and 2020 or between 2019 and 2021 (Difference = $NO_{2\ 2020\ or\ 2021}$ - $NO_{2\ 2019}$). Unit: molecules cm$^{-2}$. Wind vectors at 700 hPa from ERA-5 are shown for MAM and JJA, respectively. Here, ERA-5 denotes the European Centre for Medium-range Weather Forecasts Reanalysis 5[th] generation (DOI: 10.24381/cds.6860a573).

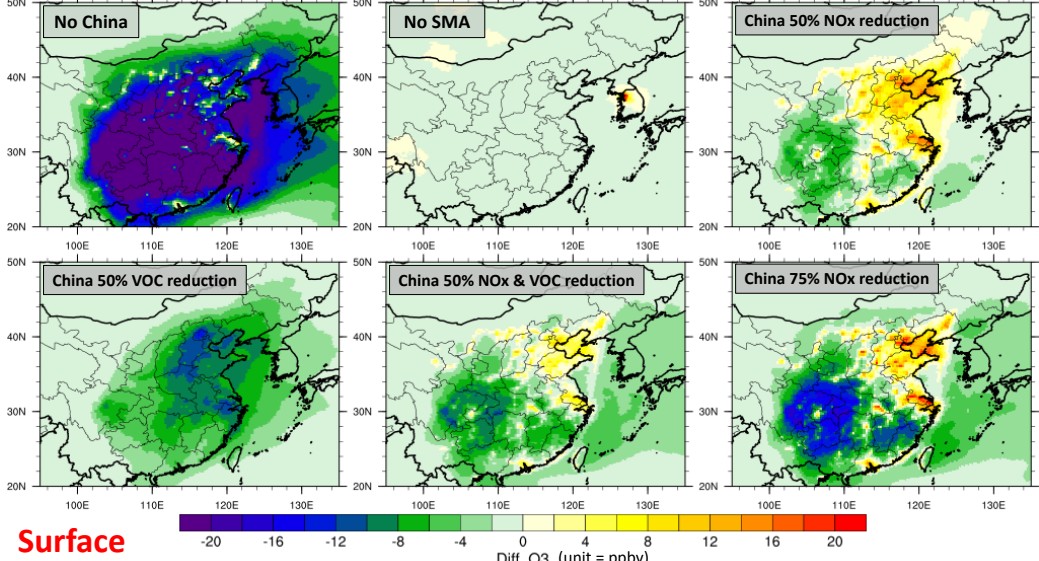

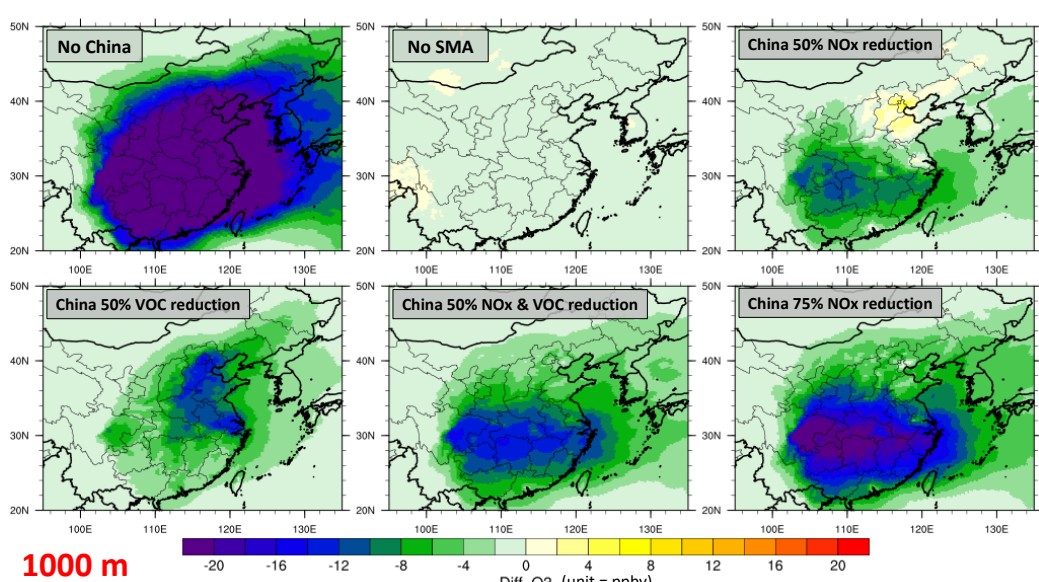

Figure 10. Differences in the WRF-Chem simulated ozone concentrations ($\Delta O_3$ = $O_3$_emission reduction case-$O_3$_control case) at (top) surface and (bottom) 1000 m above ground level. Green to blue colors (yellow to red colors) denotes reduced (increased) ozone concentration due to the emission changes. All model simulation results are utilized.

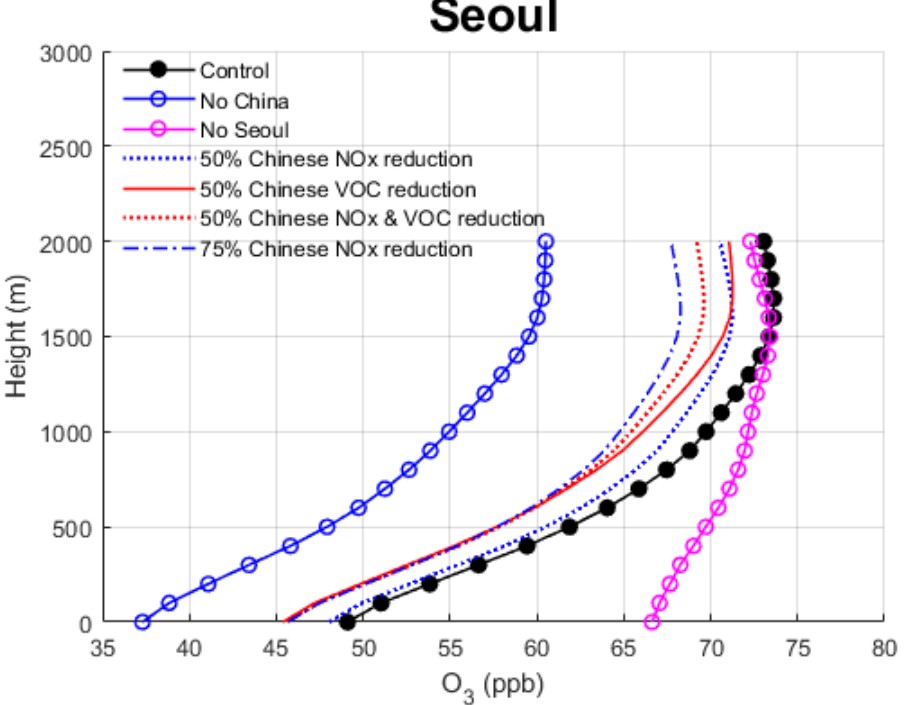

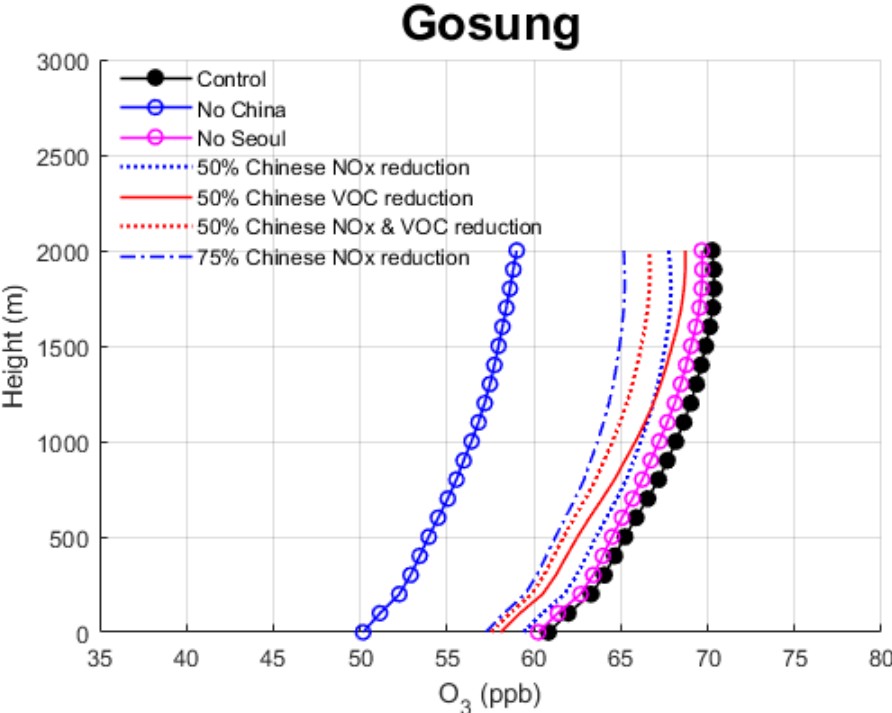

2 Figure 11. Vertical profiles of ozone from the WRF-Chem model simulations based on
3 various emission scenarios: (top) Seoul, and (bottom) Gosung, Gangwon-do. The model
4 results from 10 LT to 20 LT are averaged.
