# Peer review of "Changes in surface ozone in South Korea on diurnal to decadal time scale for the period of 2001-2021 Si-Wan Kim1\*, Kyoung-Min Kim2, Yujoo Jeong1,2, Seunghwan Seo2, Yeonsu Park2, and Jeongyeon Kim2 1Irreversible Climate Change Research Center, Yonsei University, Seoul, Sout"

_Atmospheric Chemistry and Physics, 2022_

## Referee Comment (RC2)

Report on "Changes in surface ozone in South Korea on diurnal to decadal time scale
for the period of 2001-2021" by Kim et al.

General Comments
The manuscript entitled "Changes in surface ozone in South Korea on diurnal to decadal time
scale for the period of 2001-2021" by Kim et al. gives an important quantification of surface
ozone's variabilities in South Korea. Between 2001-2021, ozone increased over most of the
monitoring sites by 1-2 ppbv/yr as well as the exceedances above the United States National
Ambiant Air Quality Standard for ozone, 70 ppbv. The COVID-lockdown period is characterized
by a decrease of ozone in spring, especially at the background sites, but not in summer.

The manuscript is well organized and written. The figures are clear and easy to read. It is very
much appreciated.
I would recommend the manuscript for publication after the authors address the following
minor comments.

Specific Comments
L2 P2: I believe there is a typo: Change "Increasing trends of tropospheric ozone in South Korea
in the last decades have reported in several studies" to "Increasing trends of tropospheric
ozone in South Korea in the last decades have been reported in several studies".

L4 P5: Could you give some details on the impact of the COVID-19 pandemic on the
atmospheric composition in spring versus in summer? Has South Korea experienced several
lock-downs in spring and summer 2020, or only in spring, with reduction of human
activities/emissions of the precursors of ozone?

L12 P5: I believe there is a typo: Change "as following" to "as follow".

L7 P6: Could you be more specific? Could you give the starting year? Are all the 500 stations still
working now? Maybe add a column "time period" in Table S1.

L9 P7: Could you be more specific on the stricter recommendations: quality assurance and
cloud fraction?

L10 P7: Have you conducted or are you aware of any sensitivity test to see how much the
compromise sampling statistics/quality may change the results?

L4 P9: Typo: Change "11st" to "11$^{th}$" (eleventh). Could you add the year?

L11 P10: Could you add the uncertainties on the trend estimate?

L14 P10: "Insignificant" is not used anymore (Wasserstein et al., 2019). Trend reliability can be
expressed with p-value (Wasserstein et al., 2019) and/or signal-to-noise (SNR) ratio (Chang et

al., 2021). Then you can apply the trend reliability scale (see table below from the *guidance note on best statistical practices for tropospheric ozone assessment report -TOAR-analyses* by Kai-Lan Chang, Martin Schultz, Gerbrand Koren and co-authors pending their approval, February 2023; the document will be posted on the TOAR website by end of April 2023 upon the TOAR steering committee approval, https://igacproject.org/activities/TOAR/TOAR-II) to report the trend and its uncertainty.

Table 3. Trend reliability scale

| *p*-value | SNR (signal-to-noise) value | Term |
|---|---|---|
| $p \leq 0.01$ | SNR ≥ 3 | very high certainty |
| $0.05 \geq p > 0.01$ | 2 ≤ SNR < 3 | high certainty |
| $0.10 \geq p > 0.05$ | 1.65 ≤ SNR < 2 | medium certainty |
| $0.33^1 \geq p > 0.10$ | 1 ≤ SNR < 1.65 | low certainty |
| $p > 0.33^1$ | SNR < 1 | very low certainty or no evidence |

[1]This boundary is meant to be fuzzy around 1/3 (Mastrandrea et al., 2010).

Table taken from the *guidance note on best statistical practices for tropospheric ozone assessment report -TOAR-analyses* by Kai-Lan Chang, Martin Schultz, Gerbrand Koren and co-authors pending their approval, February 2023; the document will be posted on the TOAR website upon the TOAR steering committee approval, https://igacproject.org/activities/TOAR/TOAR-II.

L5 P11: Spell out LT = Local Time, at least the first time it is used.

L12 P11: It would be worth adding a discussion with references on summer/spring differences: meteorology condition in Seoul and Gyeonggi-do compared with other sites/regions. That would probably fit in the "Discussions" section.

L15 P11: I found 7 sites showing more exceedances in summer than in springs according to Figure 4. Why do you report only 3 of them? I also found 10 sites showing more exceedances in spring than in summer, why do you report only 3 of them?

L7 P12: "than Incheon" is not clear. I believe there is a typo in the sentence. Could you rephrase?

L13-14 P12: Is it a statement from previous studies or from this current study? Could you give a reference or cite a figure to support this statement?

L20 P14: Does "large reduction of ozone" refer to the difference between the time periods P2 and P3? It would be helpful to clarify.

L14 P15: Does "likely to be VOC-limited" mean that VOCs did not decrease between P2 and P3 in South Korea? Any reference?

L20 P15: Do we know why there are more NO2 in MAM 2019 than JJA 2019? Specific human activities, Meteorological conditions? It would be interesting to see the maps of MAM 2020 and JJA 2020.

L20 P16: Why did you choose Seoul and Gangwon-do over other sites?

L1 P17: An evaluation of WRF-Chem above Seoul and Gangwon-do would be helpful. How does the control run compare with the observations? Any sondes launched during KORUS-AQ that can be used for this evaluation? Was this model study done with annual means or did you perform it for a specific season? Showing summer and spring would be useful to echo the seasonal results on trends estimate.

L7 P17: It seems to be very small changes (almost none). Could you be more quantitative?

L3-5 P18: You probably should inform on the altitude of both Gosung and Gangwon-do sites because it is a little confusing as it is written.

L1-2 P35: Are NO2 and CO values from CAM-Chem? It is worth clarifying in the caption.

L2 P36: Can you have colors or signs to differentiate cities, provinces and background sites, as well as the definitions of these three categories. Is it according to ozone diurnal/seasonal variability? Could you add a legend?

L2 P37: Could you add the uncertainties (2-sigma values), or p-value or signal-to-noise ratio associated with the slope values S? (see my previous comment on how to report trend and its uncertainty)

L4 P41: Is the extraction over the entire country? It should be specified in the caption and section 2.4.

L4 P44: Typo in the legend of Figure 11: change "Contorl" to "Control"

References:
Chang, K.-L., Schultz, M. G., Lan, X. et al. (2021). Trend detection of atmospheric time series: Incorporating appropriate uncertainty estimates and handling extreme events. *Elementa: Science of the Anthropocene*, 9.

Wasserstein, R.L., Schirm, A. L. & Lazar, N. A. (2019). Moving to a world beyond "$p < 0.05$". *The American Statistician.*

---

## Referee Comment (RC3)

Review of "Changes in surface ozone in South Korea on diurnal to decadal time scale for the period of 2001-2021" by Si-Wan Kim et al.

Manuscript ID: https://doi.org/10.5194/acp-2022-788

**Summary:**
This a very useful and informative paper. It has two major strengths: 1) Investigation of ozone in a region with strong local anthropogenic emissions, that also receives marine inflow with a highly polluted continent lying directly upwind of the marine area. 2) An effective incorporation of both observation and modelling based analysis. However, I believe that a major revision of the paper is required before it is ready for publication. One major need is for the authors to begin their observation-based analysis with a consideration of the ozone distribution that would be present in South Korea if there were no continental influences, i.e., if observed concentrations were due to transported baseline ozone alone. That consideration can rely on both the CESMv2.2 model calculations of these ozone concentrations (evidently shown in Figure 6), where results at ~1 km likely represent baseline ozone, and on analysis of observations as suggested below in the first major issue. This consideration would then provide a basis for understanding the continental influences, both from local South Korean emissions and from the Asian mainland emissions. Also much of the discussion is difficult to follow and requires substantial improvement; suggestions in this regard are given in the major and minor issues described below.

**Major issues:**

1) I believe that the discussion based on Table 1 requires reconsideration. I assume that these are mean ozone concentrations for the peak and base time periods in spring and summer. First, I think the period names are misleading. The 10-20 LT period has higher ozone concentrations than does the 01-06 period. However, those higher (10-20 LT) ozone concentrations are similar to that expected for northern mid-latitude baseline ozone concentrations. For example, Figure 5 of Parrish et al., (2020) shows that annual mean ozone is 30 to 40 ppb in the lower 1 km of the troposphere. Figure S14 of that paper shows that ozone at Mt. Walinguan (upwind of South Korea, but at higher elevation) has mean ozone of about 45 to 60 ppb in spring and summer. To my mind, the mean ozone in Table 1 in the 10-20 LT period predominately reflects baseline ozone transported into the country; this is the reason that these mean concentrations are similar throughout the country.

2) If the interpretation above is correct, then the lower ozone concentrations in the 01-06 period are caused by loss of ozone due to surface deposition and reaction with fresh NO emissions under a shallow nocturnal inversion. Such a diurnal cycle (low at night, higher during the day) is a ubiquitous feature of urban ozone.

3) To emphasize the similarity of the ozone concentrations throughout the country, and the predominant role of transported baseline ozone, I suggest that the background sites be included in Figure 3 and Table 1.

4) More generally, I suggest that all tables, figures and discussion clearly address the 7 cities, 9 provinces, and 3 background sites in a consistent manner to the fullest extent possible. The discussion is often difficult to follow when varying lists of cities, provinces and sites are mentioned.

5) The primary reason that mean ozone is generally higher in spring than in summer is that the lower troposphere baseline ozone is higher in spring than in summer, particularly in marine

influenced air; e.g., see Figures 4 and 6 of Parrish et al., (2020).

6) Pg. 11, lines 5-8: One reason the 01-06 LT ozone is higher in the spring is that the nocturnal inversion is tighter in the summer, so ozone loss at night is more pronounced in summer than in spring. Given the very local processes that determine the 01-06 LT ozone, for simplicity, the authors may wish to eliminate the discussion of this nighttime ozone.

7) A discussion of local CO and NOx trends begins near the bottom of pg. 13. These observation-based trends should be compared and discussed in relation to the trends of these species derived from the model emission inventories. This may be more relevant to NOx, since it does have more local influence than CO.

8) The discussion of the COVID-19 influence on ozone (pg. 14-15) is interesting, particularly the "large reduction of ozone in the background sites". There are other studies of the influence of COVID-19 emission reduction on background ozone at northern mid-latitudes. The findings in these other studies should be quantitatively compared to the present results.

9) I find the discussion beginning on line 14, page 13 and continuing to the end of the Results section on page 16 to be very confusing, with many topics discussed in a disjointed manner. Please revise and clarify this discussion. Any topic that cannot be clearly and concisely explained without speculation should be eliminated.

10) Similarly, the Section 4 discussion section is difficult to follow. The authors should aim to convey the main points of the modeling results as clearly and concisely as possible. The last two sentences of the section appear to be the main points; they should be clearly and concisely supported by the preceding discussion.

11) The Summary and Conclusions section will need to be rewritten when the issues identified here are addressed. Specifically:
   - The ozone in the 01-06 LT period is so affected by local conditions that it should not be included in the 2nd paragraph of this Section.
   - Page 19, discussion beginning on line 17 should be improved. If there is strong influence of long-range transport on the surface ozone at the background sites, then that influence must also be present at all sites throughout South Korea. That influence is not apparent at night at most sites due to rapid nighttime loss of ozone at most sites.
   - An explanation should be given as to why there is such large regional differences in overall percentage decline in $NO_2$. Perhaps this can be related to the model emission inventory?

**Minor issues:**

1) Pg. 4, Line 11: Four references are given for papers that have previously reported increasing ozone trends in South Korea. Two of those are missing from the reference list. The introduction should briefly summarize what these papers found, and discuss the advances that the authors' make in this paper beyond what is known from those earlier papers.

2) There are minor problems with the English usage, which should be corrected by editing by a native English speaker.

3) Page 5, line 9 mentions that 8 provinces are studied; however Table 1 lists 9 provinces. Please develop a list of cities, provinces, and background sites, and consistently use that list throughout the paper.

4) In the Figure 1 caption, the different colors used for the city province and site names should be described. Also it is not clear exactly what is being plotted here: Is each symbol the mean 4th highest (MDA8) over all sites in the city or province? Confidence limits should be given for all derived slopes.

5) In the description of the two models evidently different anthropogenic emission inventories are used in the two models (CMIP6 for 2000-2014 and SSP5-8-5 for 2015-2020 in CAM-Chem and WRF-Chem and EDGAR-HTAPv2). There should be a brief discussion regarding how well these inventories compare, and if any problems arise from using perhaps incompatible emissions in the two models. Also mention should be made regarding whether these inventories correctly simulate the emissions reductions during the COVID-19 period.

6) Page 10, line 11-12: For greater accuracy, I suggest changing "… increases by 1-2 ppb yr$^{-1}$ for most of cities and provinces across South Korea ..." to "… increases by 1.0-1.5 ppb yr$^{-1}$ for most cities and provinces across South Korea ..."

7) Page 10, line 12-13: For greater accuracy, I suggest changing "The most of cities and provinces have the 4$^{th}$ highest MDA8 O3 higher than 70 ppb after 2010." to "In nearly all cities and provinces, the 4$^{th}$ highest MDA8 O3 has been higher than 70 ppb since 2010 or earlier."

8) I suggest vertical lines be added to Figure 4 to separate the cities, provinces, and background sites from each other. Similarly for Figures 7 and 8. Also simplify the figure captions.

9) The discussion illustrated in Figures 4, 5 and 7 is based on "exceedances"; however, I cannot find where "exceedance" is defined in the paper. Please define. (I assume it is a day when MDA8 ozone exceeds 70 ppb).

10) Figure 5 needs to be clearly explained. If an exceedance is based on MDA8, how can there be a diurnal cycle, since there is only one MDA8 per day? Is this percent of days with ozone above 70 ppb in a given hour? I suggest using the same ordinate scale in all 3 graphs, so that the comparison is made easy for the reader. Also the general description of the sites included in the 3 graphs should be given; i.e., top = Seoul area, middle = secondary cities, bottom = remote sites.

11) It seems that the information included in Table 2 and Figure 6 are identical; I suggest that Table 2 be eliminated.

12) Please give units for the slopes in Table 3; confidence limits should be given for the derived slopes. Also please give the slopes for the background sites for comparison, if those data are available.

13) Figure 11 – x-axis labels have typo.

14) Page 20 – Please define SMA

**References:**

Parrish, D.D., et al. (2020), Zonal similarity of long-term changes and seasonal cycles of baseline ozone at northern mid-latitudes. *J. Geophys. Res.: Atmos.,* doi: 10.1029/2019JD031908.

---

## Author Comment (AC1)

Reply to the review of "Changes in surface ozone in South Korea on diurnal to decadal time scale for the period of 2001-2021"

We provide our replies below. The review is written in blue and our replies in black.

This manuscript addressed an issue of observed surface ozone increases in South Korea by analyzing a long-term dataset and 3-d air quality model simulations for divulging its attribution. The surface ozone increase in South Korea and China is a compelling issue for which previous literature extensively attempted to investigate its causes. Compared to them, I find it quite challenging that this work shows a new contribution to the scientific understanding of the issue or a new idea that needs to be investigated in the future. In addition, the manuscript should be reshaped to highlight its main findings by adding descriptions of how the authors reached conclusions, which were mostly based on immature analyses. I will elaborate on them below.

 Thank you for constructive criticism and introducing recent publications about surface ozone over South Korea that probed the sources of its abundance. We appreciate these studies and will include them for discussions in the revised manuscript as elaborated in the responses below.

Your comments are greatly appreciated. But we respectively disagree with the reviewer to the point that previous literature extensively attempted to investigate its causes and there are hardly any new contributions and ideas in our study. We hope that our responses below help better identify the values of this study and bring up many ideas to be studied/tested in the future. Past and recent publications (several publications) pointed out the possibility of long-range transport of ozone from China to South Korea and high background ozone value external to East Asia or South Korea for a certain period. However, the atmospheric/environmental science community is far from understanding the causes for the long-term trends of surface ozone over South Korea that were summarized in our study. Colombi et al. (2022) nicely demonstrated one possible cause for ozone increase over South Korea from 2015 to 2019. There are good agreements between our results and Colombi et al. (2022). And there are differences too. It is good that the two different approaches reach the similar conclusions, an importance of large background ozone in spring and existence of long-range transport from China to South Korea. Our study is different from Colombi et al. (2022) in terms of investigation of vertical sensitivity of ozone

to surface emission changes and the period of the data including the COVID-19 pandemic. We found a large reduction of ozone exceedances over most of the sites over South Korea in spring during the COVID-19 pandemic, which were not reported and were not extensively studied. We believe our study motivates more detailed modeling research encompassing the long-term period or the period including the COVID-19 pandemic for better understanding of ozone over South Korea and China.

In the responses below, we explain how we reached the conclusions and will include the discussed contents to the revised manuscript. We were preparing several manuscripts regarding the WRF-Chem and CAM-Chem performances and did not include details and evaluation results to the current manuscript. This is the reason why we omitted the model evaluations. The authors have full pictures, but the reviewer and reader would not have them. Therefore, it is helpful to provide more information about model performances as the reviewer asked. In the revised manuscript, we will include evaluations of the model ozone simulations to Supporting Information and refer to the manuscripts submitted or to be submitted.

- Papers submitted and in preparation

Jeong, YuJoo, et al., 2023, Influence of ENSO on tropospheric ozone variability in Asia, submitted. (evaluations of CAM-Chem ozone simulations)

Kim, Kyoung-Min, et al., 2023, Sensitivity of the WRF-Chem v4.4 ozone, formaldehyde, and their precursor simulations to multiple bottom-up emission inventories over East Asia during the KORUS-AQ 2016 field campaign, *in preparation*.

P2,L2 - "Increasing trends of tropospheric ozone in South Korea" is a bit misleading because ozone in surface air does not always reflect tropospheric ozone. Needs to be revised to surface ozone.
→ Gaudel et al. (2020) found that tropospheric ozone in China and South Korea increased from 1996 to 2016. Both surface and tropospheric ozone in South Korea increased during the last decades. However, for the abstract of this manuscript, we changed "Increasing trends of tropospheric ozone" to "Increasing trends of surface ozone" as the reviewer suggested.

P4,L11 - Here and elsewhere, references at not in the reference section. Please check all the citations and include other previous studies on the same issue (e.g., Colombi et al., ACPD, 2022, and the references are therein).

→ Thank you for introducing Colombi et al and references therein. We originally included the references that focused on the analysis of surface ozone measurements in South Korea. Now in the revised manuscript, we include more references including modeling or analysis studies (see the reference section in this reply).

P4,L11 - "Ozone in South Korea …" this sentence requires a citation.

→ We will cite the papers, Oh et al. (2010) and Lee and Park (2022) (see the reference section in this reply).

P8,L11 - Stratospheric ozone appears to have a significant effect on ozone in the troposphere and even in surface air in this study. However, I cannot find out how the effect of stratospheric ozone on tropospheric and surface ozone was quantified in the manuscript. I think that it should be elaborated on here.

→ CESM2.2 calculates $O_{3S}$ as a 3-D variable in space. Originally, $O_{3S}$ is $O_3$ above tropopause. The $O_{3S}$ is transported and undergoes chemical losses below tropopause as

$$O_{3S} = O_{3S} * \exp(-O_{3S}\_Loss).$$

The $O_{3S}\_Loss$ rate by chemical reactions in the troposphere is calculated:

$O_{3S}\_Loss = 2.0*O\_O_3 + O1D\_H_2O + HO_2\_O_3 + OH\_O_3 + H\_O_3 + 2.0*NO_2\_O + 2.0*jno_3\_b + 2.0*CLO\_O + 2.0*jcl_2o_2 + 2.0*CLO\_CLOa + 2.0*CLO\_CLOb + 2.0*BRO\_CLOb + 2.0*BRO\_CLOc + 2.0*BRO\_BRO +2.0*BRO\_O + CLO\_HO_2 + BRO\_HO_2 + S\_O_3 + SO\_O_3 + C_2H_4\_O_3 + C_3H_6\_O_3 + ISOP\_O_3 + MVK\_O_3 + MACR\_O_3 + MTERP\_O_3 + BCARY\_O_3.$

ISOP=isoprene

MVK= methyl vinyl ketone

MACR=methacrolein

MTERP= pinene_a + carene_3 + thujene_a + 2met_styrene + cymene_p + cymene_o + terpinolene + bornene + fenchene_a + ocimene_al + pinene_b + sabinene + camphene + limonene + phellandrene_a + terpinene_g + terpinene_a + phellandrene_b + myrcene + ocimene_t_b + ocimene_c_b

BCARY= caryophyllene_b + bergamotene_a + bisabolene_b + farnescene_b + humulene_a.

For details of chemical reactions and variables, please refer to Emmons et al. (2020). We will include explanations about $O_{3S}$ in the revised manuscript. The representation of $O_{3S}$ has uncertainties, but it can be used as a parameter that indicates the contribution of stratospheric ozone to tropospheric ozone at each altitude at least qualitatively. We will explain how $O_{3S}$ is calculated and mention uncertainty of using $O_{3S}$ in the revised manuscript.

Sections 2.4, 2.5. – This study used model simulations to understand the observed characteristics of surface ozone in South Korea. Therefore, an extensive model evaluation should be conducted and discussed somewhere in the manuscript by focusing on how good the model is to reproduce the observations and their variability.

→ We have extensively evaluated our model results with the airborne and surface observations acquired during the KORUS-AQ campaign and the routine surface monitors in China and South Korea. The results are summarized and will be submitted as a separate manuscript to a relevant journal:

Kim, Kyoung-Min, et al., 2023, Sensitivity of the WRF-Chem v4.4 ozone, formaldehyde, and their precursor simulations to multiple bottom-up emission inventories over East Asia during the KORUS-AQ 2016 field campaign, *in preparation*.

For example, the diurnal variations of the model and observed surface ozone concentrations in China and South Korea are compared below (Figure R1 and Table R1). We found decent model performances in the surface ozone concentrations with the bottom-up emission inventories EDGAR-HTAPv2(EDV2), EDGAR-HTAPv3(EDV3), and KORUS-AQv5(KOV5). EDV3 and KOV5 performed a little better.

[Figure]

**Figure R1.** Averaged O$_3$ from the ground-based observations and model results for regional boxes that distinguish urban (red box) and non-urban (green box) region (central plot). Box averaged diurnal cycle (solid lines) of O$_3$ and 1/4 of standard deviations (filled area) from observations (black), the WRF-Chem simulations using EDGAR-HTAP version 2 (EDV2, green), EDGAR-HTAP version 3 (EDV3, blue), and KORUS-AQ version 5 (KOV5, red) are shown. The diurnal cycle plots represent Northern China (NOC, 38-42˚N/106-110˚E), Sichuan-Chongqing-Guizhou (SCG, 27-33˚N/103-109˚E), Pearl River Delta (PRD, 21.5-24˚N/112-115.5˚E), Southeastern China (SEC, 24-28˚N/116-120˚E), Yangtze River Delta (YRD, 30-33˚N/119-122˚E), South Korea (KOR, 34.5-38˚N/126-130˚E), North China Plain (NCP, 34-41˚N/113-119˚E), and Northeastern China (NEC, 43-47˚N/124-130˚E).

**Table R1.** Comparison of the ground-based hourly O$_3$, NO$_2$, and CO observations with the simulations utilizing EDGAR-HTAP v2 (EDV2) and v3 (EDV3) and KORUS v5 (KOV5) in each regional box (unit = ppb).

| Region | | | [1]NCP | [1,a]SCG | [1]YRD | [1]PRD | [1,b]KOR (SMA) | [2,c]NEC | [2,d]NOC | [2,e]SEC |
|---|---|---|---|---|---|---|---|---|---|---|
| | **N** | | 190 | 104 | 93 | 68 | 358 (125) | 45 | 28 | 43 |
| | **OBS** | **Mean** | 44.5 | 34.6 | 38.2 | 27.9 | 41.5 (36.6) | 40.9 | 44.3 | 26.1 |
| | | **Mean** | 32.2 | 53.5 | 21.6 | 27.6 | 40.5 (31.1) | 28.6 | 39.4 | 40.8 |
| | **EDV2** | **Bias** | -12.3 | 18.9 | -16.6 | -0.3 | -1.0 (-5.5) | -12.3 | -4.9 | 14.7 |
| | | **R** | 0.65 | 0.53 | 0.62 | 0.61 | 0.59 (0.60) | 0.48 | 0.63 | 0.52 |
| | | **Mean** | 43.4 | 57.5 | 35.7 | 34.7 | 41.0 (32.6) | 35.2 | 43.7 | 45.5 |
| **O$_3$** | **EDV3** | **Bias** | -1.1 | 23.0 | -2.5 | 6.8 | -0.5 (-4.0) | -5.7 | -0.6 | 19.4 |
| | | **R** | 0.68 | 0.55 | 0.66 | 0.65 | 0.56 (0.57) | 0.63 | 0.67 | 0.55 |
| | | **Mean** | 49.0 | 55.3 | 41.1 | 35.7 | 42.2 (33.1) | 37.1 | 43.8 | 42.4 |
| | **KOV5** | **Bias** | 4.5 | 20.7 | 2.8 | 7.8 | 0.7 (-3.5) | -3.8 | -0.5 | 16.3 |
| | | **R** | 0.71 | 0.53 | 0.65 | 0.70 | 0.62 (0.64) | 0.62 | 0.67 | 0.54 |

1) Urban area, 2) Non-urban area
a) Sichuan-Chongqing-Guizhou, b) South Korea (SMA-Seoul Metropolitan Area), c) Northeastern China, d) Northern China, e) Southeastern China

Evaluation of the model results with the aircraft data acquired during the KORUS-AQ campaign are shown below (Figure R2 and Table R2).

[Figure]

**Figure R2.** Averaged model and airborne observations of (a) $O_3$, (b) $NO_2$, (c) CO, and (d) HCHO (bars) and 1/4 of standard deviations (whiskers) (unit: ppbv) under 2 km height for the Local, Transport, and Chungnam cases from DC-8 (grey), EDV2 (green), EDV3 (blue), and KOV5 (red). The Chungnam (Chungcheongnam-do) region has large point sources like coal-burning power plants and petrochemical facilities that are not well-represented in the bottom-up emission inventories. The local case (May/4, May/20, June/2, June/3) and transport case (May/25, May/26, June/1) represent the dates with the smallest and largest influence from Chinese emissions, respectively. The Chungnam case represents the dates when DC-8 had survey flights targeting the urban and point sources in Chungcheongnam-do and downwind.

**Table R2.** Comparison of aircraft-based 1-minuite-interval $O_3$, $NO_2$, CO, and HCHO observations with EDV2, EDV3, and KOV5 in each case distinguished by China contribution to $O_3$ concentration under 2 km height (unit = ppb).

| Species | Case | Type | N | Mean | Bias | σ | R |
|---|---|---|---|---|---|---|---|
| $O_3$ | **Local** (5/4,20 , 6/2,3) | **OBS** | 1125 | 81.2 | | 15.3 | |
| | | **EDV2** | | 65.2 | -15.9 | 13.4 | 0.66 |
| | | **EDV3** | | 65.2 | -16.0 | 12.8 | 0.59 |
| | | **KOV5** | | 62.6 | -18.5 | 11.5 | 0.70 |
| | **Transport** (5/25,26 , 6/1) | **OBS** | 605 | 95.6 | | 19.1 | |
| | | **EDV2** | | 87.3 | -8.3 | 13.8 | 0.64 |
| | | **EDV3** | | 93.1 | -2.5 | 16.0 | 0.67 |
| | | **KOV5** | | 84.8 | -10.8 | 14.3 | 0.69 |
| | **Chungnam** (5/22 , 6/5) | **OBS** | 812 | 98.4 | | 17.8 | |
| | | **EDV2** | | 61.6 | -36.8 | 14.3 | 0.14 |
| | | **EDV3** | | 60.2 | -38.2 | 14.2 | 0.07 |
| | | **KOV5** | | 60.3 | -38.1 | 14.0 | 0.17 |

In summary, the model reasonably simulated ozone concentrations (particularly for the Transport Case), but they are overall underestimated compared to the observations. Potential causes for the discrepancy are underestimated CO and volatile organic compound emissions/concentrations in China and South Korea and/or uncertainties in the background ozone external to East Asia. Details about the model performances of precursor emissions are discussed in the manuscript by Kim, Kyoung-Min et al. (2023) and are beyond the scope of this study. We included some of the model results for discussions for our manuscript and will add some evaluation results to Supporting Information.

P9,L4 – Years for the WRF-Chem simulations were missing. Did you conduct simulations for all years or for a particular year?
→ The WRF-Chem model was conducted for 2016. We will specify the model year in the revised manuscript.

P9,L7 – It appears that the authors used different meteorology to drive CAM-Chem simulations and WRF-Chem simulations. Have you ever thought about using identical meteorology for both models?
→ The WRF-Chem and CAM-Chem model results were shown for different purposes. The WRF-Chem runs were used to analyze the sensitivity of ozone over South Korea to the emissions over China and South Korea for a limited time window (May-June 2016). The CAM-Chem runs inform the seasonal changes in the background ozone including the contribution of stratospheric ozone to the troposphere for the long-term period. Thorough comparisons of the two model results are beyond the scope of this study. Meanwhile, both WRF-Chem and CAM-Chem accurately simulated meteorology (Table R3 and Figure R3).

**Table R3.** Comparison of surface meteorological observations and WRF-Chem for the KORUS-AQ campaign period. R (RMSE) denotes correlation coefficient (root-mean-square-error).

| Nation | | Eastern China (sites = 271) | | | South Korea (sites = 48) | | |
|---|---|---|---|---|---|---|---|
| Variable | | Temperature (°C) | Relative humidity (%) | Wind speed (m/s) | Temperature (°C) | Relative humidity (%) | Wind speed (m/s) |
| | N | 83698 | 83696 | 79595 | 14948 | 14946 | 14103 |
| Mean | Obervation | 20.13 | 65.02 | 2.87 | 18.94 | 65.81 | 2.56 |
| | WRF-Chem | 19.22 | 65.35 | 4.12 | 17.23 | 71.35 | 3.84 |
| R | | 0.90 | 0.85 | 0.55 | 0.88 | 0.76 | 0.62 |
| Mean bias | | -0.91 | 0.32 | 1.25 | -1.71 | 5.54 | 1.27 |
| RMSE | | 3.20 | 13.94 | 2.45 | 2.84 | 15.88 | 2.31 |

[Figure]

**Figure R3.** The examples of CAM-Chem U, V wind components for spring, 2003. Without nudging, the model simulated U, V do not closely agree with the MERRA2 data.

P9,L14 – The time information of emissions inventory used in the model is missing. Did you also consider biomass burning emissions in the model?
→ EDGAR-HTAPv2 represents the year 2010. Since there are 6 years difference in EDGAR-HTAPv2 from 2016, we also utilized EDGAR-HTAPv3 representing the year 2016 in the revised manuscript. Park et al (2021) informed that biomass burning was not an important factor affecting air quality in South Korea during KORUS-AQ. Therefore, we did not include the biomass burning.

P10,L9, - You analyzed the 4th highest MDA8 O3. I wonder how this metric well represents ozone air quality because these could be rather extreme events, which rarely happen. In other words, how frequently people in South Korea were exposed to this metric?
→ The published works on the trend of surface ozone in South Korea presented the ozone metrics such as annual mean of hourly ozone, annual mean of MDA8 ozone, annual mean of daily maximum hourly ozone, and frequency of hourly concentrations greater than 120 ppb. The trends based on those metrics have already been published (e.g., Yeo and Kim, 2021). Since the US EPA National Ambient Air Quality Standard (NAAQS) for ozone is 70 ppb, as the fourth-highest MDA8 ozone concentration, averaged across three consecutive years, and the recent study by Wang et al. (2022) adopted the 4th highest MDA8 ozone concentrations as one of the metrics for study of Chinese ozone pollution, it would be nice to have analyses adopting the 4th highest MDA8 ozone for a global comparison. The EPA standard is also designed for public health protection. Exceedances presented in our study are similar to the frequency exposed to MDA8 ozone > 70 ppb (relevant to EPA standard).

P10,L14 – The trend in Jeollanam-do differs from other provinces. This is explained by "MDA8O3 in Jeollanam-do is high before 2010". I do not understand why this is the case. Here and elsewhere, please check out the proper usage of provinces and city names.

→ The monitoring sites in Jeollanam-do include the Yeosu-Kwangyang region in which many petrochemical industry (e.g., GS Caltex, LG Chem), and iron steel complexes (e.g., POSCO) are located, similar to Houston, Texas. This region experienced severe ozone problems in the 1990's to early 2000's (Ghim, Y. S. 2000). We will mention large unique sources in this area in the revised manuscript. And we will double-check the consistency of names for provinces and cities.

P10,L15-17 – This sentence includes several factors, contributing to ozone increases in South Korea. Proper citations are required.

→ "Widely increasing long-term ozone trends in South Korea indicate a regional nature of this pollutant" is the statement we made from our analysis. However, we will include some references that support this statement with modeling (e.g., Lee and Park, 2022, Colombi et al., 2022).

P11,L2 – "Investigating seasonal differences in ozone in South Korea" has been examined by Lee and Park (2022). Any consistency or dissimilarity from the previous study is worth being mentioned.

→ Lee and Park (2022) reported the April mean ozone concentration of 39.3 ppb, which is slightly higher than the July counterpart (38.3 ppb) from their model simulations for the year 2016 and the selected surface monitor sites for 4 main regions (Seoul, Chungbuk, Gwangju, and Pusan). Our study summarizes the differences between spring (March, April, May) and summer (June, July, August) for 21 years including 192 monitoring sites covering the whole of South Korea focusing on the analysis of long-term surface ozone observations. On overage, the observed spring mean ozone is 34.3 ppb and the summer mean ozone is 29.0 ppb over South Korea in our study. Lee and Park (2022) indicated that ozone air quality in South Korea is determined mainly by year-round regional background contributions (peak in spring). With some differences in details, the results from the two studies are qualitatively similar arguing high springtime background ozone value. In the revised manuscript, we will add discussions above. One unique aspect of our modeling study is demonstrations of the impact of emission in Seoul on Gangwon-do, causing slight ozone decrease in Gangwon-do with zero-Seoul emissions from surface to 2 km in May

2016. Our study highlights the diverse impacts of surface emission changes (over China or Seoul) on downwind ozone at different altitudes (Figure 11 in our original manuscript). In the future, more detailed analysis of ozone in Gangwon-do will be helpful since the Gangwon-do region is highly elevated (Figure R4), potentially receiving upwind ozone at high altitude. In the original manuscript, "Gangwon-do" meant "Gosung, Gangwon-do" in Figure 11. In the revised manuscript, we will correct this title and typing error in the x-axis (Ozobe → Ozone). We will also include the map of South Korea to Supporting Information and explain potential paths of ozone transport from China to Seoul to Gosung with a simplified diagram.

[Figure]

Figure R4. (Top) topography map of South Korea and (bottom) West-East vertical cross-section of terrain connecting Seoul and Gosung. Seoul and Gosung in Gangwon-do are highlighted with color circles. Color bar denotes elevations above sea level (m). A simplified potential ozone transport path is depicted in the bottom plot. Here the ozone layer is colored orange and the terrain is colored gray.

P13,L1-13 – Stratospheric influences are quite large, which are still debatable. As I mentioned above, how did you obtain the stratospheric ozone influences on low tropospheric and surface ozone concentrations in South Korea? Does the model reproduce observations well? You have to elaborate a lot on this part.

→ $O_{3s}$ was explained in the response above. In this reply, we show how the CAM-Chem simulations are compared with the ozonesonde data acquired over Pohang, South Korea from 1996 to 2020 (Figure R5). The model results and observations reasonably agree in terms of seasonal variability and absolute values. The model run with ~1 degree horizontal resolution reduces positive model surface ozone biases in summer compared to ~2 degree horizontal resolution run, but increases biases in late autumn to early spring at 500-850 hPa. At the 200 hPa level (close to tropopause), the CAM-Chem with both resolutions agree well with the observations. We presented the results with ~1 degree resolution in the original manuscript. We will explain the performance of CAM-Chem against the ozonesonde data in the revised manuscript.

[Figure]

Figure R5. Monthly variations of CAM-Chem simulated ozone concentration (red line) and observed ozone concentration (black solid line) at each pressure level near Pohang, between 14-17 KST. (Left panel) ~1 degree horizontal resolution simulation, and (right panel) ~2 degree horizontal resolution simulation.

 – Colombi et al. (2022) already performed a nice analysis on the effect of precursor changes on observed surface ozone increases in South Korea. You have to compare your work with theirs.

→ We think it is very nice to have several publications about the topic of ozone in South Korea, which agree in general, but have a different emphasis and details. A main difference between Colombi et al. (2022) and our study is the period of the study and whether it focuses on the surface ozone or vertical sensitivity explaining ozone variability at different locations in South Korea. Our study investigated surface ozone and ozone at various altitudes to consider the transport within and above the boundary layer between China and South Korea. Colombi et al. (2022) analyzed the surface ozone and $NO_2$ concentrations mainly over the Seoul Metropolitan Area from 2015 to 2019. The increase of ozone was mostly attributed to decrease in $NO_2$ for the studied period in their study. It is nice to identify one possible cause for the increase of surface ozone in the SMA from 2015 to 2019 as in Colombi et al. (2022). We will mention this study in the revised manuscript. One question that remains is the existence of long-term increasing trends of surface ozone over South Korea (SMA and other regions) when $NO_x$ concentrations were steady before 2015-2019. What is the cause for this increase? Further research would be necessary to understand the long-term trends of ozone over South Korea.

Both Colombi et al. (2022) and Lee and Park (2022) indicated high background ozone concentration external to East Asia (or South Korea), suggesting difficulty of achieving ozone standards.  Our study agrees to this point. Probably one different message is that reducing emissions of NOx and VOC here and there all together have positive impacts on reducing ozone downwind. For example, emission reductions associated with the COVID-19 would lead to decrease of ozone at most sites over South Korea in spring. Global efforts associated with greenhouse mitigation (use of cleaner fuel) eventually help to alleviate ozone pollution.

 – Previous studies published the observed increase in ozone in China and South Korea during the pandemic due to less titration of NOx. This result is contrary to previous studies and please compare the differences between this and previous work.

→ This is the novel aspect of our manuscript. There are several studies reporting the increase of near-surface ozone after COVID lockdowns in the urban areas (e.g., Shi & Brasseur, 2020) because of expected non-linear relationship between ozone and $NO_x$ in the highly polluted regions. However, there are also studies reporting reductions of ozone concentrations from 1 to 8 km altitude in the northern extratropics during COVID

(Steinbrecht et al., 2021). Our study shows both increases and decreases of ozone with COVID-like $NO_x$ emission changes: near-surface ozone concentrations over the polluted regions increase, but there are reductions of ozone concentrations in the elevated layer (Figure R6 and R7). Novel findings in our study are **the decrease of downwind ozone near surface to upper layer with reductions of $NO_x$/VOC emission in upwind pollution hot spots** (see Figure R6 and R7 for several sensitivity runs). For example, 50%-75% of Chinese $NO_x$ emission reductions decrease ozone concentrations in Korea and surrounding seas and the Pacific Ocean from the surface to upper layers although near-surface ozone in Northeast China increases due to these emission changes. Therefore, our study does not fully support the findings in Lee et al. (2021) that stated "These $NO_x$-saturated conditions in megacities contribute to the increased $O_3$ due to $NO_x$ reduction, which could also affect the enhanced $O_3$ concentrations throughout the Asia–Pacific region via long-range transport". Chinese VOC reductions cause reduced ozone concentrations from surface to upper layer and from hot spots to downwind areas. Our study suggests potential changes in photochemical regimes with altitudes over the pollution hot spots ($NO_x$-saturated near surface versus NOx-limited in the elevated layer). Thus, combined effects of vertical and horizontal ozone transport and local production dependent on altitude would determine the ultimate changes in ozone concentrations at certain locations and altitudes. We will add the discussions in the revised manuscript with Figure R6. One thing to note is that the assessment also depends on the accuracy of VOC emissions estimations. This part is vastly uncertain and is the matter of further study.

[Figure]

Figure R6. Differences in the WRF-Chem simulated ozone concentrations ($\Delta O_3$ = $O_3$_emission reduction case-$O_3$_control case) at (top) surface and (bottom) 1000 m above ground level. Green to blue colors (yellow to red colors) denotes reduced (increased) ozone concentration due to the emission changes.

[Figure]

[Figure]

Figure R7. Vertical profiles of ozone from the WRF-Chem model simulations based on various emission scenarios: (top) Seoul, and (bottom) Gosung, Gangwon-do. The model ozone is averaged from 10 LT to 20 LT. Averaged boundary layer height is shaded as cyan color.

Section 4. You presented simulated vertical profiles in Seoul and Gangwondo during the KORUS-AQ. Could you include aircraft observations in Figure 11? I also wonder how the model simulates surface ozone concentrations.

➔ The vertical profiles of ozone from the DC-8 observations and co-located the WRF-Chem results in our study are shown below (Figure R8). The model generally follows the vertical distributions measured by the DC-8 aircraft. The model ozone has a low bias of 16-19 ppb for the cases influenced by the local emissions (Local case: May/4, May/20, June/2, June/3). The model performed better for the cases strongly influenced by the Chinese emissions (Transport case: May/25, May/26, June/1) with a low bias of 3-11 ppb. The EDGAR-HTAP v3 emissions led to the smallest bias for the Transport case. The emission sensitivity runs with doubling Chinese CO and VOC emissions and with doubling both

Chinese and South Korean CO and VOC emissions improve ozone simulations for the Local case, but overestimate ozone concentration for the Transport case. This indicates that more efforts need to be put into the evaluation and improvement of the local CO and VOC emissions estimations. It is still important to improve the emission estimations for China for better ozone simulations of South Korea and beyond. Both surface and boundary layer ozone in the model runs were evaluated and discussed in the responses above. We include this discussion in the Supporting Information. In the revised manuscript, we replace the WRF-Chem model results using EDGAR-HTAPv2 by those using EDGAR-HTAPv3.

[Figure]

Figure R8. Vertically averaged O$_3$ from DC-8 (black), EDV2 (green), EDV3 (blue), and KOV5 (red) for the Local and Transport cases under 2 km height above ground level. The 1/2 of standard deviations are represented with black whiskers in each 200m layer. Sensitivity tests are conducted with doubled anthropogenic CO and VOC emissions in China (EDV3_Ch2, blue triangle dots and dashed lines) and both China and South Korea (EDV3_ChKo2, blue open square and dotted lines). The model results co-located with the observations are sampled and compared with each other. The sampling numbers in the layers are represented with magenta color. (a) and (b) include the data from all flights while (c) and (d) select the data over SMA (Seoul Metropolitan Area).

**References:**

Colombi, N. K., et al., 2022, Why is ozone in South Korea and the Seoul Metropolitan Area so high and increasing? ACP Discuss., https://doi.org/10.5194/egusphere-2022-1366.

Emmons, L. K. et al., 2020, The chemistry mechanism in the Community Earth System Model Version 2 (CESM2), Journal of Advances in Modeling Earth Systems, https://doi.org/10.1029/2019MS001882.

Gaudel, A., et al., 2020, Aircraft observations since the 1990s reveal increases of tropospheric ozone at multiple locations across the Northern Hemisphere, *Sci. Adv.*, 6, eaba8272.

Ghim, Y. S., 2000, Trends and factors of ozone concentration variations in Korea, Journal of Korean Society for Atmospheric Environment, Vol. 16, No. 6, pp 607-623.

Hwang, S.-H., et al., 2007, Observation of secondary ozone peaks near the tropopause over the Korean Peninsula associated with stratosphere-troposphere exchange, *Journal of Geophysical Research: Atmosphere* 112(D16). https://doi.org/10.1029/2006JD007978.

Jeong, YuJoo, et al., 2023, Influence of ENSO on tropospheric ozone variability in Asia, submitted.

Kim, H., et al., 2020, Factors controlling surface ozone in the Seoul Metropolitan Area during the KORUS-AQ Campaign, *Elementa* 8(46): 10.1525/elementa.444. https://doi.org/10.1525/elementa.444.

Kim, Kyoung-Min, et al., 2023, Sensitivity of the WRF-Chem v4.4 ozone, formaldehyde, and their precursor simulations to multiple bottom-up emission inventories over East Asia during the KORUS-AQ 2016 field campaign, *in preparation*.

Lee, H.-J., et al., 2021, Ozone continues to increase in East Asia despite decreasing $NO_2$: Causes and Abatements. *Remote Sensing* 13(11):2177. https://doi.org/10.3390/rs13112177.

Lee, H.-M. and R. J. Park, 2022, Factors determining the seasonal variation of ozone air quality in South Korea: Regional Background versus Domestic Emission Contributions, *Environmental Pollution* 308:119645, https://doi.org/10.1016/j.envpo.2022.119645.

Oh, I.-B., et al., 2010, Elevated ozone layers over the Seoul Metropolitan Region in Korea: Evidence for long-range ozone transport from Eastern China and its contribution to surface concentrations, *Journal of Applied Meteorology and Climatology* 49: 203-20. Httos://doi.org/10.1175/2009JAMC2213.1.

Park et al. (2021). Multi-model intercomparisons of air quality simulations for the KORUS-AQ campaign. ELEMENTA: Science of the Anthropocene, 9(1), 00139, doi: https://doi.org/10.1525/elementa.2021.00139

Shi, X., & Brasseur, G. P. (2020). The response in air quality to the reduction of Chinese economic activities during the COVID-19 outbreak. *Geophysical Research Letters*, 47, e2020GL088070. https://doi.org/10.1029/2020GL088070.

Steinbrecht, W., Kubistin, D., Plass-Dülmer, C., Davies, J., Tarasick, D. W., von der Gathen, P., et al. (2021). COVID-19 crisis reduces free tropospheric ozone across the Northern Hemisphere. *Geophysical Research Letters*, 48, e2020GL091987. https://doi. org/10.1029/2020GL091987.

Wang, W., Parrish, D. D., Wang, S., Bao, F., Ni, R., Li, X., Yang, S., Wang, H., Cheng, Y., and Su, H.: Long-term trend of ozone pollution in China during 2014–2020: distinct seasonal and spatial characteristics and ozone sensitivity, Atmos. Chem. Phys., 22, 8935–8949, https://doi.org/10.5194/acp-22-8935-2022, 2022.

Yeo, M. J., and Y. P. Kim (2021), Long-term trends of surface ozone in Korea, Journal of Cleaner Production, 294, https://doi.org/10.1016/j.jclepro.2020.125352.

---

## Author Comment (AC2)

Reply to the review 2 of "Changes in surface ozone in South Korea on diurnal to decadal time scale for the period of 2001-2021"

Thank you for your comments that improve our manuscript. Our replies to the specific comments are written below (the reviewer's comments in blue and our replies in black).

L2 P2: I believe there is a typo: Change "Increasing trends of tropospheric ozone in South Korea in the last decades have reported in several studies" to "Increasing trends of tropospheric ozone in South Korea in the last decades have been reported in several studies".
→ We corrected this typo in the revised manuscript.

L4 P5: Could you give some details on the impact of the COVID-19 pandemic on the atmospheric composition in spring versus in summer? Has South Korea experienced several lock-downs in spring and summer 2020, or only in spring, with reduction of human activities/emissions of the precursors of ozone?
→ Nationwide social distancing protocol enforced by Korean government started February 25 of 2020 and lasted until April 18 of 2022, although levels of protocol differ. During spring in 2020 (until May 6, 2020), facilities for public use (libraries, swimming pools, museums, and national parks) and religious, indoor sports, entertainment facilities were forced to close, and people were refrain from going out except for buying necessities, visiting a doctor, and commuting to/from work. Since May 6 of 2020, as number of new confirmed COVID-19 cases remain relatively steady, the guidelines have shifted from social distancing to distancing in daily life, no restrictions on people going out. Because a cluster of new COVID-19 cases emerged in mid-August, social distancing protocol (since August 16 until early October) was again forced by the government, people were strongly recommended to stay indoors. After August 16 of 2020, there were well-defined government protocols as Level1, 2, and 3: Level1 is no restricted personal gathering and daily life, Level 2 allows personal gathering up to 8 people and discourage unnecessary and unurgent travel, and Level 3 allows personal gathering up to 3 people, requires remote work and

online classes, and discourage travels. Most days in spring and summer in 2021 were the period under the Level2 protocol. In summary, most distinct changes in social-distancing protocols and traffic/mobile activities occurred between spring and summer in 2020. This discussion is now included in the revised manuscript.

L12 P5: I believe there is a typo: Change "as following" to "as follow".

→ We corrected this typo in the revised manuscript.

L7 P6: Could you be more specific? Could you give the starting year? Are all the 500 stations still working now? Maybe add a column "time period" in Table S1.

→ More specific information on the time period of the observations and missing period is given in a excel file as Supporting Information.

L9 P7: Could you be more specific on the stricter recommendations: quality assurance and cloud fraction?

→ The stricter recommended filter is selecting pixels passing quality assurance > 0.75 and cloud radiance fraction < 0.5.

L10 P7: Have you conducted or are you aware of any sensitivity test to see how much the compromise sampling statistics/quality may change the results?

→ We conducted the sensitivity test by applying different sampling conditions and found consistent results irrespective of quality control parameters: larger tropospheric $NO_2$ column reduction during spring than during summer between 2019 and 2020-2021 (COVID-19 periods). Differences between KNMI and NASA retrievals are large when the original filter was applied (quality assurance > 0.5 and cloud radiance fraction < 0.4). When the stricter filter was applied, differences between KNMI and NASA retrievals are small. Therefore, in the revised manuscript, the stricter filter (quality assurance > 0.75 and cloud radiance fraction < 0.5) is used. Since the NASA product released in November, 2022 were generated in a consistent manner for May 2018-December 2021, we presented the NASA MINDS product in the revised manuscript

instead of the KNMI product. We summarized the sensitivity tests in the Supporting Information. The distribution of absolute tropospheric $NO_2$ columns for different years are also shown in the Supporting Information.

L4 P9: Typo: Change "11st" to "11th" (eleventh). Could you add the year?
→ Yes. We added year "2016". We corrected this part to 11th June 12 UTC in 2016.

L11 P10: Could you add the uncertainties on the trend estimate?
→ We included the uncertainties on the trend as the reviewer suggested.

L14 P10: "Insignificant" is not used anymore (Wasserstein et al., 2019). Trend reliability can be expressed with p-value (Wasserstein et al., 2019) and/or signal-to-noise (SNR) ratio (Chang et al., 2021). Then you can apply the trend reliability scale (see table below from the guidance note on best statistical practices for tropospheric ozone assessment report -TOAR-analyses by Kai-Lan Chang, Martin Schultz, Gerbrand Koren and co-authors pending their approval, February 2023; the document will be posted on the TOAR website by end of April 2023 upon the TOAR steering committee approval, https://igacproject.org/activities/TOAR/TOAR-II) to report the trend and its uncertainty.

Table 3. Trend reliability scale

| p-value | SNR (signal-to-noise) value | Term |
|---|---|---|
| $p \le 0.01$ | $SNR \ge 3$ | very high certainty |
| $0.05 \ge p > 0.01$ | $2 \le SNR < 3$ | high certainty |
| $0.10 \ge p > 0.05$ | $1.65 \le SNR < 2$ | medium certainty |
| $0.33[1] \ge p > 0.10$ | $1 \le SNR < 1.65$ | low certainty |
| $p > 0.33[1]$ | $SNR < 1$ | very low certainty or no evidence |

[1]This boundary is meant to be fuzzy around 1/3 (Mastrandrea et al., 2010).

Table taken from the guidance note on best statistical practices for tropospheric ozone assessment report -TOAR-analyses by Kai-Lan Chang, Martin Schultz, Gerbrand Koren and coauthors pending their approval, February 2023; the document will be posted on the TOAR website upon the TOAR steering committee approval, https://igacproject.org/activities/TOAR/TOAR-II.

→ We added p-value and SNR in a separate Table in the main text. The table is displayed below.

Table R4. Trends estimates based on the 4th highest MDA8 $O_3$ values

| Location | | Slope (ppb yr$^{-1}$) | 2-Sigma (ppb yr$^{-1}$) | P value | SNR |
|---|---|---|---|---|---|
| City | Seoul (SUL) | 1.19 | 0.38 | < 0.01 | 6.23 |
| | Incheon (INC) | 1.07 | 0.37 | < 0.01 | 5.72 |
| | Daejeon (DJN) | 1.22 | 0.49 | < 0.01 | 4.96 |
| | Gwangju (GWJ) | 0.98 | 0.46 | < 0.01 | 4.30 |
| | Busan (BSN) | 0.98 | 0.36 | < 0.01 | 5.47 |
| | Ulsan (ULS) | 1.40 | 0.34 | < 0.01 | 8.14 |
| | Daegu (DGU) | 1.12 | 0.46 | < 0.01 | 4.89 |
| Province | Gyeonggi-do (GGI) | 1.26 | 0.27 | < 0.01 | 9.33 |
| | Chungcheongbuk-do (CCB) | 0.79 | 0.51 | < 0.01 | 3.09 |
| | Chungcheongnam-do (CCN) | 1.45 | 0.47 | < 0.01 | 6.12 |
| | Jeollabuk-do (JLB) | 1.83 | 0.32 | < 0.01 | 11.30 |
| | Jeollanam-do (JLN) | 0.08 | 0.39 | 0.67 | 0.41 |
| | Jeju Island (JEJ) | 0.66 | 0.46 | < 0.01 | 2.89 |
| | Gyeongsangnam-do (GSN) | 0.83 | 0.52 | < 0.01 | 3.18 |
| | Gyeongsangbuk-do (GSB) | 1.10 | 0.35 | < 0.01 | 6.32 |
| | Gangwon-do (GWO) | 0.67 | 0.48 | < 0.01 | 2.79 |

L5 P11: Spell out LT = Local Time, at least the first time it is used.

→ Corrected.

L12 P11: It would be worth adding a discussion with references on summer/spring differences: meteorology condition in Seoul and Gyeonggi-do compared with other sites/regions. That would probably fit in the "Discussions" section.

→ The mean temperature, mean maximum temperature, and mean wind velocity values during spring and summer, 2001 – 2021 are listed in Table R5. Unlike opposite patterns of spring/summer peak time ozone in Seoul and Gyeonggi-do, the meteorological factors show

similar differences in the area of interests. Thus, the meteorological factors are not main drivers of high summertime exceedances in Seoul and Gyeonggi-do region. The data are obtained from the Korea Meteorological Administration (KMA) website (https://data.kma.go.kr/).

Table R5. Spring and summer mean temperatures, mean maximum temperatures, and mean wind velocities in Korean metropolitan cities and provinces. Differences in values between spring and summer are in the parenthesis. The cities and provinces listed in the table are in counterclockwise order in regards to the South Korean map.

| Location | | Mean temperature (°C) | Mean maximum temperature (°C) | Mean wind velocity (m/s) |
|---|---|---|---|---|
| | | Spring / Summer (difference) | | |
| City | Seoul | 12.4 / 24.9 (-12.5) | 17.7 / 29.0 (-11.3) | 2.6 / 2.2 (0.4) |
| | Incheon | 11.6 / 23.9 (-12.3) | 16.1 / 27.5 (-11.4) | 3.2 / 2.5 (0.7) |
| | Daejeon | 12.9 / 24.9 (-12.0) | 19.1 / 29.3 (-10.2) | 2.0 / 1.8 (0.2) |
| | Gwangju | 13.5 / 25.2 (-11.7) | 19.7 / 29.8 (-10.1) | 2.0 / 2.0 (0.0) |
| | Busan | 13.7 / 24.0 (-10.3) | 18.1 / 27.4 (-9.3) | 3.5 / 3.2 (0.3) |
| | Ulsan | 13.6 / 24.4 (-10.8) | 19.1 / 28.7 (-9.6) | 2.3 / 2.0 (0.3) |
| | Daegu | 14.3 / 25.5 (-11.2) | 20.3 / 30.3 (-10.0) | 2.4 / 2.2 (0.2) |
| Province | Gyeonggi-do | 11.5 / 24.0 (-12.5) | 17.1 / 28.4 (-11.3) | 2.3 / 2.0 (0.3) |
| | Chungcheongbuk-do | 11.6 / 23.7 (-12.1) | 18.4 / 28.8 (-10.4) | 2.1 / 1.5 (0.6) |
| | Chungcheongnam-do | 11.3 / 24.0 (-12.7) | 17.8 / 28.8 (-11.0) | 2.0 / 1.6 (0.4) |
| | Jeollabuk-do | 12.3 / 24.7 (-12.4) | 18.7 / 29.6 (-10.9) | 1.9 / 1.6 (0.3) |
| | Jeollanam-do | 12.6 / 24.2 (-11.6) | 18.0 / 28.2 (-10.2) | 3.0 / 2.5 (0.5) |
| | Jeju-do | 14.7 / 25.1 (-10.4) | 18.4 / 28.1 (-9.7) | 3.1 / 2.8 (0.3) |
| | Gyeongsangnam-do | 13.0 / 24.4 (-11.4) | 19.6 / 29.4 (-9.8) | 1.8 / 1.5 (0.3) |
| | Gyeongsangbuk-do | 12.4 / 23.7 (-11.3) | 18.8 / 28.7 (-9.9) | 2.3 / 1.7 (0.6) |
| | Gangwon-do | 11.5 / 23.4 (-11.9) | 17.6 / 28.2 (-10.6) | 2.0 / 1.6 (0.4) |

L15 P11: I found 7 sites showing more exceedances in summer than in springs according to Figure 4. Why do you report only 3 of them? I also found 10 sites showing more exceedances in spring than in summer, why do you report only 3 of them?

→ We just exemplified the diurnal cycles for representative cases since Figure 4 also have this information. In the revised manuscript, we included the diurnal variations at all locations in the Supporting Information.

L7 P12: "than Incheon" is not clear. I believe there is a typo in the sentence. Could you rephrase?

→ We changed to "compared to the time of exceedance in Incheon".

L13-14 P12: Is it a statement from previous studies or from this current study? Could you give a reference or cite a figure to support this statement?

→ During nighttime, NO reacts with ozone forming $NO_2$ and oxygen molecule, which is the main loss of ozone (Jacob, D. J., 1999; Seinfeld and Pandis, 2016). In Figure R9, both model and observations exhibit high $NO_2$ concentrations and low ozone concentrations during night.

[Figure]

Figure R9. The diurnal variations of observed and simulated $O_3$ and $NO_2$ averaged for the simulation period.

L20 P14: Does "large reduction of ozone" refer to the difference between the time periods P2 and P3? It would be helpful to clarify.

→ Yes. It means the time periods P2 and P3. We clarified it.

L14 P15: Does "likely to be VOC-limited" mean that VOCs did not decrease between P2 and P3 in South Korea? Any reference?

→ It meant that "VOC-limited" is a dominant photochemical regime in the cities over South Korea (e.g., Kim et al., 2020). We clarified in the manuscript.

L20 P15: Do we know why there are more $NO_2$ in MAM 2019 than JJA 2019? Specific human activities, Meteorological conditions? It would be interesting to see the maps of MAM 2020 and JJA 2020.

→ Meteorological condition such as sunlight is the main driver. $NO_2$ concentrations at surface or vertically integrated column concentrations are lower during summer than during spring because of enhanced OH radical concentrations due to increased sunlight during summer increase loss of $NO_2$ via a reaction of $NO_2$ with OH (Martin et al., 2003; Lamsal et al., 2010). The reduced chemical lifetime of $NO_x$ leads to decreased $NO_2$ columns in JJA 2019 compared to those in MAM 2019. We also included the maps of TROPOMI $NO_2$ columns for MAM 2020 and JJA 2020 in the Supporting Information.

L20 P16: Why did you choose Seoul and Gangwon-do over other sites?

→ In the reply to the Reviewer1, we explained the reason to investigate Gangwon-do, in particular Gosung. The elevations of monitoring sites in Gangwon-do are high as in Table R6. Gosung (Ganseong-eup in Table R6) is elevated to ~600 m, is located to leeside of mountain, and is close to the East Coast of South Korea. Therefore, this remote site is ideally located to investigate the impacts of long-range transport of ozone at high elevations.

Table R6. Altitudes (m) of monitoring sites in Gangwon-do. Ganseong-eup represents Gosung.

| | Name | Latitude | Longitude | Altitude |
|---|---|---|---|---|
| | Jungangno | 37.87564 | 127.72048 | 110.1613 |
| | Seoksa-dong | 37.85707 | 127.7495 | 195.0629 |
| | Okcheon-dong | 37.76003 | 128.90297 | 81.9188 |
| | Jungang-dong | 37.35279 | 127.94746 | 194.5183 |
| Gangwon | Bangok-dong | 37.3356 | 127.9771 | 274.9333 |
| | Ganseong-eup | 38.28744 | 128.38521 | 586.4231 |
| | Bangsan-myeon | 38.22439 | 127.95856 | 456.5462 |
| | Bukpyeong-myeon | 37.43023 | 128.66476 | 631.8139 |
| | Chiaksan | 37.36014 | 128.12509 | 587.2285 |

L1 P17: An evaluation of WRF-Chem above Seoul and Gangwon-do would be helpful. How does the control run compare with the observations? Any sondes launched during KORUS-AQ that can be used for this evaluation? Was this model study done with annual means or did you perform it for a specific season? Showing summer and spring would be useful to echo the seasonal results on trends estimate.

→ The model results from the WRF-Chem control run were compared with the observations from the surface monitor over Seoul and Gosung in Figure R10 and R11. The model decently simulated the observations in an hourly basis (Figure R10) and on average (Figure R11). The model was conducted for the KORUS-AQ field campaign (May 1 – June 10 in 2016) and was averaged for the period. The model simulation period covers mainly springtime. Longer simulations will be required to contrast spring and summer. This is an interesting modeling topic for future study. In reply to the Reviewer 1, we showed the evaluations of vertical profiles of simulated ozone with the DC-8 aircraft observations.

[Figure]

Figure R10. The time series of observed and simulated hourly ozone in (top) Seoul and (bottom) Gosung. Basic statistics are shown as follows. Mean bias (MB): Seoul -6.2 ppb /Gosung -0.9 ppb, Root Mean Square Errors (RMSE): Seoul 18.2 ppb/Gosung 13.7 ppb, Correlation Coefficient(R): Seoul 0.68/Gosung 0.54.

[Figure]

Figure R11. Diurnal variations of observed and simulated ozone concentrations averaged for the entire simulation period: (top) Seoul and (bottom) Gosung. Basic statistics are shown in the plot.

L7 P17: It seems to be very small changes (almost none). Could you be more quantitative?

→The reduction is -1 ppb (-2%). In the revised manuscript, we used EDGAR-HTAPv3 emission inventory. This statement was omitted.

L3-5 P18: You probably should inform on the altitude of both Gosung and Gangwon-do sites because it is a little confusing as it is written.

→ The altitudes are informed in Table R6. We include this information in Supporting Information.

L1-2 P35: Are $NO_2$ and CO values from CAM-Chem? It is worth clarifying in the caption.

→ The trends are calculated from the surface monitor observations (www.airkorea.or.kr). We clarified it.

L2 P36: Can you have colors or signs to differentiate cities, provinces and background sites, as well as the definitions of these three categories. Is it according to ozone diurnal/seasonal variability? Could you add a legend?

→ We used the colors to differentiate the three categories. We added it to the Figure caption.

L2 P37: Could you add the uncertainties (2-sigma values), or p-value or signal-to-noise ratio associated with the slope values S? (see my previous comment on how to report trend and its uncertainty)

→ We added p-value and SNR in the newly added Table in the revised manuscript.

L4 P41: Is the extraction over the entire country? It should be specified in the caption and section 2.4.

→ Yes. It was extracted over the entire country. Now we include this information in the Figure caption in the revised manuscript.

L4 P44: Typo in the legend of Figure 11: change "Contorl" to "Control"

→ The typo is corrected in the revised manuscript. Thank you for paying attention to detail.

References:

Chang, K.-L., Schultz, M. G., Lan, X. et al. (2021). Trend detection of atmospheric time series: Incorporating appropriate uncertainty estimates and handling extreme events. Elementa: Science of the Anthropocene, 9.

Wasserstein, R.L., Schirm, A. L. & Lazar, N. A. (2019). Moving to a world beyond "p < 0.05". The American Statistician.

**References**

Jacob, D. J. (1999), Introduction to Atmospheric Chemistry, *Princeton University Press*, 260pp.

Kim, H., et al., 2020, Factors controlling surface ozone in the Seoul Metropolitan Area during the KORUS-AQ Campaign, Elementa 8(46): 10.1525/elementa.44    4.

Lamsal, L. N., et al., 2010, Indirect validation of tropospheric nitrogen dioxide retrieved from the OMI satellite instrument: Insight into the seasonal variation of nitrogen oxides at northern midlatitudes, 115, D5, https://doi.org/10.1029/2009JD013351.

Randall, V. M., et al., 2003, Global inventory of nitrogen oxide emissions constrained by space-based observations of $NO_2$ columns, 108, D17, https://doi.org/10.1029/2003JD003453.

Seinfeld, J. H. and Pandis, S. N. (2016), Atmospheric Chemistry and Physics: From Air Pollution to Climate Change (3rd ed.), Wiley, 1152pp.

---

## Author Comment (AC3)

Reply to the review 3 of "Changes in surface ozone in South Korea on diurnal to decadal time scale for the period of 2001-2021"

Thank you very much for your insights about trend and seasonality of background ozone values at northern midlatitude. The background ozone beyond Asia should have been discussed in the manuscript. In the revised manuscript, we included the references and mentioned this point. We also thank you for recognizing the strengths of our study. Our replies to the major concerns and specific comments are written below (the reviewer's comments in blue and our replies in black).

The reviewer's main concern was the use of surface $O_3$ data from the base time (01-06 LT) to gain information about background value because $O_3$ loss reacting with NO is dominant at this time over the highly polluted area. It is typical to ignore the data at this time when analyzing trends over polluted regions. However, in this study, we would like to utilize $O_3$ data at this time to find information about background $O_3$ because ozone is transported throughout a day and this process is very important in the region of study. The Figure R12 shows the WRF-Chem simulated surface $O_3$ in Seoul from various emission scenarios. Blue line in the plot denotes the model results only with local emissions (zero-out Chinese emissions, labeled as "No China") and black line represents the results from Control run with all emissions. The local emissions case (blue line) shows much reduced $O_3$ compared to the Control case throughout a day (including 01-06 LT). The difference between the Control case (black line) and local emissions case (blue line) at 01-06 LT indicates increase of ozone from transport from upwind sources at this time.

High $NO_x$ condition in Seoul tends to suppress the photochemical production of $O_3$ during daytime and enhance $O_3$ destruction during nighttime as exhibited in differences between black (Control case) and magenta lines (zero-out Seoul emission case, labeled as "No Seoul"). This indicates that chemistry plays a critical role in determining $O_3$ value in Seoul. Therefore, similarity of mean $O_3$ values in the Control case to clean background tropospheric $O_3$ value (climatological value) may be just a coincidence. These modeling exercises demonstrate that $O_3$ at the base time can be analyzed to derive information about background ozone even over the highly polluted (high $NO_x$) sites. The other point the reviewer commented is the impact

of different NO$_x$ concentrations during spring and summer on background ozone at 01-06 LT. Because it is not daytime, differences in boundary layer height between the two seasons should be small. Lower stable boundary layer height during summer than during spring is not well theoretically supported.

[Figure]

Figure R12. Diurnal variations of the model ozone concentrations at surface from various emission scenarios. The model results were averaged for the full simulation period.

The reviewer's suggestion to construct frames to analyze O$_3$ in South Korea by using the observations and chemical transport model results sounds interesting, but we are not sure if that can/should be conducted in this study. It would not be straightforward to delineate background O$_3$ (without continental influences) and to assess the impacts of local South Korean emissions and Asian mainland emissions by mainly analyzing observations for the complex atmospheric environment of South Korea. We agree with the reviewer to the point that the models like CESM constrain many important parameters to develop the model reproducing O$_3$ seasonality and trends. To rely on the models, however, the uncertainties of the models should be well accounted for. This alone is a quite challenging work. It would be interesting to conduct

research the reviewer suggested. But it would require considerable times and that work would be beyond the scope of this study. In this study, we used models to help interpret observations as shown in the discussion section in the original manuscript, which is moved to the result section in the revised manuscript.

**Summary**:
This a very useful and informative paper. It has two major strengths: 1) Investigation of ozone in a region with strong local anthropogenic emissions, that also receives marine inflow with a highly polluted continent lying directly upwind of the marine area. 2) An effective incorporation of both observation and modelling based analysis. However, I believe that a major revision of the paper is required before it is ready for publication. One major need is for the authors to begin their observation-based analysis with a consideration of the ozone distribution that would be present in South Korea if there were no continental influences, i.e., if observed concentrations were due to transported baseline ozone alone. That consideration can rely on both the CESMv2.2 model calculations of these ozone concentrations (evidently shown in Figure 6), where results at ~1 km likely represent baseline ozone, and on analysis of observations as suggested below in the first major issue. This consideration would then provide a basis for understanding the continental influences, both from local South Korean emissions and from the Asian mainland emissions. Also much of the discussion is difficult to follow and requires substantial improvement; suggestions in this regard are given in the major and minor issues described below.
→ Please see the replies above.

**Major issues**:
1)  I believe that the discussion based on Table 1 requires reconsideration. I assume that these are mean ozone concentrations for the peak and base time periods in spring and summer. First, I think the period names are misleading. The 10-20 LT period has higher ozone concentrations than does the 01-06 period. However, those higher (10-20 LT) ozone concentrations are similar to that expected for northern mid-latitude baseline ozone concentrations. For example, Figure 5 of Parrish et al., (2020) shows that annual mean ozone is 30 to 40 ppb in the lower 1 km of the troposphere. Figure S14 of that paper shows that ozone at Mt. Walinguan (upwind of South Korea, but at higher elevation) has mean ozone of about 45 to 60 ppb in spring and summer. To my mind, the mean ozone in Table 1 in the 10-20 LT period predominately reflects baseline ozone transported into the country; this is the reason that these mean concentrations are similar throughout the country.

→ We agree with the reviewer about the possibility of baseline ozone transported into the country, judging from similar mean values throughout the country. However, Figure R12 also illustrates various responses of surface ozone to emission scenarios in Seoul. It demonstrates

that chemistry is an important factor to determine mean annual ozone in Seoul and other regions in South Korea. Therefore, we would like to avoid oversimplification of factors to determine the ozone in South Korea.

2) If the interpretation above is correct, then the lower ozone concentrations in the 01-06 period are caused by loss of ozone due to surface deposition and reaction with fresh NO emissions under a shallow nocturnal inversion. Such a diurnal cycle (low at night, higher during the day) is a ubiquitous feature of urban ozone.

→ Agreed. See the replies above (including Figure R12). Ozone in the 01-06 LT period is lower than that in the 10-20 LT period because of different chemical and physical processes involved. But there are still influence of transport in the 01-06 LT period as shown in Figure R12. Therefore, we would like to use the data in the 01-06 period.

3) To emphasize the similarity of the ozone concentrations throughout the country, and the predominant role of transported baseline ozone, I suggest that the background sites be included in Figure 3 and Table 1.

→ It was difficult to derive the trends for the background sites because some of ozone season data are missing for the sites. The data from March 31, 2011 to August 31, 2011 are missing in Gosung, Gangwon-do. The data from Mary 1, 2012 to June 8, 2012 are missing in Gosan, Jeju. The data from March 30, 2011 to June 30, 2011 are missing in Ulleung Island (in Gyeongsangbuk-do). Therefore, we limit the trend analysis for the region with multiple monitoring sites covering the full period of analysis.

4) More generally, I suggest that all tables, figures and discussion clearly address the 7 cities, 9 provinces, and 3 background sites in a consistent manner to the fullest extent possible. The discussion is often difficult to follow when varying lists of cities, provinces and sites are mentioned.

→ We presented the results for the 7 cities, 9 provinces, and 2 background sites consistently in the revised manuscript whenever possible. We omitted Gosan, Jeju Island because $NO_2$ and CO data need quality assurance from mid of 2010 to current date. The names of the sites were

updated consistently throughout the manuscript. For the trend study (Figure 2 and 3), we did not include the background sites because of some missing data during ozone season.

5) The primary reason that mean ozone is generally higher in spring than in summer is that the lower troposphere baseline ozone is higher in spring than in summer, particularly in marine influenced air; e.g., see Figures 4 and 6 of Parrish et al., (2020).

→ Thank you for the reference. We explained the seasonal difference including marine influenced air in the revised manuscript and referred to Parrish et al (2020).

6) Pg. 11, lines 5-8: One reason the 01-06 LT ozone is higher in the spring is that the nocturnal inversion is tighter in the summer, so ozone loss at night is more pronounced in summer than in spring. Given the very local processes that determine the 01-06 LT ozone, for simplicity, the authors may wish to eliminate the discussion of this nighttime ozone.

→ We don't think that there are clear mechanisms driving differences in nocturnal inversion between spring and summer. See the replied above for the reason why we keep the discussions about the ozone concentrations in the 01-06 LT period.

7) A discussion of local CO and NOx trends begins near the bottom of pg. 13. These observation-based trends should be compared and discussed in relation to the trends of these species derived from the model emission inventories. This may be more relevant to NOx, since it does have more local influence than CO.

→ We listed the trends of NOx and CO emissions from linear fits of the data covering 2001-2020, obtained from Clean Air Policy Support System (CAPSS) emission inventory (https://www.air.go.kr/) (Table R7 and R8 for emission inventories and ambient concentrations, respectively). Overall, signs of slopes agree between emission inventory and ambient concentrations at least for the cities, but site-to-site variations do not agree even for the cities. And there are disagreements of signs of slopes between emission inventory and ambient concentrations for the provinces. This can be attributed to the uncertainties in long-term emission inventories of NOx and CO.

Table R7. The trends of NOx and CO emissions from linear fits of the data covering 2001-2020.

| Stations | | NOx (kton/yr) Slope (Correlation Coefficient) | CO (kton/yr) Slope (Correlation Coefficient) |
|---|---|---|---|
| City | Seoul | -2.35 (-0.72) | -8.02 (-0.97) |
| | Incheon | -1.14 (-0.60) | -0.74 (-0.73) |
| | Daejeon | -0.56 (-0.84) | -0.84 (-0.88) |
| | Gwangju | -0.29 (-0.63) | -0.72 (-0.94) |
| | Busan | -1.23 (-0.77) | -2.01 (-0.94) |
| | Ulsan | -1.27 (-0.90) | -0.12 (-0.37) |
| | Daegu | -0.85 (-0.74) | -1.37 (-0.87) |
| Province | Gyeonggi-do | -1.30 (-0.47) | -1.51 (-0.67) |
| | Chungcheongbuk-do | 0.52 (0.46) | 0.40 (0.40) |
| | Chungcheongnam-do | -5.32 (-0.74) | 1.49 (0.93) |
| | Jeollabuk-do | -0.66 (-0.82) | 0.61 (0.53) |
| | Jeollanam-do | 0.74 (0.57) | 1.63 (0.75) |
| | Jeju-do | 0.27 (0.64) | 0.31 (0.58) |
| | Gyeongsangnam-do | -5.47 (-0.83) | -0.09 (-0.14) |
| | Gyeongsangbuk-do | 0.78 (0.52) | 2.19 (0.76) |
| | Gangwon-do | 0.25 (0.17) | 0.95 (0.67) |

Table R8. The observed trends of $NO_2$ and CO concentrations from linear fits of the data covering 2001-2021.

| Stations | | $NO_2$
Spring / Summer
Slope (Correlation Coefficient) | CO
Spring / Summer
Slope (Correlation Coefficient) |
|---|---|---|---|
| City | Seoul | -0.77 (-0.85)/-0.72(-0.91) | -7.56(-0.77)/-5.34(-0.83) |
| | Incheon | -0.37(-0.62)/-0.50(-0.62) | -7.65(-0.71)/-4.64(-0.66) |
| | Daejeon | -0.10(-0.29)/-0.12(-0.50) | -15.53(-0.79)/-9.71(-0.64) |
| | Gwangju | -0.51(-0.85)/-0.35(-0.88) | -10.64(-0.81)/-8.00(-0.69) |
| | Busan | -0.64(-0.89)/-0.49(-0.90) | -12.32(-0.83)/-11.05(-0.81) |
| | Ulsan | -0.04(-0.08)/-0.06(-0.16) | -4.80(-0.39)/-0.75(0.07) |
| | Daegu | -0.65(-0.87)/-0.51(-0.89) | -23.49(-0.90)/-19.87(-0.87) |
| Province | Gyeonggi | -0.41(-0.66)/-0.44(-0.79) | -14.50(-0.95)/-8.82(-0.94) |
| | Chungcheongbuk | -0.18(0.39)/-0.16(-0.45) | -17.68(-0.78)/-6.49(-0.61) |
| | Chungcheongnam | -0.10(-0.30)/-0.12(-0.41) | -20.95(-0.76)/-9.33(-0.69) |
| | Jeollabuk | -0.17(-0.42)/-0.25(-0.65) | -21.33(-0.87)/-15.07(-0.85) |
| | Jeollanam | -0.21(-0.51)/-0.21(-0.58) | -5.86(-0.53)/-5.32(-0.48) |
| | Jeju Island | -0.18(-0.38)/-0.16(-0.46) | -10.74(-0.71)/-6.95(-0.50) |
| | Gyeongsangnam | -0.12(-0.31)/-0.10(-0.40) | -6.76(-0.58)/-3.92(-0.46) |
| | Gyeongsangbuk | -0.76(-0.89)/-0.49(-0.88) | -27.54(-0.82)/-17.48(-0.78) |
| | Gangwon | -0.16(-0.50)/-0.20(-0.69) | -15.31(-0.86)/-9.03(-0.71) |

8)   The discussion of the COVID-19 influence on ozone (pg. 14-15) is interesting, particularly the "large reduction of ozone in the background sites". There are other studies of the influence of COVID-19 emission reduction on background ozone at northern mid-latitudes. The findings in these other studies should be quantitatively compared to the present results.

→ Thank you for introducing publications. In the revised manuscript, we refer the study by Steinbrecht et al. (2021) that reported about 7% reductions of mid-latitude free atmosphere ozone concentrations in 2020 from the climatology value covering 2000-2020. Our study focused on the analysis surface ozone over South Korea that substantially increased for the period of 2000-2020. Thus, it is not straightforward to quantitatively compare the anomaly in 2020 from climatology in this study with that in Steinbrecht et al. (2021). It is still worthwhile to mention agreement in declining ozone concentration/exceedances during COVID in our study and Steinbrecht et al (2021).

9) I find the discussion beginning on line 14, page 13 and continuing to the end of the Results section on page 16 to be very confusing, with many topics discussed in a disjointed manner. Please revise and clarify this discussion. Any topic that cannot be clearly and concisely explained without speculation should be eliminated.

→ Agreed. There are indeed many topics. Following your suggestions, to clarify the contents, we made several subsections with appropriate titles within the results section. The discussion section is also incorporated into the result section. The titles for the subsections in the results section in the revised manuscript are written below.

3.1 Surface ozone trends

3.2 Difference between spring and summer: background value, exceedance, stratospheric influence, and precursor concentrations

 3.2.1 Background values at the base and peak times

 3.2.2 Ozone exceedances

 3.2.3 Influence of stratospheric ozone

 3.2.4 Long-term trends of surface $NO_2$ and CO concentrations

3.3 Changes detected during the COVID-19 pandemic (2020-2021) compared to 2002-2019

 3.3.1 Changes in ozone exceedances and local precursors during springtime

 3.3.2 Changes in ozone exceedances and local precursors during summertime

 3.3.3 Changes in precursor concentrations at a regional scale during spring and summer: TROPOMI tropospheric $NO_2$ columns

3.4. Impacts of changes in East Asian emissions on surface/boundary layer ozone in South Korea: a modeling analysis

 3.4.1. Changes in surface/boundary layer ozone due to emissions reductions: East Asian region

 3.4.2. Vertical sensitivity of ozone changes in South Korea to East Asian emission Reductions

 3.4.3. Comparisons with recent modeling research

10) Similarly, the Section 4 discussion section is difficult to follow. The authors should aim to convey the main points of the modeling results as clearly and concisely as possible. The last two sentences of the section appear to be the main points; they should be clearly and concisely supported by the preceding discussion.

→ The discussion section is now incorporated into the results section for better support of the content and a smooth connection. We emphasized the last two sentences by reorganizing the results section and adding more explanations.

11) The Summary and Conclusions section will need to be rewritten when the issues identified here are addressed. Specifically:

• The ozone in the 01-06 LT period is so affected by local conditions that it should not be included in the 2nd paragraph of this Section.
→ Please see our replies above. We kept using data at 01-06 LT to get information about background/transport.

• Page 19, discussion beginning on line 17 should be improved. If there is strong influence of long-range transport on the surface ozone at the background sites, then that influence must also be present at all sites throughout South Korea. That influence is not apparent at night at most sites due to rapid nighttime loss of ozone at most sites.
→ Please see our replies above. Even with rapid nighttime loss of ozone at most sites, there is still information about long-range transport. We would like to maximize the use of the data at 01-06 LT.

• An explanation should be given as to why there is such large regional differences in overall percentage decline in $NO_2$. Perhaps this can be related to the model emission inventory?
→ There are many sources of $NO_2$ besides mobile sources in South Korea, such as power plant and industries. Thus, decline of $NO_2$ varies at the monitoring sites that have different source profiles. As mentioned, uncertainty in the emission inventory is generally large and was not extensively estimated.

**Minor issues**:
1) Pg. 4, Line 11: Four references are given for papers that have previously reported increasing ozone trends in South Korea. Two of those are missing from the reference list. The introduction should briefly summarize what these papers found, and discuss the advances that the authors' make in this paper beyond what is known from those earlier papers.
→ The missing references were added. We clarify the contribution of our study compared to the previous study. In reply to the Reviewer 1, we wrote "The published works on the trend of surface ozone in South Korea presented the ozone metrics such as annual mean of hourly

ozone, annual mean of MDA8 ozone, annual mean of daily maximum hourly ozone, and frequency of hourly concentrations greater than 120 ppb. The trends based on those metrics have already been published (e.g., Yeo and Kim, 2021). Since the US EPA National Ambient Air Quality Standard (NAAQS) for ozone is 70 ppb, as the fourth-highest MDA8 ozone concentration, averaged across three consecutive years, and the recent study by Wang et al. (2022) adopted the 4th highest MDA8 ozone concentrations as one of the metrics for study of Chinese ozone pollution, it would be nice to have analyses adopting the 4th highest MDA8 ozone for a global comparison. The EPA standard is also designed for public health protection. Exceedances presented in our study are similar to the frequency exposed to MDA8 ozone > 70 ppb (relevant to EPA standard)". This state some of our contributions to ozone analysis over South Korea, compared to the previous studies. This study reveals characteristics of newly defined exceedances (hourly O3 concentration > 70 ppb) that captured large changes of ozone during COVID and emphasizes long-range transport of ozone over eastern part of South Korea such as Gangwon-do and Ulleung-Island.

2) There are minor problems with the English usage, which should be corrected by editing by a native English speaker.

→ We improved English for the revised manuscript with the aid of a native speaker without changing contents.

3) Page 5, line 9 mentions that 8 provinces are studied; however Table 1 lists 9 provinces. Please develop a list of cities, provinces, and background sites, and consistently use that list throughout the paper.

→ We kept listing 9 provinces in the main text, tables, and figures in the revised manuscript.

4) In the Figure 1 caption, the different colors used for the city province and site names should be described. Also it is not clear exactly what is being plotted here: Is each symbol the mean 4th highest (MDA8) over all sites in the city or province? Confidence limits should be given for all derived slopes.

→ In the revised manuscript, we explained the meaning of different colors in Figure 1. In Table 1 in the revised manuscript, we showed slope, standard deviation, P-value, and signal-to-noise value. The information about all sites in the city or province is shown in the Supporting Information.

5) In the description of the two models evidently different anthropogenic emission inventories are used in the two models (CMIP6 for 2000-2014 and SSP5-8-5 for 2015-2020 in CAM-Chem and WRF-Chem and EDGAR-HTAPv2). There should be a brief discussion regarding how well these inventories compare, and if any problems arise

from using perhaps incompatible emissions in the two models. Also mention should be made regarding whether these inventories correctly simulate the emissions reductions during the COVID-19 period.

→ CMIP6 is based on EDGAR v4.2 or v4.3.2 described in Feng et al. (2020). SSP5-8.5 and EDGAR-HTAP v3 can be compared for the KORUS-AQ campaign period in 2016, as the WRF-Chem simulations were conducted during the period. In this reply, we compared $NO_x$ emissions of SSP5-8.5, EDGAR-HTAP v3, v2, and KORUS v5. In Table R9, over China, SSP5-8.5 NOx emissions are slightly larger than those in KORUS v5 and are lower than those in EDGAR-HTAP v3. SSP5-8.5 has much lower NOx emissions over South Korea and SMA, compared to EDGAR-HTAP v3. "No SMA" simulations with WRF-Chem may help estimate the uncertainty in the simulated $O_3$ originated from the emission discrepancy. "No SMA" increases $O_3$ concentrations over South Korea (SMA) by 1.87 (22.1) ppb.

We acknowledge the emission differences for the two models. However, we are conducting research utilizing CAM-Chem and WRF-Chem separately for different purposes. Separate papers for different models are in review and in preparation. In this study, we utilized the results from CAM-Chem to analyze the contribution of stratospheric ozone to tropospheric ozone and use WRF-Chem model to investigate the impacts of anthropogenic emission changes on local and regional air quality. Thus, one-to-one comparison of the two models are beyond the scope of this study.

Table R9. The area sum emissions in Eastern China (27.7-40Ṅ, 115-123Ė), South Korea (34.5-38Ṅ, 126-130Ė), and Seoul Metropolitan Area (SMA: 37.2-37.8Ṅ, 126.5-127.3Ė) in May 2016 for NOx.

| NOx emission (unit = mols/s) | SSP5-8.5 | EDGAR-HATP v3 | EDGAR-HATP v2 | KORUS v5 |
|---|---|---|---|---|
| China | 6638 | 9034 | 10063 | 5482 |
| South Korea | 303 | 1097 | 990 | 886 |
| SMA | 26 | 214 | 196 | 191 |

Feng, L., Smith, S. J., Braun, C., Crippa, M., Gidden, M. J., Hoesly, R., Klimont, Z., van Marle, M., van den Berg, M., and van der Werf, G. (2020). The generation of gridded emissions data for CMIP6. Geoscientific Model Development, 13, 461-482, doi.org/10.5194/gmd-13-461-2020

6) Page 10, line 11-12: For greater accuracy, I suggest changing "… increases by 1-2 ppb yr-1 for most of cities and provinces across South Korea ..." to "… increases by 1.0-1.5 ppb yr-1 for most cities and provinces across South Korea ..."

→ We changed it to 1.0-1.5 ppb yr$^{-1}$.

7) Page 10, line 12-13: For greater accuracy, I suggest changing "The most of cities and provinces have the 4th highest MDA8 O3 higher than 70 ppb after 2010." to "In nearly all cities and provinces, the 4th highest MDA8 O3 has been higher than 70 ppb since 2010 or earlier."

→ Thank you for your suggestion. We replaced the original sentence by the one the reviewer suggested.

8) I suggest vertical lines be added to Figure 4 to separate the cities, provinces, and background sites from each other. Similarly for Figures 7 and 8. Also simplify the figure captions.

→ We noted cities, provinces, and background sites with labels and lines in Figure 4, 7, and 9.

The names of the location were redefined and were used consistently throughout the

manuscript.

9) The discussion illustrated in Figures 4, 5 and 7 is based on "exceedances"; however, I cannot find where "exceedance" is defined in the paper. Please define. (I assume it is a day when MDA8 ozone exceeds 70 ppb).

→ Agreed. The mistakes in the abstract were corrected. The definition of exceedances is

clarified in the abstract. In this study, exceedance is defined as hourly $O_3$ > 70 ppb.

10) Figure 5 needs to be clearly explained. If an exceedance is based on MDA8, how can there be a diurnal cycle, since there is only one MDA8 per day? Is this percent of days with ozone above 70 ppb in a given hour? I suggest using the same ordinate scale in all 3 graphs, so that the comparison is made easy for the reader. Also the general description of the sites included in the 3 graphs should be given; i.e., top = Seoul area, middle = secondary cities, bottom = remote sites.

→ Please see the reply above for minor point (9). We also used the same ordinate scale for

Figure 5. The general description of the sites is included in the figure caption as suggested by

the reviewer.

11) It seems that the information included in Table 2 and Figure 6 are identical; I suggest that Table 2 be eliminated.

→ Agreed. We deleted Table 2 in the original manuscript and moved to the Supporting

Information for the readers who may want to obtain the details.

12)    Please give units for the slopes in Table 3; confidence limits should be given for the derived slopes. Also please give the slopes for the background sites for comparison, if those data are available.

→ The units are shown in the table caption. The results from statistical analysis are included in

the revised manuscript. Because of discontinuous record of the data, the slopes for the

background sites are not shown.

13)    Figure 11 – x-axis labels have typo.

→ Corrected.

14)    Page 20 – Please define SMA

→ SMA (Seoul Metropolitan Area) was defined in Page 2 in the original manuscript.

References:
Parrish, D.D., et al. (2020), Zonal similarity of long-term changes and seasonal cycles of baseline ozone at northern mid-latitudes. J. Geophys. Res.: Atmos., doi: 10.1029/2019JD031908.

References

Feng, L., Smith, S. J., Braun, C., Crippa, M., Gidden, M. J., Hoesly, R., Klimont, Z., van Marle, M., van den Berg, M., and van der Werf, G. (2020). The generation of gridded emissions data for CMIP6. Geoscientific Model Development, 13, 461-482, doi.org/10.5194/gmd-13-461-2020

---

## Editor Decision (ED1)

Second review of "Changes in surface ozone in South Korea on diurnal to decadal time scale for the period of 2001-2021" by Si-Wan Kim et al.

Manuscript ID: https://doi.org/10.5194/acp-2022-788

**Summary:**

The authors have made marked improvements in the paper. However, the major revisions that I identified in my first review as necessary before this paper is can be published have not been adequately addressed. Until they are addressed, I cannot support publication of this paper. Those major revisions are addressed in further detail below. For the most part the minor issues identified have been addressed, but a few remaining are also listed below.

**Major issues:**

1) In my judgement, the authors must begin their observation-based analysis with a consideration of the ozone distribution that would be present in South Korea if there were no local continental influences, i.e., if observed concentrations were due to transported baseline ozone alone. In my first review I suggested how this might be approached. This consideration would then provide a basis for understanding the continental influences, both from local South Korean emissions and from the Asian mainland emissions.

   However, the authors have not attempted this approach. Instead they argue that "It would not be straighforward to delineate background O3 (without continental influences) and to assess the impacts of local South Korean emissions and Asian mainland emissions by mainly analyzing observations for the complex atmospheric environment of South Korea." This argument is not adequate. In fact the background ozone is quite readily approximated to the degree of accuracy required. As I noted in my first review, Figure 5 of Parrish et al. (2020) shows that annual mean ozone is 30 to 40 ppb in the lower 1 km of the troposphere. Figure 6 of that paper shows that there is a small seasonal cycle ($\sim \pm 5$ ppb) in the background O3 outside of the marine boundary layer. Thus, the surface concentration that would be expected in South Korea in the absence of continental emissions is ~35-45 ppb at the spring-summer seasonal maximum; this expectation is in close accord with the peak time ozone concentrations at city and province sites throughout South Korea. To my mind, this discussion must be the starting point for the discussion of O3 concentrations throughout South Korea.

   The authors also respond: "However, Figure R12 also illustrates various responses of surface ozone to emission scenarios in Seoul. It demonstrates that chemistry is an important factor to determine mean annual ozone in Seoul and other regions in South Korea. Therefore, we would like to avoid oversimplification of factors to determine the ozone in South Korea." However, Figure R12 shows only relatively small differences in mean annual ozone at the diurnal peak times, even in Seoul, the largest city in South Korea. The Control simulation gives a maximum of ~60 ppb. No Seoul emission simulation gives ~ 70 ppb and No China emission simulation gives ~47 ppb. This clearly emphasizes my point that the ~35-45 ppb expected for background only is an excellent starting point for the discussion of the local and regional South Korean influences.

2) In my first review I objected to the author's attempt to use the lower ozone concentrations in the 01-06 period to characterize transport of background ozone, because loss of ozone beneath the nocturnal inversion, both due to reaction with fresh local NOx emissions, but

also due to surface deposition, especially to vegetated surfaces. Thus, nighttime ozone concentrations do not provide direct information regarding transported baseline ozone.

However, if the authors insist upon inclusion of analysis of the data in the 01-06 period, it is essential to base that analysis of $Ox = O_3 + NO_2$ concentrations. The figure below is derived

[Figure]

from the Seoul data that the authors included in their Figure R9. The Ox concentrations are not affected by the reaction of $O_3$ with fresh local NOx emissions (but is affected by loss to surface deposition), providing further reactions to form $NO_3$, $N_2O_5$ and nitrate are not important. Ox recorded during the 01-06 period is a much more accurate indicator of transported baseline ozone than is $O_3$ itself.

**Figure 1.** The diurnal variations of observed $O_3$, $NO_2$ and Ox averaged for the simulation period, based on reading data from the authors' Figure R9.

3) In my first review I suggested that the background sites be included in Figure 3 and Table 1 in order to emphasize the similarity of the ozone concentrations throughout the country, and the predominant role played by transported baseline ozone. The authors have not made that inclusion; they argue that missing data require that exclusion. However, the data are missing only for periods of only 1 to 4 months out of 21 years. Such minor periods of missing data do not significantly compromise trend analyses. The great value of the background sites for comparison with other south Korean sites is shown in the authors' Figure R11 which clearly demonstrates that peak, mid-day mean ozone concentrations are very similar (in both the observations and model simulation) at the largest South Korean urban area (Seoul) and one of the background sites (Gosung). In my view, it is imperative that all tables, figures and discussion clearly address the 7 cities, 9 provinces, and 2 background sites in a consistent manner to the fullest extent possible. I do understand that measurements of precursor species may not be available from background sites, and thus cannot be included. However, on lines 12-15 of page 30 in the Conclusions Section the authors state: "The 4th highest maximum daily 8-hour average (MDA8) ozone concentrations showed an increasing trend in all cities, most provinces, and background sites during this period, with a yearly increase of 1-2 ppb." Certainly the data from the background sites must be shown in the paper to support that conclusion. The increasing trend at background sites should also be included in the similar sentence in the Abstract (lines 8-9, page 2).

4) In my first review I mentioned that one reason the 01-06 LT ozone is higher in the spring is that the nocturnal inversion is tighter in the summer, so ozone loss at night is more pronounced in summer than in spring. The authors disagree, and respond that they "don't think that there are clear mechanisms driving differences in nocturnal inversion between spring and summer." I am not a meteorologist, but is has been my understanding that atmospheric stability is at a minimum in spring and significantly greater in summer – hence tighter inversion layers in summer. I believe that this may be clearly shown in the authors' Table R5, which shows that mean wind velocity is generally higher in spring than in summer

at all of the stations considered. The nocturnal inversion is much more sharply defined in calm than in windy conditions. Regardless of my meager meteorological expertise, if the authors really wish to compare nighttime ozone concentrations between spring and summer, it is their responsibility to demonstrate that the nocturnal boundary layer dynamics do not change between seasons to a degree large enough to affect that comparison – they have not provide that demonstration. Again, given the very local processes that determine the 01-06 LT ozone, including the unaccounted for effects of surface deposition, and for the reasons discussed in my point 2) above, I recommend that the authors eliminate the discussion of this nighttime ozone; this discussion does not seem to be central to the main points of the paper. Notably, the analysis discussed in Section 3.2.1 is not mentioned either in the Abstract or in Section 4 Conclusions.

5) The Conclusions section requires some modifications. Specifically:
- On page 31, lines 4-7 has the sentence: "The majority of ozone exceedances occurred between 16-19 LT (4-7 PM). Interestingly, exceedances also occurred frequently at night in background sites such as Gosung and Ulleung Island, indicating a strong influence of long-range transport on surface ozone levels in these locations." I suggest that the final phrase "in these locations." be changed to "over South Korea". The only reason that nighttime exceedances are not seen at most sites in South Korea is that loss of ozone to fresh NOx emissions at night reduce the ozone concentrations below the concentration of transported background ozone.
- Page 31, lines 13-14 has the sentence: "Therefore, it is not clear what drove increase of ozone exceedances over South Korea from P1 to P2." I disagree; I believe that it is abundantly clear that the increases in ozone exceedances in South Korea can be attributed to increased anthropogenic emissions in China. This certainly follows from the following sentence which states: "We observed significant reductions in ozone exceedances across all monitoring sites in South Korea during the spring of the COVID-19 pandemic (period P3, 2020-2021), which was attributed to decreased anthropogenic activities and subsequent lower emissions in both China and South Korea." This discussion should be clarified.

**Minor issues:**

1) Table 1: I presume that the tabulated ozone concentrations are means over all days in each season. This should be stated in the Table caption.

2) The description of the two models indicates that different anthropogenic emission inventories are used in the two models. The authors have now given a brief discussion regarding how well these inventories compare (their Table R9), but they include that discussion only in their response to the reviewers' comments. This is very important discussion; it should also be included in the paper itself, possible in the Supplementary Material.

3) Pg. 14, line 1 contains the term "Asian emissions". I think more specificity is required here. Perhaps "Chinese emissions" or East Asian emissions".

---

## Author Response (AR2)

**Reply to the reviewer 3 of the revised manuscript "Changes in surface ozone in South Korea on diurnal to decadal time scale for the period of 2001-2021"**

Summary:
The authors have made marked improvements in the paper. However, the major revisions that I identified in my first review as necessary before this paper is can be published have not been adequately addressed. Until they are addressed, I cannot support publication of this paper. Those major revisions are addressed in further detail below. For the most part the minor issues identified have been addressed, but a few remaining are also listed below.

Thank for reviewing the manuscript again. We appreciate the reviewer's comments improving this manuscript. We made our efforts to address the concerns and suggestions raised by the reviewer to the best of our abilities. Our replies are written below in black.

Major issues:

1)      In my judgement, the authors must begin their observation-based analysis with a consideration of the ozone distribution that would be present in South Korea if there were no local continental influences, i.e., if observed concentrations were due to transported baseline ozone alone. In my first review I suggested how this might be approached. This consideration would then provide a basis for understanding the continental influences, both from local South Korean emissions and from the Asian mainland emissions.

However, the authors have not attempted this approach. Instead they argue that "It would not be straightforward to delineate background O3 (without continental influences) and to assess the impacts of local South Korean emissions and Asian mainland emissions by mainly analyzing observations for the complex atmospheric environment of South Korea." This argument is not adequate. In fact the background ozone is quite readily approximated to the degree of accuracy required. As I noted in my first review, Figure 5 of Parrish et al. (2020) shows that annual mean ozone is 30 to 40 ppb in the lower 1 km of the troposphere. Figure 6 of that paper shows that there is a small seasonal cycle (~ ± 5 ppb) in the background O3 outside of the marine boundary layer. Thus, the surface concentration that would be expected in South Korea in the absence of continental emissions is ~35-45 ppb at the spring-summer seasonal maximum; this expectation is in close accord with the peak time ozone concentrations at city and province sites throughout South Korea. To my mind, this discussion must be the starting point for the discussion of O3 concentrations throughout South Korea.

The authors also respond: "However, Figure R12 also illustrates various responses of surface ozone to emission scenarios in Seoul. It demonstrates that chemistry is an important factor to determine mean annual ozone in Seoul and other regions in South Korea. Therefore, we would like to avoid oversimplification of factors to determine the ozone in South Korea." However, Figure R12 shows only relatively small differences in mean annual ozone at the diurnal peak

times, even in Seoul, the largest city in South Korea. The Control simulation gives a maximum of ~60 ppb. No Seoul emission simulation gives ~ 70 ppb and No China emission simulation gives ~47 ppb. This clearly emphasizes my point that the ~35-45 ppb expected for background only is an excellent starting point for the discussion of the local and regional South Korean influences.

→ The reviewer suggested an interesting approach to analyze surface ozone over South Korea. We value the reviewer's opinion. Current manuscript introduced several metrics emphasizing maximum values to mean values over various time scales characterizing surface ozone over South Korea, which stimulates multiple studies in the future. A study focusing on the mean status (background value) of ozone and its deviations in South Korea that was suggested by the reviewer would be much appreciated. Unfortunately, because of large extent of current manuscript, this topic and approach suggested by the reviewer should be addressed in another manuscript.

2)      In my first review I objected to the author's attempt to use the lower ozone concentrations in the 01-06 period to characterize transport of background ozone, because loss of ozone beneath the nocturnal inversion, both due to reaction with fresh local NOx emissions, but also due to surface deposition, especially to vegetated surfaces. Thus, nighttime ozone concentrations do not provide direct information regarding transported baseline ozone. However, if the authors insist upon inclusion of analysis of the data in the 01-06 period, it is essential to base that analysis of Ox = O3 + NO2 concentrations. The figure below is derived from the Seoul data that the authors included in their Figure R9. The Ox concentrations are not affected by the reaction of O3 with fresh local NOx emissions (but is affected by loss to surface deposition), providing further reactions to form NO3, N2O5 and nitrate are not important. Ox recorded during the 01-06 period is a much more accurate indicator of transported baseline ozone than is O3 itself.

[Figure]

Figure 1. The diurnal variations of observed O3, NO2 and Ox averaged for the simulation period, based on reading data from the authors' Figure R9.

→ We have analyzed Ox. It is insightful to examine Ox values and their changes with seasons. Actually, $NO_2$ concentrations in spring are consistently, distinctively higher than those in summer, which made the comparison of Ox between the two seasons somewhat complicated. $NO_2$ concentrations at peak and base time in spring and summer are summarized in Table R1. Chemical lifetime of NOx is larger in spring than summer. This affects $NO_2$ levels during nighttime. It seems that the reviewer's concerns about seasonal changes in boundary layer height and deposition during nighttime are minor issues compared to the changes in chemical lifetime. See also our responses to major comments 4) below. Furthermore, $NO_2$ concentrations vary substantially with locations, which made the comparison of seasonal Ox differences among different locations somewhat complicated. In Table R2, $O_X$ concentrations at peak and base time in spring and summer are summarized. Overall, Ox during spring is higher than that during summer and nighttime differences are prominent, which is similar to the conclusions from the analysis of $O_3$. Therefore, we do not change the original content. **Because we think both $O_3$ and Ox are useful, the analysis of $NO_2$ and Ox are now included in the Supporting Information following the reviewer's request (SI1, Table S6 and S7). Please refer to the changes in P15, L20 – P16, L4 in the final revised manuscript.**

Table R1. Spring and summer $NO_2$ concentrations in Korean metropolitan cities and provinces. Both peak time (10-20 LT) and base time (01-06 LT) averages are shown. Differences in concentrations between spring and summer ($NO_{2\ spring} - NO_{2\ summer}$) are in the parenthesis.

| Location | | Peak time | Base time |
|---|---|---|---|
| | | Spring / Summer (difference) | Spring / Summer (difference) |
| City | Seoul (SUL) | 36.1 / 29.5 (6.6) | 36.9 / 24.6 (12.3) |
| | Incheon (INC) | 30.5 / 24.3 (6.3) | 28.5 / 19.7 (8.8) |
| | Daejeon (DJN) | 16.1 / 12.0 (4.1) | 20.5 / 13.6 (6.9) |
| | Gwangju (GWJ) | 19.8 / 13.4 (6.4) | 19.0 / 10.4 (8.6) |
| | Busan (BSN) | 20.9 / 13.9 (7.0) | 24.3 / 14.5 (9.8) |
| | Ulsan (ULS) | 23.9 / 20.1 (3.8) | 21.2 / 15.3 (5.9) |
| | Daegu (DGU) | 24.3 / 16.6 (7.7) | 26.0 / 15.4 (10.6) |
| Province | Gyeonggi-do (GGI) | 27.3 / 20.8 (6.5) | 31.2 / 20.7 (10.5) |
| | Chungcheongbuk-do (CCB) | 18.9 / 13.6 (5.3) | 20.8 / 13.7 (7.1) |
| | Chungcheongnam-do (CCN) | 17.4 / 12.9 (4.5) | 18.0 / 11.9 (6.1) |
| | Jeollabuk-do (JLB) | 15.1 / 11.2 (3.9) | 14.1 / 9.2 (4.9) |
| | Jeollanam-do (JLN) | 17.4 / 14.0 (3.4) | 14.1 / 10.5 (3.6) |
| | Jeju Island (JEJ) | 12.0 / 8.5 (3.5) | 8.5 / 6.0 (2.5) |
| | Gyeongsangnam-do (GSN) | 18.6 / 15.0 (3.6) | 18.9 / 13.3 (5.6) |
| | Gyeongsangbuk-do (GSB) | 17.9 / 13.4 (4.5) | 20.5 / 13.6 (6.9) |
| | Gangwon-do (GWO) | 14.0 / 9.9 (4.1) | 13.4 / 8.1 (5.3) |
| Background | Ulleung Island (ULL) | 3.4 / 2.7 (0.7) | 3.3 / 2.8 (0.5) |
| | Gosung (GSU) | 4.5 / 2.6 (1.9) | 4.6 / 2.8 (1.8) |

Table R2. Spring and summer $O_x$ (=$O_3$+$NO_2$) concentrations in Korean metropolitan cities and provinces. Both peak time (10-20 LT) and base time (01-06 LT) averages are shown. Differences in concentrations between spring and summer ($O_{X\ spring} - O_{X\ summer}$) are in the parenthesis.

| Location | | Peak time | Base time |
|---|---|---|---|
| | | Spring / Summer (difference) | Spring / Summer (difference) |
| City | Seoul (SUL) | 70.5 / 65.1 (5.4) | 57.5 / 42.1 (15.4) |
| | Incheon (INC) | 65.1 / 57.4 (7.7) | 53.6 / 39.9 (13.7) |
| | Daejeon (DJN) | 57.3 / 49.0 (8.3) | 43.3 / 32.7 (10.6) |
| | Gwangju (GWJ) | 59.7 / 48.8 (10.9) | 47.5 / 34.4 (13.1) |
| | Busan (BSN) | 61.2 / 48.1 (13.1) | 54.6 / 36.9 (17.7) |
| | Ulsan (ULS) | 62.6 / 53.5 (9.1) | 47.0 / 34.0 (13.0) |
| | Daegu (DGU) | 63.9 / 54.2 (9.7) | 50.0 / 35.0 (15.0) |
| Province | Gyeonggi-do (GGI) | 64.8 / 59.3 (5.5) | 52.0 / 38.7 (13.3) |
| | Chungcheongbuk-do (CCB) | 61.0 / 53.0 (8.0) | 45.6 / 34.3 (11.3) |
| | Chungcheongnam-do (CCN) | 58.7 / 50.6 (8.1) | 47.6 / 35.0 (12.6) |
| | Jeollabuk-do (JLB) | 53.4 / 46.2 (7.2) | 40.8 / 32.8 (8.0) |
| | Jeollanam-do (JLN) | 59.9 / 49.1 (10.8) | 47.1 / 34.6 (12.5) |
| | Jeju Island (JEJ) | 61.3 / 42.7 (18.6) | 52.5 / 34.6 (17.9) |
| | Gyeongsangnam-do (GSN) | 62.9 / 55.0 (7.9) | 47.8 / 35.2 (12.6) |
| | Gyeongsangbuk-do (GSB) | 63.0 / 51.4 (11.6) | 49.0 / 34.2 (14.8) |
| | Gangwon-do (GWO) | 58.5 / 49.0 (9.5) | 41.3 / 28.6 (12.7) |
| Background | Ulleung Island (ULL) | 60.0 / 46.6 (13.4) | 59.2 / 45.9 (13.3) |
| | Gosung (GSU) | 62.8 / 45.7 (17.1) | 62.7 / 47.9 (14.8) |

3)      In my first review I suggested that the background sites be included in Figure 3 and Table 1 in order to emphasize the similarity of the ozone concentrations throughout the country, and the predominant role played by transported baseline ozone. The authors have not made that inclusion; they argue that missing data require that exclusion. However, the data are missing only for periods of only 1 to 4 months out of 21 years. Such minor periods of missing data do not significantly compromise trend analyses. The great value of the background sites for comparison with other south Korean sites is shown in the authors' Figure R11 which clearly demonstrates that peak, mid-day mean ozone concentrations are very similar (in both the observations and model simulation) at the largest South Korean urban area (Seoul) and one of the background sites (Gosung). In my view, it is imperative that all tables, figures and discussion clearly address the 7 cities, 9 provinces, and 2 background sites in a consistent manner to the fullest extent possible. I do understand that measurements of precursor species may not be available from background sites, and thus cannot be included. However, on lines 12-15 of page 30 in the Conclusions Section the authors state: "The 4th highest maximum daily 8-hour average (MDA8) ozone concentrations showed an increasing trend in all cities, most provinces, and background sites during this period, with a yearly increase of 1-2 ppb." Certainly the data from the background sites must be shown in the paper to support that conclusion. The

increasing trend at background sites should also be included in the similar sentence in the Abstract (lines 8-9, page 2).

→ Agreed. We added the trends of the 4th highest MDA8 ozone from 2001 to 2021 for the background sites to Table 1. Although there are missing data during the ozone season, the 4th highest MDA8 ozone values were calculated for these sites. Because of this limitation and low certainty, the plots of the trend of the 4th highest MDA8 ozone for the background sites are not presented in Figure 2 or 3. In Table 1, the trends for 2001-2019 are also shown for the background sites because the increasing trends for this shorter period (prior to the COVID-19 pandemic) are more certain. Averages of ozone, $NO_2$, and Ox for the background sites at peak and base time for spring and summer are added to Table 2 and Supporting Information (Table S6 and S7). Because of discontinuity of the data, the trend of $NO_2$ and CO for the background sites are not shown in Table 3 and 4 in the final revised manuscript. We note this information in the Table captions for Table 1, 3, and 4. **Please refer to the changes in P13, L20 – P14, L8 and P15 L5-6, P15 L 11-13 and Table 1 and 2 in the final revised manuscript and Table S6 and S7 in SI1.**

4) In my first review I mentioned that one reason the 01-06 LT ozone is higher in the spring is that the nocturnal inversion is tighter in the summer, so ozone loss at night is more pronounced in summer than in spring. The authors disagree, and respond that they "don't think that there are clear mechanisms driving differences in nocturnal inversion between spring and summer." I am not a meteorologist, but is has been my understanding that atmospheric stability is at a minimum in spring and significantly greater in summer – hence tighter inversion layers in summer. I believe that this may be clearly shown in the authors' Table R5, which shows that mean wind velocity is generally higher in spring than in summer at all of the stations considered. The nocturnal inversion is much more sharply defined in calm than in windy conditions. Regardless of my meager meteorological expertise, if the authors really wish to compare nighttime ozone concentrations between spring and summer, it is their responsibility to demonstrate that the nocturnal boundary layer dynamics do not change between seasons to a degree large enough to affect that comparison – they have not provided that demonstration. Again, given the very local processes that determine the 01-06 LT ozone, including the unaccounted for effects of surface deposition, and for the reasons discussed in my point 2) above, I recommend that the authors eliminate the discussion of this nighttime ozone; this discussion does not seem to be central to the main points of the paper. Notably, the analysis discussed in Section 3.2.1 is not mentioned either in the Abstract or in Section 4 Conclusions.

→ Understanding of stably stratified turbulence is limited and parameterization of stable boundary layer and its height is challenging (Cuxart et al., 2006; Fernando and Weil, 2010). There are currently several issues in practical applications of the state-of-the-art model and observations in association with stable boundary layer height. The boundary layer height from typical global and mesoscale models are not readily comparable to the observed nocturnal boundary layer heights.  For example, the stable boundary layer from ceilometer reflects the observed vertical profiles of aerosols rather than thermal and mechanical turbulence structure. Meanwhile, the stable boundary layer height from WRF model output is subject to the definition of the model critical Richardson number and is not readily comparable to observations of boundary layer height during nighttime. Detailed discussions about the definitions of stable boundary layer height that is determined by stability and wind shear are beyond the scope of this paper.

According to Kim and Park (1996), seasonal changes in nighttime wind speed, stability, and friction velocity (a measure of turbulence intensity) between spring and summer over Seoul Metropolitan Area were small. The seasonal changes in nighttime dry deposition velocity of $NO_2$ over this area were also very small. Please refer to the figures below (Figure 5, 6, 7, and 10 from Kim and Park, 1996). However**, in final the revised manuscript, we included the $NO_2$ and Ox analysis in the Supporting Information following the reviewer's comments (SI1 Table S6 and S7).** Please see also our response to the reviewer's major comment 2).

[Figure]

**Fig. 5.** The dirunal variations of seasonal mean (a) wind speed (m/s) and (b) friction velocity (m/s).

**Fig. 6.** The same as in Fig. 5 except for (a) solar radiation (W/m2) and (b) temperature difference between surface temperature and ground temperature (deg).

[Figure]

**Fig. 7.** The diurnal variations of mean (a) $SO_2$ and (b) $NO_2$ concentration (ppb).

**Fig. 10.** The same as in Fig. 7 except for (a) $SO_2$ and (b) $NO_2$ dry deposition velocity (cm/s)

5)      The Conclusions section requires some modifications. Specifically:
•       On page 31, lines 4-7 has the sentence: "The majority of ozone exceedances occurred between 16-19 LT (4-7 PM). Interestingly, exceedances also occurred frequently at night in background sites such as Gosung and Ulleung Island, indicating a strong influence of long-range transport on surface ozone levels in these locations." I suggest that the final phrase "in these locations." be changed to "over South Korea". The only reason that nighttime exceedances are not seen at most sites in South Korea is that loss of ozone to fresh NOx emissions at night reduce the ozone concentrations below the concentration of transported background ozone.

→ We agree with you about underlying causes for reduced ozone during nighttime at most sites in South Korea. However, here, we wanted to highlight the Gosung and Ulleung Island sites for frequent nighttime exceedances, which is different from highly polluted cities like Seoul. Therefore, we kept the original content.

•       Page 31, lines 13-14 has the sentence: "Therefore, it is not clear what drove increase of ozone exceedances over South Korea from P1 to P2." I disagree; I believe that it is abundantly clear that the increases in ozone exceedances in South Korea can be attributed to increased anthropogenic emissions in China. This certainly follows from the following sentence which states: "We observed significant reductions in ozone exceedances across all monitoring sites in South Korea during the spring of the COVID-19 pandemic (period P3, 2020-2021), which was attributed to decreased anthropogenic activities and subsequent lower emissions in both China and South Korea." This discussion should be clarified.

→ As the reviewer mentioned, it is likely that the increases in ozone exceedances in South Korea from P1 to P2 are attributed to increased anthropogenic emissions in China. However, we would like to conclude this after completion of our long-term model simulations and analysis covering this period. And it would be also important to address the impact of the climate changes (e.g., large positive temperature anomaly in 2010's) on changes in ozone concentrations over South Korea. Following the reviewer's comments, in the final revised manuscript, we stated "The observed increase in ozone exceedances from P1 to P2 in South Korea may be partially attributed to the rise in anthropogenic emissions originating from China. More modeling experiments covering the P1 to P2 period would help identify the main factors behind the ozone increases. It is important to investigate not only changes in anthropogenic emissions but also the impact of climate change on ozone variations during this period". **Please refer to changes in P32 L10-L15 in the final revised manuscript.**

Minor issues:

1)   Table 1: I presume that the tabulated ozone concentrations are means over all days in each season. This should be stated in the Table caption.

→ We clarified this information. A sentence "Data during typical ozone season (May-September) are used" are added to **the caption in Table 1**.

2)   The description of the two models indicates that different anthropogenic emission inventories are used in the two models. The authors have now given a brief discussion regarding how well these inventories compare (their Table R9), but they include that discussion only in their response to the reviewers' comments. This is very important discussion; it should also be included in the paper itself, possible in the Supplementary Material.

→ Following the reviewer's suggestions, we included the Table R3 in this reply to **the Supporting Information (SI1 Table S5)**. The discussions are shown in the note attached to the table.

3)   Pg. 14, line 1 contains the term "Asian emissions". I think more specificity is required here. Perhaps "Chinese emissions" or East Asian emissions".

→ Corrected to "East Asian emissions". **Refer to P14, L9 in the final revised manuscript.**

**References**

Cuxart, J., and Coauthors, 2006: Single-column model intercomparison for a stably stratified atmospheric boundary layer. Bound.-Layer Meteor., 118, 273–303, doi:10.1007/s10546-005-3780-1.

Fernando, H. J. S., and J. C. Weil, 2010: Whither the stable boundary layer? A shift in the research agenda. Bull. Amer. Meteor. Soc., 91, 1475–1481, doi:10.1175/2010BAMS2770.1.

Kim, S.-W. and S.-U. Park, 1996: Estimation of dry acidic deposition in the Seoul Metropolitan Area, Journal of Korean Meteorological Society, 32, 2 (In Korean).

**Reply to the reviewer 4 of the revised manuscript "Changes in surface ozone in South Korea on diurnal to decadal time scale for the period of 2001-2021"**

We thank you for reviewer's time and helpful comments. Our replies are written in black.

The second version of the manuscript has improved, and most of the referees' comments have been correctly addressed by the authors.

I think that a figure presenting the atmospheric transport is missing. While a Lagrangian analysis would be valuable but maybe out of the scope of this paper, one possibility is to overlay on maps of Figure 9 the seasonal average wind vectors to have a better sense of the mean transport in spring and summer.

→ We added the wind vectors for each season in **Figure 9 and Supporting Information (SI3 Figure S2 and S5)**. We have a plan to pursue a Lagrangian analysis for another manuscript.

Concerning the conclusion, the last sentences present some perspectives on how to improve our understanding of the ozone trends in South Korea. It is my opinion that a network of ozonesondes at a few key locations (Seoul, Gosung, a background site) with the capability of a weekly launch would be important to understand the evolution of ozone for the next decade. It will be complementary (and much cheaper) than a large field campaign like KORUS-AQ . If the authors agree, they could add a few sentences on the value of monitoring ozone with ozonesonde launches on a regular basis.

→ Agreed. We added "Monitoring ozone within the boundary layer and at higher altitudes is crucial for enhancing our understanding of ozone trends in South Korea. A network of ozonesondes at multiple sites with the capability of weekly launches would be a valuable complement to a large field campaign" to the **last paragraph of the final revised manuscript**.

Technical comments:
I don't understand Figure 4. What kind of ratio is presented here exactly?
→ First, we calculated the exceedances in each season. Here, the exceedances are defined as the fraction of hourly ozone concentration greater than 70 ppb among all available data. What is plotted is the ratio of exceedances in summer to exceedances in spring. In the final revised manuscript, we included this detail in **the caption of Figure 4**.

Figure 5: You should use the same naming convention as in the previous figures: province and background sites.
→ Agreed. We used the same naming convention in **Figure 5** as much as possible in the final revised manuscript.

---

## Author Response (AR3)

**Dear editor,**

**We revised the final manuscript following editor's comments. Thank you very much for your time and helpful comments. We also appreciate reviews that improved the manuscript.**

I concur with referee #3 that referring to the 10-20 LT period as the "Peak" and the 01-06 period as the "Base" can lead to confusion. It appears to me that the 01-06 LT period does not accurately represent the background air masses passing over South Korea. Otherwise, how can we explain the occurrence of average maximum values between 01-06 LT over the background sites? This observation is clearly demonstrated by WRF-Chem, where the "No Seoul" scenario exhibits higher ozone concentrations compared to the control run (with an increase of >20ppbv between 01-06 LT). Additionally, when considering Ox instead of O3, the average maximum values of Ox are observed over Seoul, while the lowest values are found over the background sites. As the authors mentioned, both high values of NOx and VOC in the limited regime suppress ozone production during the day.

Hence, I suggest renaming the "Peak" period as the "daytime period," and the "Base" period as the "nighttime period" to minimize confusion.

→ **Thank you for your suggestions. We changed "Peak" period to "Daytime" period and "Base" period to "Nighttime" period. Refer to page 15 and 16 (line 5-6) and Table 2 (page 53) in the final manuscript (a pdf file with track changes).**

Nonetheless, I believe a detailed analysis of the background air is unnecessary. The current paper already contains ample material to warrant publication. The examination of the contribution from background air variability, chemistry, and long-range transport will remain qualitative.

Upon addressing the following points, I will accept your manuscript for publication:

1) In Table 2, it would be beneficial to include the standard deviation of the mean values. This addition will provide a better understanding of the statistical significance of differences between summer and spring, as well as differences between ground sites.

→ **In Table 2 in the final manuscript, standard deviations are included (Table 2) and discussed in the main text (page 15). Due to changes in calculation method, numbers (generally last decimal digits) were changed.**

2) In section 3.4.3, you mention that reducing NOx and CO would positively impact ozone levels in spring. This is indeed accurate, but it's worth noting the reverse effect on some ground sites during summertime.

→ **We agree with the editor. In the final manuscript, we mentioned the reverse effect. In page 31 line 9-11(a pdf file with track changes), we added "However, our study also indicates that the ozone exceedances notably increased in SMA and Chungcheongnam-do (where large mobile, industrial, and power plant emission sources are located) during the summer of the COVID-19 pandemic".**

3) In the Conclusion on page 32, lines 15-18, you discuss a reduction in ozone exceedance during COVID in springtime. It's important to also acknowledge the increase in ozone levels at some of the ground sites during summertime.

→ **We agree with the editor. In the final manuscript, we acknowledged the ozone increases during summertime. In page 33 line 6-8 (a pdf file with track changes), we added "It should be noted that ozone exceedances substantially increased in SMA and Chungcheongnam-do during the summer of the COVID-19 pandemic".**

**We also updated the reference in the final manuscript because Jeong et al. (2023) was published.**

**Jeong, Y., et al.: Influence of ENSO on tropospheric ozone variability in East Asia, _J. Geophys. Res_., 128, 16, https://doi.org/10.1029/2023JD038604, 2023.**

**The reference below will be posted soon for discussions.**

**Kim, K.-M., et al.: Sensitivity of the WRF-Chem v4.4 ozone, formaldehyde, and their precursor simulations to multiple bottom-up emission inventories over East Asia during the KORUS-AQ 2016 field campaign, 2023 (submitted to _Geosci. Model Dev._).**